# Articular surface interactions distinguish dinosaurian locomotor joint poses

Armita R. Manafzadeh [1,2,3] ✉, Stephen M. Gatesy [4] &
Bhart-Anjan S. Bhullar [2,3]

Our knowledge of vertebrate functional evolution depends on inferences about joint function in extinct taxa. Without rigorous criteria for evaluating joint articulation, however, such analyses risk misleading reconstructions of vertebrate animal motion. Here we propose an approach for synthesizing raycast-based measurements of 3-D articular overlap, symmetry, and congruence into a quantitative "articulation score" for any non-interpenetrating six-degree-of-freedom joint configuration. We apply our methodology to bicondylar hindlimb joints of two extant dinosaurs (guineafowl, emu) and, through comparison with in vivo kinematics, find that locomotor joint poses consistently have high articulation scores. We then exploit this relationship to constrain reconstruction of a pedal walking stride cycle for the extinct dinosaur *Deinonychus antirrhopus*, demonstrating the utility of our approach. As joint articulation is investigated in more living animals, the framework we establish here can be expanded to accommodate additional joints and clades, facilitating improved understanding of vertebrate animal motion and its evolution.

Vertebrate animal motion – from feeding to flying – relies on the mobility of synovial joints such as jaws, shoulders, and knees. Although these joints are complex organs composed of tissues including bone, cartilage, and ligaments, and are crossed by muscles and tendons[1], much of vertebrate functional morphology has relied on the assumption that bony articular shape in some way reflects the joint motion used by an animal in life [e.g.[2–7]]. Inferences about joint kinematics based on bones alone have thus heavily shaped our reconstructions of extinct animals – and, in turn, our fundamental understanding of vertebrate functional evolution [e.g.[8–24]].

Joint configurations (here defined as simultaneous excursions in all three rotational and all three translational degrees of freedom) are generally excluded from functional reconstructions when they result in bone-bone contact (i.e., "bony stops") or improper articulation (i.e., "disarticulation", "subluxation", or "misalignment"). However, whereas computational methodology for detecting bone-bone contact has grown significantly more reliable and reproducible in recent years, that for evaluating joint articulation has lagged behind (see a review by[25]). Current criteria for virtual articulation analyses: (1) remain subjective, depending on assumptions about whether a joint "looks right" or "fits"; (2) require manual evaluation, limiting the fraction of total possible joint configurations that can be tested; and/or (3) have not been ground-truthed using data collected about articular surface interactions from extant animals. Recently, we discovered that imposing bounds on joint translation in lieu of establishing an explicit definition of "proper articulation" either risks incorrectly eliminating rotational combinations (i.e., joint poses) routinely used in life (too strict) or precludes excluding very many poses at all (too permissive)[26]. In our opinion, this sobering conclusion calls into question even the most methodologically advanced computational analyses of fossil joints to date, challenging the validity of existing inferences about the in vivo behavior of extinct animals and representing a severe obstacle to the successful reconstruction of functional evolution.

[1]Yale Institute for Biospheric Studies, Yale University, New Haven, CT 06520, USA. [2]Department of Earth & Planetary Sciences, Yale University, New Haven, CT 06520, USA. [3]Yale Peabody Museum of Natural History, New Haven, CT 06520, USA. [4]Department of Ecology, Evolution, and Organismal Biology, Brown University, Providence, RI 02912, USA. ✉e-mail: armita.manafzadeh@yale.edu

Here we aim to establish an articulation analysis framework that enables rigorous, data-driven reconstruction of vertebrate animal motion from bones alone. Toward this end, we propose synthesizing quantitative, raycast-based measurements of 3-D articular surface overlap, symmetry, and congruence to calculate an "articulation score" (Fig. 1; Supplementary Movie 1) measuring the quality of joint articulation for any non-interpenetrating, static six-degree-of-freedom joint configuration. First, we analyze joint articulation across osteological estimates of joint mobility in two extant avian dinosaurs (the flighted neognath *Numida meleagris* [Helmeted Guineafowl] and the flightless paleognath *Dromaius novaehollandiae* [Common Emu]). By comparing our results against in vivo locomotor kinematics, we uncover a consistent relationship between articulation score and the joint poses actually used during terrestrial locomotion (i.e., locomotor joint poses). We then demonstrate the immediate utility of articulation analysis by applying these findings to a pedal reconstruction of the exceptionally well-preserved paravian dinosaur *Deinonychus antirrhopus*, whose locomotor morphology, although somewhat autapomorphic, broadly represents an intermediate state between the ancestral theropod form and the modified anatomy of living birds[27,28]. Finally, we discuss the power of our articulation analysis framework to transform functional reconstructions of additional joints and behaviors throughout the vertebrate tree.

## Results

### Extant dinosaurs

In agreement with our previous work[26], conservative, six-degree-of-freedom estimates of joint mobility based on bone-bone contact alone occupy much of rotational pose space and demonstrate no clear relationship with the joint poses used during a typical stride (Fig. 2a–b; Supplementary Fig. 1). However, mapping our articulation scores onto this mobility estimate enhances our knowledge of each pose beyond a

binary "possible" or "impossible" (see[29]) and uncovers spatial patterns in articular quality across pose space. Plotting experimentally derived walking and running poses (*n* = 11,505) on top of the results of articulation analysis reveals that true locomotor joint poses fall within the highest-scoring (articulation score = 95–100) band of pose space (Fig. 2c, d). Only the most flexed joint poses near mid-swing phase of highest-speed running (*n* = 221, or approximately 1.9% of all measured poses) deviate by up to five degrees in long-axis rotation into regions of pose space with articulation scores of 81 to 95. In other words, 98.1% of measured locomotor joint poses and 100% of all walking poses occupy regions of pose space with articulation scores of 95–100.

This relationship holds true not only for the guineafowl ankle joint figured in Fig. 2a–e, but also for the primary flexion-extension rotations of additional bicondylar avian hindlimb joints (the guineafowl third metatarsophalangeal joint and emu ankle and third metatarsophalangeal joints [Fig. 2f; Supplementary Figs. 1–2]), whose locomotor excursions likewise fall in regions of pose space with articulation scores of 95–100. We found that our articulation score distributions are robust to sensitivity analyses conducted using additional individuals, increased translational allowance, and alternative score formulas (Supplementary Figs. 3–5). Articulation analysis thus reliably captures differences in the pattern of articulation quality across pose space – and as a result, is able to successfully predict observed differences in in vivo locomotor joint excursions – from the shapes of static guineafowl and emu bones alone.

### Extinct dinosaur

Six-degree-of-freedom mobility estimates for the twelve pedal joints of the three main digits of *Deinonychus* (Fig. 3a) again occupied much of rotational pose space, whereas articulation analysis revealed a small high-scoring region for each joint (Fig. 3b; see also Supplementary Figs. 6–7). These high-scoring regions clearly capture the severely

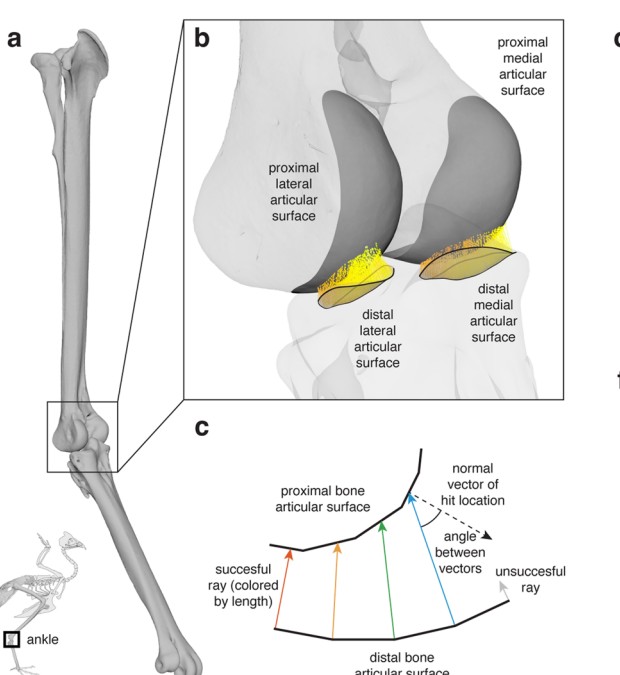

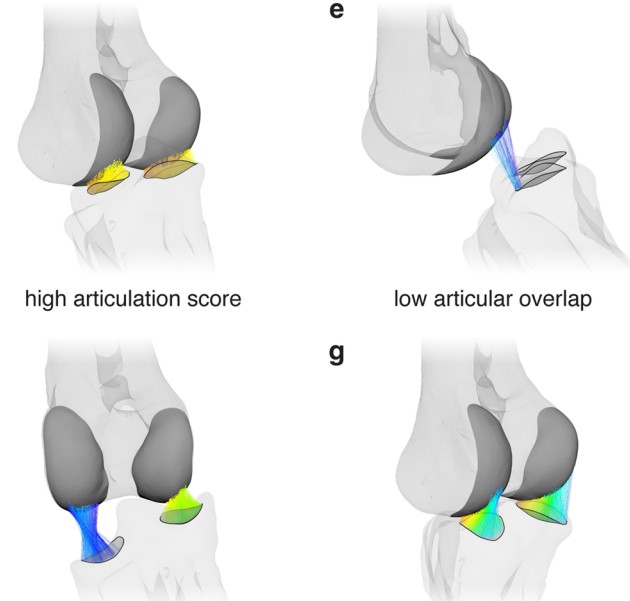

**Fig. 1 | The articulation analysis framework proposed here relies on raycast-based measurements of 3-D articular surface overlap, symmetry, and congruence.** At a given bicondylar joint (e.g., a right guineafowl ankle joint [**a**, anterolateral view]) the vertices of each concave articular surface on the distal bone (e.g., the tarsometatarsus) can be raycast along their normal vectors (**b**) and their relationships with each convex articular surface on the proximal bone (e.g., the tibiotarsus) can be assessed (**c**), diagrammatic representation in lateral view).

A high articulation score ((**d**), anterolateral view) requires high overlap, symmetry, and congruence, whereas low scores can result from a low number of successful rays (i.e., low overlap [**e**, lateral view]), poor symmetry in average successful ray length between the two halves of the joint (i.e., low symmetry [**f**, anterior view]), and/or high average angles between successful cast rays and normal vectors of their hit locations (i.e., low congruence [**g**, anterolateral view]). All rays colored by length from shortest (red) to longest (blue). See also Supplementary Movie 1.

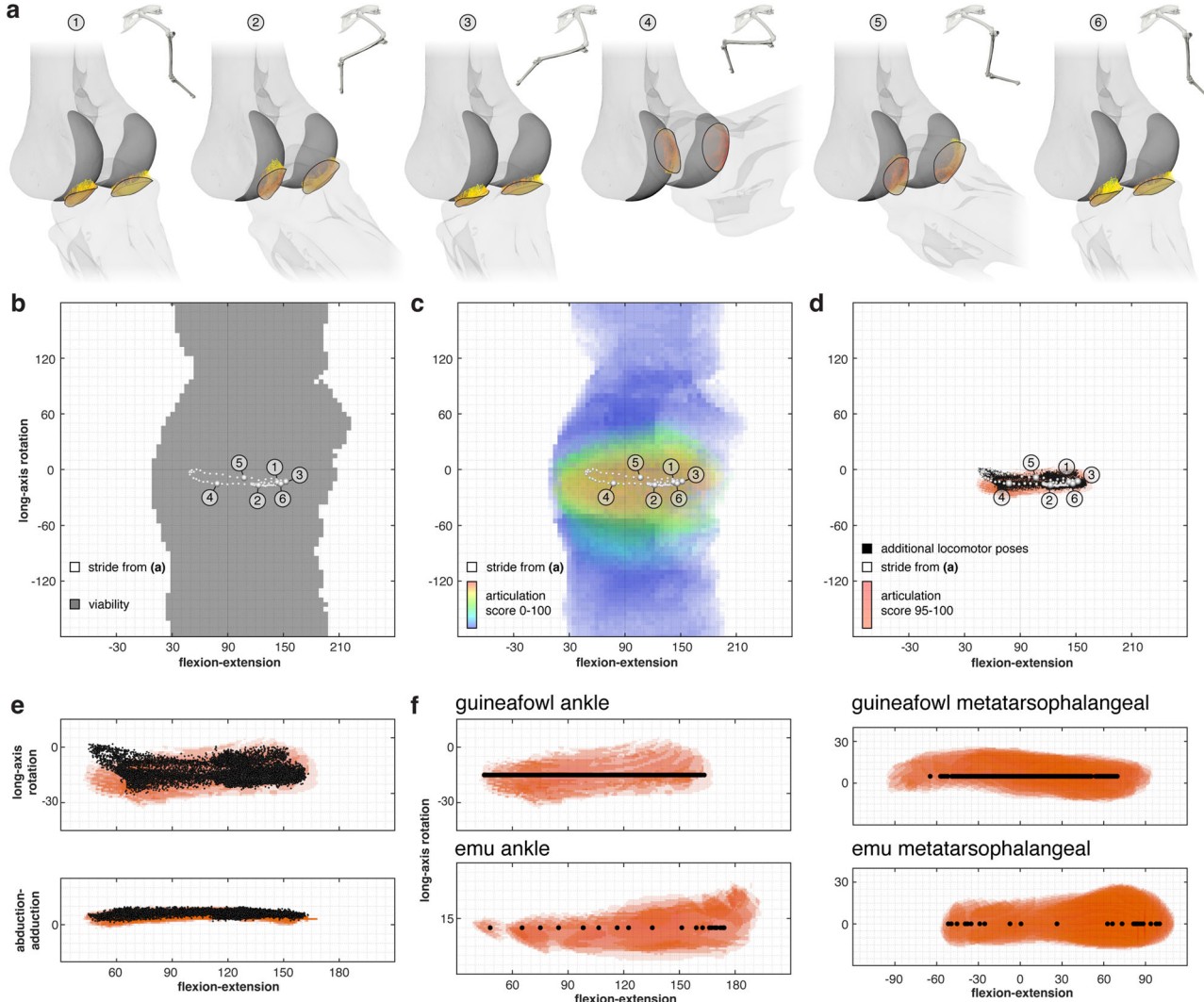

**Fig. 2 | Articulation analysis for extant dinosaurs reveals that articular surface interactions distinguish dinosaurian locomotor joint poses. a** Six ankle configurations from a guineafowl stride cycle measured using marker-based X-ray Reconstruction of Moving Morphology (XROMM[64] see also ref. 59 Fig. 1) in lateral view and their corresponding articular raycasts colored by length in anterolateral view. **b** A six-degree-of-freedom mobility estimate for the guineafowl ankle based on bone-bone contact alone occupies much of rotational pose space (3-D stride cycle poses superimposed). **c** Mobility estimate for the guineafowl ankle colored by articulation score from 0–100, analyzed at five-degree angular resolution (3-D stride cycle poses superimposed). **d** 11,505 additional XROMM-derived locomotor poses measured from the guineafowl ankle occupy a small subset of rotational pose

space and demonstrate strong correspondence with the subset of poses with articulation scores of 95–100, analyzed at one-degree angular resolution (3-D locomotor poses superimposed). **e** The relationship between locomotor poses and the highest-scoring region of pose space holds in 3-D, as shown in flexion-extension/long-axis rotation and flexion-extension/abduction-adduction views (3-D locomotor poses superimposed). **f** In vivo locomotor flexion-extension excursions (horizontal axis) measured from additional extant dinosaurian joints also fall within the highest-scoring (plotted long-axis rotation [vertical axis] values arbitrary; articulation score = 95–100, in red) regions of pose space, analyzed at one-degree angular resolution. See also Supplementary Figs. 1–2.

restricted locomotor long-axis rotation potential of the highly ginglymoid, specialized interphalangeal joints of the distinctive paravian pedal digit II, as well as broad similarities among the highest-scoring ranges of more generalized interphalangeal joints in digits III and IV. By using published guineafowl locomotor kinematics as a viable starting point for intrapedal coordination and then imposing articulation constraints using the morphology of all twelve *Deinonychus* pedal joints along with an articulation score cutoff (95) informed by our results from extant taxa (see Methods), we were able to reconstruct a six-degree-of-freedom walking stride cycle for *Deinonychus* that is consistent with evidence from articular surface interactions (Fig. 3c, see also Supplementary Movie 2–4). Given the extreme restriction of high-scoring regions for the interphalangeal joints of the second digit, we infer that digit II was held in a hyperextended posture during locomotion and underwent minimal interphalangeal motion during

any one stride. Considering the extent and curvature of the keratinous ungual sheath when preserved[30], such hyperextension might in fact have been required for the claw tip of *Deinonychus* to clear the ground.

## Discussion

The articulation analysis framework proposed here begins to formalize the implicit understanding of "proper articulation" that has underpinned centuries of osteological inferences about vertebrate animal motion. The three criteria contributing to our articulation score quantitatively capture various aspects of this intuition: for example, overlap can suffer as joints undergo large excursions in translation or flexion-extension such that articular surfaces slide past each other, symmetry can suffer as joints abduct or adduct such that one half of the joint moves farther apart than the other, and congruence can suffer as joints long-axis rotate such that the curvature of mating

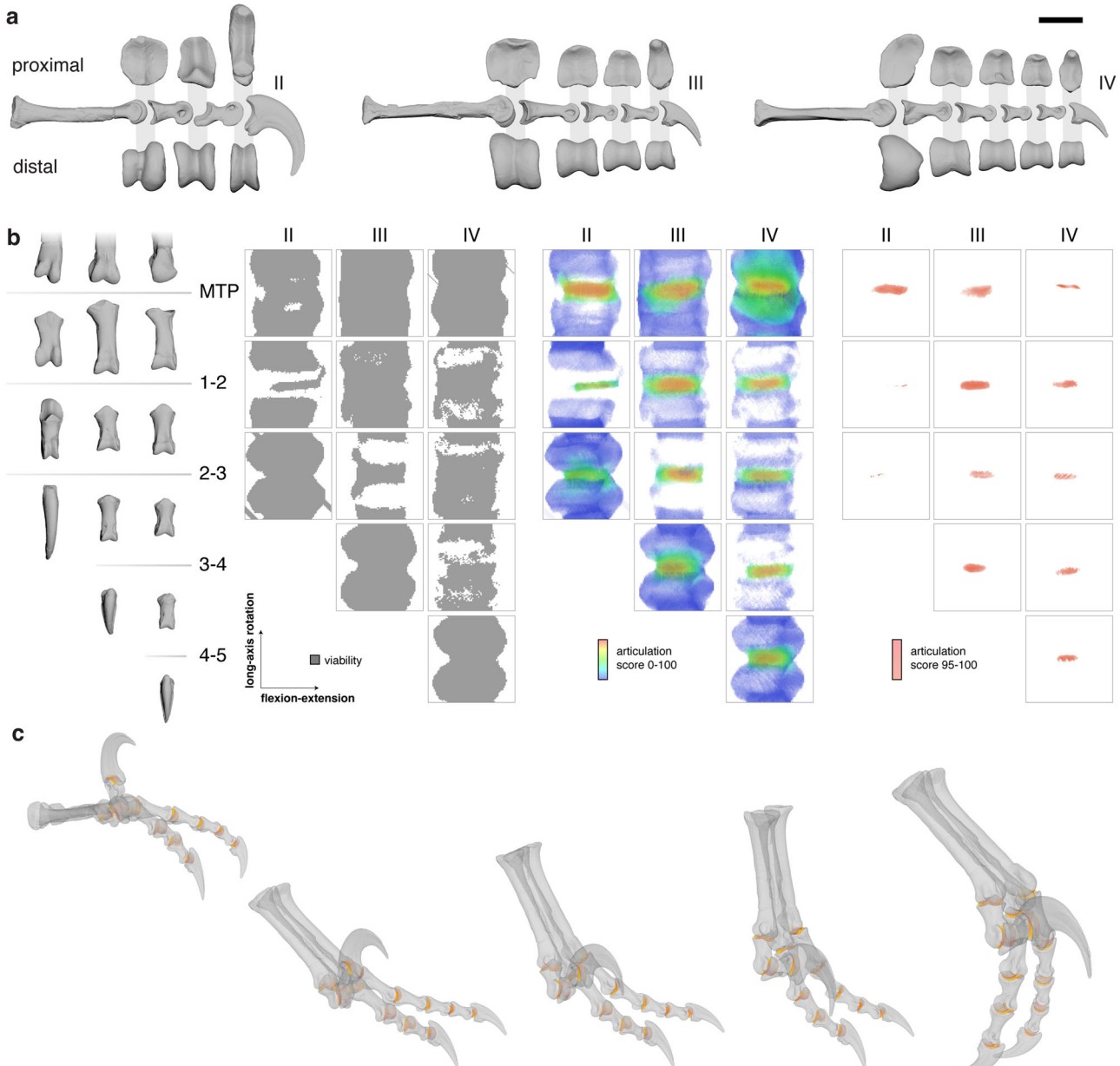

**Fig. 3 | Articulation analysis for *Deinonychus antirrhopus* enables data-driven reconstruction of a six-degree-of-freedom walking stride cycle. a** Left metatarsals and phalanges of the three weight-bearing digits of *Deinonychus* specimen YPM 5205 in medial view, with proximal and distal articular surface views, after[27]. Scale bar represents 4 cm for medial views and 2 cm for articular views. **b** Pedal elements of YPM 5205 in dorsal view, with results of metatarsophalangeal and interphalangeal articulation analyses colored by viability based on bone-bone contact alone (left) and articulation score (middle and right), analyzed at five-degree (left and middle) and one-degree (right) angular resolution. Flexion-extension and long-axis rotation axes both range from −180 to 180 degrees in all graphs. See also Supplementary Figs. 6–7. **c** Representative pedal configurations from a *Deinonychus* walking stride cycle reconstructed based on articulation analysis and their corresponding articular raycasts colored by length in anteromedial view; see also Supplementary Movie 2–4.

articular regions becomes mismatched. By contrast with many existing computational articulation criteria that rely on simple closest-point distances between mating bones (e.g., see[13]), our approach exploits information about local articular surface curvature to determine the direction of the cast rays from which our overlap, symmetry, and congruence parameters are measured (Fig. 1; see Methods). As a result, we are empowered to harness the 3-D complexity of osteological morphology directly preserved in the fossil record, and in museum specimens of inaccessible extant taxa, more fully than ever before.

Despite relying on data from static bony morphology alone – with no information or assumptions about the size, shape, or material properties of other articular tissues such as cartilage, ligaments,

menisci, or bursae – articulation analysis successfully distinguishes the subset of joint poses used during extant dinosaurian terrestrial locomotion from an otherwise expansive region of rotational pose space. Applying this finding to the foot of *Deinonychus* thus enabled us to constrain reconstruction of a potential six-degree-of-freedom walking stride cycle for a non-avialan paravian dinosaur, refining qualitative inferences made by previous workers based on observation and physical manipulation of fossil bones [e.g., 27, 32; see also Supplementary Fig. 8]. The Paraves node is especially significant in the evolution of birds because it appears to be the point on the avian stem at which some sort of flight had been achieved, and at which several major transformations of the trunk and locomotor skeleton had occurred[31].

Although it is large for a paravian and has some notable autapomorphic pedal features[32–34], *Deinonychus* is in several key ways conservative in its hindlimb anatomy: it resembles other deinonychosaurs on the one hand (including primitive, flighted taxa such as *Anchiornis*) and basal avialans, notably *Archaeopteryx*, on the other, in its general arrangement of digits and in having an elaborated, anatomically distinct pedal digit II[34,35]. It is not unlikely that all of these taxa had stride cycles at least somewhat similar to that of *Deinonychus*, including the habitually hyperextended posture of pedal digit II (even if degree of hyperextension varied as dictated by variation in joint shape).

As additional experimental data are collected from extant dinosaurs to determine what kinetic and kinematic conditions enable their joints to deviate from the highest-scoring region of pose space, the articulation analysis conducted here can be revisited to shed light upon the enigmatic function of the eponymous "terrible claw" of *Deinonychus*. Under the theoretical assumption that maintenance of high articulation score is advantageous for efficient synovial joint motion (see[36]), our data tentatively lend stronger support to a rigid-digit stabbing or pinning function[34,37–39] than to a more arc-like slashing or digging function of the claw[27,32,40–43], because the latter requires flexion of the proximal interphalangeal joint into poses with relatively low articulation scores. We note that stabbing is also directly supported by the position of the extraordinary *Velociraptor* specimen IGM 100/25, which is preserved in a typical avian kicking position, its claw embedded in the neck of a specimen of the small ceratopsian *Protoceratops*. Although this example highlights the future power of articulation analysis, we caution that further investigation of pose space occupation in extant animals is critical before articulation data are used to formally test non-locomotor behavioral hypotheses.

The tripartite overlap-symmetry-congruence approach we propose for bicondylar joints can readily be modified to accommodate analysis of additional joint morphologies, such as planar, saddle, or ball-and-socket joints, from throughout the vertebrate Bauplan (see Supplementary Fig. 9), and it is our hope that future workers will adopt, test, critique, and improve both the general methodology and the specific formula we propose here. Comparing articulation score distributions against in vivo kinematics for further joints and taxa will illuminate the broader utility of articulation analysis, facilitating the widespread application of these data in vertebrate functional reconstruction through integration with existing information from trackways [e.g.[44]], bone microstructure [e.g.[45]], musculoskeletal simulation [e.g.[16,46,47]], and robotic modeling [e.g.[18,48,49]]. In the process, this work will also place ongoing orthopedic research on human joint articulation [e.g.[50–55]] within its broader evolutionary context.

Exploring joint motion within the framework established here will allow us to more explicitly characterize how specific differences in articular form (e.g., as quantified by geometric morphometrics or statistical shape modeling) correlate with differences in articular function – an essential first step in generating mechanistic hypotheses about this relationship (see[5,56–58]) and in understanding the morphological basis of past and present functional adaptation. At the same time, it will underscore myriad remaining uncertainties about how joints work. Each additional correlation we identify between articulation score and in vivo kinematics generates new questions about why such relationships exist and what developmental and evolutionary processes have contributed to their existence. Answering these questions and others that arise from this line of research will bring us closer to elucidating the fundamental principles that underlie form-function relationships across the vertebrate tree, ultimately transforming our knowledge of vertebrate animal motion and its evolution.

## Methods

All procedures conducted with live animals were approved by the Institutional Animal Care and Use Committee at Brown University.

### Computed tomography scans, laser scans, and model creation

Computed tomography or micro-computed tomography scans of the tibiotarsus, tarsometatarsus, and third proximal pedal phalanx of three Helmeted Guineafowl (*Numida meleagris*) and one emu (*Dromaius novaehollandiae*), as well as structured light scans of the metatarsals and phalanges of one *Deinonychus antirrhopus*, were collected for analysis.

For the guineafowl ankle and metatarsophalangeal (MTP) analyses, micro-computed tomography scans previously collected by Manafzadeh et al.[59] (scanning parameters therein) of the right tarsometatarsus and right tibiotarsus of three guineafowl individuals, and the right third proximal pedal phalanx of one individual, were used. For the emu ankle joint analysis, the right tarsometatarsus and right tibiotarsus of one emu individual were scanned (GE Lightspeed Pro 16-detector unit; 0.625 mm slice thickness at 140 kV and 200 mA; "boneplus" reconstruction algorithm), and for the emu MTP joint the left tarsometatarsus and left third proximal pedal phalanx of another individual were scanned (Animage Fidex CT scanner; 0.294 mm voxel size at 110 kV and 0.1 mA). Because artifacts caused by metal inclusions in the fossils prevented the acquisition of clear micro-computed tomography scans of *Deinonychus antirrhopus*, all left pedal (metatarsal and phalangeal) elements of specimen YPM (Yale Peabody Museum) 5205 were instead structured light scanned using an Artec Space Spider scanner with 3D point accuracy of 0.05 mm.

A mesh model of each bone was created in Amira v. 6.0.1 (guineafowl ankle/MTP and emu ankle; Mercury Systems, MA, USA), OsiriX v.4.1.2 (emu MTP; Geneva, Switzerland[60]), or Artec Studio 14 Software (*Deinonychus*) and cleaned and smoothed in Geomagic Wrap 2017 (3D Systems, Morrisville, NC, USA). Polygonal faces representing the medial and lateral articular surfaces of the distal tibiotarsus, proximal tarsometatarsus, distal tarso/metatarsal third condyle, and proximal third pedal phalanx for each extant taxon, and for the medial and lateral articular surfaces of the proximal and distal metatarsals and phalanges and proximal unguals for *Deinonychus*, were identified and isolated based on regional curvature changes using the Extract Curvature tool in Geomagic Wrap.

### Pose space sampling and articulation score calculation

Bone mesh models were assembled into forward kinematic rigs (i.e., digital marionettes[61]) for ankle, metatarsophalangeal (MTP), and interphalangeal (IP) joints in Maya 2023 animation software (Autodesk, San Rafael, CA, USA) using the archosaur joint coordinate system conventions and workflow outlined by[62]. Although Gatesy et al.[62] did not explicitly outline conventions for IP joint coordinate systems, these systems were constructed following the same philosophy as MTP joints using planes fit to the proximal articular surfaces and cylinders fit to the distal articular surfaces of each phalanx. In a break from the "joint-inspired, segment-based" philosophy of Gatesy et al., ungual anatomical coordinate systems were strictly joint-inspired and created by fitting a cylinder to the proximal articular surfaces of each element to orient the Z-axis and using the proximal extent of the articular surface (rather than information about the position of the ungual tip) to orient the X- and Y-axes. At the ankle joint, a flexion-extension measurement of 180° corresponds to a fully extended joint, with flexion decreasing the flexion-extension angle. By contrast, at the MTP and IP joints, a flexion-extension measurement of 0° corresponds to a fully extended joint, with positive values corresponding to dorsiflexion and negative values corresponding to plantarflexion. At the ankle, abduction is positive whereas adduction is negative, and external rotation is positive whereas internal rotation is negative. At the MTP and IP joints, adduction is positive whereas abduction is negative, and external rotation is positive whereas internal rotation is negative.

Each joint was rotated through full potential Euler rotational pose space at five-degree resolution (360°/5 × 180°/5 × 360°/5 = 186,624 unique poses) using the *rotateSRJ.mel* Maya Embedded Language

script[25]. At each rotational pose, 343 potential translation combinations were allowed (7 × 7 × 7 translations sampled using the prism-based hinge joint translation method proposed by[26] see Supplementary Fig. 10), yielding a total of 64,012,032 unique joint configurations per joint. Translation ranges were selected to be intentionally conservative (i.e., to minimize risk of inadvertently excluding rotational poses used in life) based on the radius and height of a cylinder fit to condyles of the distal tibiotarsus (ankle), distal third tarso/metatarsus (MTP), or distal phalanx (IP) using Geomagic Wrap 2017 (3D Systems, Morrisville, NC, USA) and are reported as Supplementary Data 1; see also Supplementary Fig. 10. At each six-degree-of-freedom configuration, interpenetration was detected based on the presence of at least one positive dot product between (1) the vector connecting a point on one mesh and its closest point on the other and (2) the surface normal of that closest point.

We then conducted an articular raycast at each viable (i.e., non-interpenetrating) configuration. The articular raycasting approach we propose here aims to capture information about the morphological relationship of a pair of mating articular surfaces in any viable (i.e., non-interpenetrating), static six-degree-of-freedom joint configuration. Conceptually, our approach provides data about the relationship between these surfaces' 3-D curvatures. We select this emphasis here because interactions between articular surface curvatures are fundamental to joint function (see[5,36]). The relationship between the curvatures of mating articular surfaces in any given configuration dictates the paths of minimum work along which joints habitually move, as well as the capacity of a joint to effectively distribute and evenly transmit loads.

Our approach relies on calculation of the "vertex normals" of both articular surfaces to capture information about their local curvatures, following numerous previous studies of joint function (e.g.[52–54]). A 3-D polygonal mesh of an articular surface is composed of vertices which are joined to form triangular faces. Any single face has a 3-D orientation in space, and a vector drawn orthogonally (i.e., perpendicularly) to that face based on its orientation is called a "face normal" (Supplementary Fig. 11a; readers may have pre-existing familiarity with this concept given the mathematical normal vector to a plane). Averaging the face normals of all faces surrounding a vertex yields a "vertex normal" for each vertex of the mesh (Supplementary Fig. 11b).

To conduct an articular raycast in this study (we note that readers already familiar with "raycasting" as it relates to graphical rendering should take care not to confuse these applications), we sent out (i.e., "cast") rays – vectors of infinite length – in the direction of all of the vertex normals of one articular surface (Supplementary Fig. 11c). We then checked how many of those rays were oriented such that they hit the mating articular surface, forming vectors of a specific length connecting both surfaces, and how many rays instead shot past the mating articular surface, missing it entirely (Supplementary Fig. 11d). We propose that the formation of a successful ray hit means that articular curvatures are aligned such that there is a capacity for meaningful biomechanical interaction between the pair of articular surfaces in vivo. Therefore, if even a single ray hit successfully, we assigned the joint configuration an articulation score of greater than zero (see formula, below). If no rays succeeded in hitting the mating articular surface, we considered the joint to be unscorable, and gave the joint configuration an articulation score of zero. Articulation score was then calculated using three parameters – overlap, symmetry, and congruence – each of which receives an individual subscore from 0 to 1.

Articular overlap was calculated as the percentage of cast rays that successfully hit mating articular surfaces (Supplementary Fig. 12). If 0% of cast rays hit successfully, overlap received a subscore of 0; if 100% of cast rays hit successfully, overlap received a subscore of 1. For the hinge joints considered here, overlap was calculated separately for the medial and lateral pairs of mating articular surfaces, resulting in separate medial overlap and lateral overlap subscores.

Articular symmetry was calculated as the ratio of average lengths of successful rays between the medial and lateral condyles (Supplementary Fig. 13). For each half of the joint (i.e., medial or lateral), the length of each successful ray was measured, and the average length per side was calculated. Then, the smaller resulting average was divided by the larger resulting average. If average ray lengths between sides could hypothetically be infinitely different, this would have received a subscore of 0; if average ray lengths were exactly identical, this would have received a subscore of 1. (N.B., we suggest that in future work on additional joint types, rather than scoring symmetry on the basis of a ratio of average ray length between the two halves of a bicondylar joint, it can be evaluated using the statistical spread of the distribution of ray lengths for a single pair of articular surfaces.)

Articular congruence was calculated as the average difference in angle between the direction of each successful ray and the normal vector of the location it hit on the mating bone (Supplementary Fig. 14). For each half of the joint (i.e., medial or lateral), the 3-D angle between each successful ray and the normal vector of its hit location was calculated, and then the average angle per side was calculated. This average was then subtracted from 90, and the result was divided by 90, yielding a subscore between 0 and 1, where higher scores indicate a better curvature match between articular surfaces. If successful rays could hit on average completely tangentially to the mating surface (average angle difference of 90), this would have received a subscore of 0, if successful rays could hit on average in perfect alignment with the curvature of the mating surface (average angle difference of 0), this would have received a subscore of 1. For the hinge joints considered here, congruence was calculated separately for the medial and lateral pairs of mating articular surfaces, resulting in separate medial congruence and lateral congruence subscores.

Articulation score was calculated from these subscores using the following formula: Medial Overlap + Lateral Overlap + (Average Overlap) × (Symmetry + Medial Congruence + Lateral Congruence) (Supplementary Fig. 15). This formula was selected to weight the sum of all non-overlap parameters by full-joint overlap. Because each subscore has a potential range of 0 to 1, raw articulation score could range from 0 to 5. A joint-specific final articulation score was then calculated by dividing all sampled raw articulation scores for a joint by the maximum obtained articulation score, and multiplying the resulting values by 100, resulting in scores ranging from 0 to 100. Scores for all six-degree-of-freedom configurations were calculated in this way, and the configuration yielding the maximum articulation score per rotational pose was plotted in rotational pose space. Analysis time was roughly 5–12 h per joint (depending primarily on number of polygonal mesh faces in articular regions) on a 2021 Macbook Pro with an Apple M1 Max processor and 64 GB of RAM, requiring no specialized computing equipment.

After initial articulation analysis was completed for all of rotational pose space, targeted reanalysis of high-scoring poses was then conducted. The bounding box surrounding the subset of rotational poses yielding scores between 85–100 of the overall maximum score at five-degree angular resolution was resampled at more detailed one-degree resolution, again with the same 343 translation combinations allowed at each rotational pose, and articulation scores were recalculated based on the new maximum articulation score. Translational sensitivity analysis was performed for guineafowl individual 1 by allowing 1331 translational combinations over a larger range (two additional increments of equal size at each end of the X, Y, and Z translation ranges) at five-degree angular resolution, and sensitivity to articulation score formula was evaluated by conducting an analysis for this individual at the original rotational and translational resolution, weighting congruence by single-condyle overlap rather than full-joint overlap.

Non-interpenetrating joint poses were plotted as opaque squares colored by viability (i.e., lack of interpenetration) or semi-transparent

squares colored by articulation score in Maya 2023. Pose data were not cosine-corrected[63] for ease of interpretation and to allow square size to represent gridded angular sampling resolution; this choice was deemed acceptable because this study did not aim to quantitatively compare rotational mobility among joints and the highest scoring regions were in the least distorted portion of rotational pose space (low magnitude values in abduction-adduction).

### Comparative pose data collection

Guineafowl ankle pose comparison data ($n = 11,510$) were collected from guineafowl walking and running on a treadmill using marker-based XROMM[64] and were originally reported by Kambic et al.[65,66] and Manafzadeh et al.[59] (detailed methods including specifics of X-ray technique provided therein). Guineafowl MTP angles ($n = 439$; walking from one individual) were calculated using a combination of marker-based XROMM and scientific rotoscoping[61] following[67]. Additional flexion-extension angles for the guineafowl MTP ($n = 355$; 220 walking and 135 running from one individual), as well as comparison data for the emu ankle ($n = 22$; 12 walking from six individuals and ten running from five individuals) and emu MTP ($n = 28$; 18 walking from 12 individuals and 10 running from five individuals) were measured in 2-D by registering rigged skeletal or outline models to frames of standard light video (guineafowl; see[65,66]) and photographs available on the internet (emu) in Maya 2023.

### *Deinonychus* walking stride cycle analysis

An inverse kinematic animation rig was used to animate a rough walking stride cycle for the pes (i.e., the MTP and IP joints of the second, third, and fourth digits) of *Deinonychus* using inspiration from published extant avian toe tip motion by importing published videos and reconstructed animations of walking guineafowl toe tip motion into Maya and aligning the metatarsus and distal phalanges of *Deinonychus* to the guineafowl metatarsus and distal phalanges[67,68]. The joint rotations from this rig were then transferred to a forward kinematic animation rig[61] using the prism-based hinge joint translation rigging approach developed by[26]. Rotations and translations were then manually refined from their initial inverse kinematic calculation using Maya manipulators to ensure that each joint met or exceeded an articulation score of 95 (based on our one-degree angular resolution analysis) throughout the entire stride cycle. We emphasize that a score threshold of 95 was selected for use here at these bicondylar pedal joints because of the correspondence found using in vivo data for bicondylar ankle and metatarsophalangeal joints in extant dinosaurs – our goal was to reconstruct a walking stride, and all walking poses (and the vast majority of running poses) fell within the region of pose space with articulation scores 95–100 for the guineafowl and emu ankle and metatarsophalangeal joints studied (see Results). We therefore reiterate that application of this approach to non-dinosaurian joints, to joints from other parts of the dinosaurian body, or in reconstructing other behaviors will require confirmation of a similarly consistent relationship and determination of new, appropriate score thresholds through the comparison of articulation score distributions with in vivo pose space occupation.

### Reporting summary

Further information on research design is available in the Nature Portfolio Reporting Summary linked to this article.

## Data availability

Guineafowl CT scans and associated in vivo XROMM data have been previously published by Manafzadeh et al. (2021) and are available on the XMAPortal at https://xmaportal.org/webportal/larequest.php?request=CollectionView&StudyID=20&instit=BROWN&collectionID=15; emu CT scans were provided by John R. Hutchinson and are available on request directly from him. 3D meshes used to demonstrate the potential for broader application of articulation analysis in the Supplementary Information are available in the National Institutes of Health 3D Portal at https://3d.nih.gov/entries/3DPX-000387 and the Harvard Dataverse https://doi.org/10.7910/DVN/XP3JVZ. 3D meshes of the *Deinonychus antirrhopus* pedal elements studied here are available through *MorphoSource* at https://www.morphosource.org/projects/000592906?locale=en or on request from the Yale Peabody Museum of Natural History.

## Code availability

The *rotateSRJ.mel* Maya Embedded Language Script is fully described by[25] and is available at https://bitbucket.org/xromm/xromm_other_mel_scripts/src/main/joint_mobility/. All other analyses were conducted using manual selection of native Autodesk Maya commands.

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

## Acknowledgements

We thank J. Hermanson and K. Roorda for providing a cadaveric guineafowl specimen (guineafowl individual 3); D. Mares, Songline Emu Farm, H. O'Neill, D. Sustaita for providing and CT scanning the emu MTP joint specimen; J. Hutchinson for CT scanning the emu ankle joint specimen; P. L. Falkingham, R. E. Kambic, and T. J. Roberts for their involvement in in vivo guineafowl data collection; J. A. Gauthier, M. Fox, D. Brinkman and V. Rhue for access to *Deinonychus* specimen YPM 5205; C. Morét for scanning the *Deinonychus* specimen; E. Herbst for providing human glenohumeral data for use in responses to reviewers; and E. L. Brainerd, J. A. Gauthier, and T. J. Roberts for thoughtful criticism. This work was supported by the Bushnell Research and Education Fund (A.R.M.), US NSF (Grants IOS-0925077 [S.M.G.], DBI-0552051 [S.M.G.], IOS-0840950 [S.M.G.], DBI-1262156 [S.M.G.], EAR-1452119 [S.M.G.], GRFP [A.R.M.], PRFB [DBI-2209144; A.R.M.]), Sigma Xi Grant-in-Aid of Research (A.R.M.), Society of Vertebrate Paleontology Cohen Award for Student Research (A.R.M.), Association for Women Geoscientists/Paleontological Society Winifred Goldring Award (A.R.M.), Brown University Presidential Fellowship (A.R.M.), and Yale Institute for Biospheric Studies Gaylord Donnelley Postdoctoral Environmental Fellowship (A.R.M.).

## Author contributions

A.R.M. conceived the project; A.R.M. and S.M.G. designed the project; A.R.M. conducted analyses; A.R.M., S.M.G., and B.-A.S.B. discussed the results and contributed to preparation and revision of the manuscript.

## Competing interests

The authors declare no competing interests.
