## [Peer Review File · Nature Communications]

Articular surface interactions distinguish dinosaurian locomotor joint posesEditorial Note: Figure on page 121 of this file has been redacted as indicated to remove third-party material where no permission to publish could be obtained.

Reviewers' Comments:

Reviewer #1:

Remarks to the Author:

Manafzadeh et al. describe a new biomechanical analytical method for limb joints which determines range of motion and digit position during locomotion. They digitally captured bone joint surface morphology data for two walking extant birds and use this to constrain a stride cycle for the extinct dromaeosaurid theropod dinosaur *Deinonychus*, a close relative of the ancestor of birds.

As someone who investigated *Deinonychus*' range of motion using just "the fit" of foot bones, I am familiar with the fossil material and that of extinct paravian relatives. I am not practiced in computational biomechanics, so my comments on that aspect are limited. I saw the platform presentation for this paper at SVP2022, so was already familiar with the research.

The paper presents a creative new method and in my opinion it should be 'published with minor edits'. It's a great piece of work, so my few comments really only concern some language choices and the morphology of *Deinonychus*.

WRITING

The writing is a bit dense and academic, which is understandable as this is a scientific publication, and it would represent a masterstroke of editing and dense writing if it were submitted to a specialized journal. However, for a more general journal it doesn't make it more readable, and I did not find it easy to follow sometimes. I think the readership for this paper is thus slightly narrowed by the writing style. This is, however, no major complaint and not a barrier to publication.

QUALITATIVE ROM STUDIES & DEINONYCHUS.

I caution the tone of the introduction in challenging previous work on range of motion. The proposed new methodology is an advance, but I do not think that previous range of motion studies are "called into question", have their "validity challenged", or are subject to a "severe obstacle" (lines 50-53). In using such language, others may be dismissive of qualitative range of motion, which can and has made legitimate observations and help formulate hypotheses.

For example (and these observations are especially valid for the current study, see later), Our paper (Fowler et al, 2011) describes notable differences in foot morphology between derived dromaeosauridae (specifically including *Deinonychus*), basal dromaeosauridae, and troodontidae (and other theropoda). Basal dromaeosauridae have an elongate subarctometatarsalian metatarsus (probably basal for Deinonychosauria: troodontidae + dromaeosauridae), which is lost in derived dromaeosauridae (which have a short broad non arctometatarsalian metatarsus), but is further developed into the full arctometatarsalian condition in troodontids. We suggest that this is due to increase in cursorial adaptation in troodontids, with derived dromaeosaurids conversely adapting towards greater overall foot strength. Dromaeosaurids have a more flexible ball-joint on MT-IV which allows more variable lateral positioning of D-IV (I note with interest that this is not mentioned in the current manuscript). Troodontids by comparison have a MT-I with a ball joint (as do Caenagnathid oviraptorosaurians), whereas in dromaeosaurids, D-I has more restricted range of movement. Dromaeosaurids have greater hypertrophy of the claw on D-II, whereas it is not unusually enlarged in troodontids and basal avialans. Derived dromaeosaurids like *Deinonychus* have strongly ginglymoid articulations on D-II, moderate ginglymoidy on D-III and D-IV. Troodontids, by comparison, have ginglymoid articulations on all phalanges of D-II, but on D-III and D-IV the interphalangeal joints are only ginglymoid on the terminal nonungual phalanx where it articulates with the unguis (presumably strengthening the unguis joint).

In my experience, hypotheses can originate with such qualitative observations, not by starting with massive datasets of random measurements and hoping a pattern emerges out of the cloud (which is not to say that some people may meet success with such an approach).

I apologise for laboring the point, but it is important.

The above observations on Deinonychosaurian pedal morphology may contradict Line 136-141 in the current manuscript, which states:

"Although it is large for a paravian [34], Deinonychus is otherwise conservative in its hindlimb anatomy: it resembles other deinonychosaurs on the one hand (including primitive, flighted taxa such as Anchiornis) and 7 basal avialans, notably Archaeopteryx, on the other [35]. It is not unlikely that all of these taxa had stride cycles broadly similar to that of Deinonychus, including the habitually hyperextended posture of pedal digit II."

I don't agree with this. Deinonychus is not really conservative and instead represents a close-to-endmember morphology for derived dromaeosaurids. Most notably, the metatarsals are not subarctometatarsalian (as in basal dromaeosaurs) and are actually rather short, especially compared to basal dromaeosaurids, troodontids, and even basal birds like Archaeopteryx. The D-II ungual is unusually large for Deinonychosauria as a whole, which may affect the way it is carried. D-IV is unusually long (being subequal to D-III; in cursors D-III is usually much longer). Various other proportions of the phalanges, orientations of the toes, and ginglymoidy/joint shapes are representative of a derived dromaeosaurid (see above paragraph).

Deinonychus and other derived dromaeosaurids have feet adapted towards grasping and strength rather than being cursorial. Whether this affects their stride mechanics in a way that would impact the current study, I do not know.

It has been assumed that the D-II claw is held in extension above the ground, but it's actually not all that well founded, and certainly wasn't very well founded in 1969 when suggested by Ostrom. The sheer size of the ungual (and the additional length of the sheath) would mean that the non-ungual phalanges would require some extension just so that the claw tip did not embed into the ground (which I was cheered to see noted in the manuscript). However, footprint data of extant *Seriema* (for example) have a small point where the tip of the claw touches the ground. I don't think people have looked for this in supposed dromaeosaurid didactyl footprints. This is part of a future mini project for me, revisiting assumptions of how the D-II claw is carried (with alternative explanation), but I thought I would bring it up here since it may affect the authors' work.

Also, note that Ostrom had the orientation of D-III and (to a lesser extent) D-IV incorrect in the classic figure of the Deinonychus foot in the 1969 monograph, and it looks like this reconstruction is being followed in the supp info animations. Ostrom shows D-III as being more or less straight, following the straightness of the main shaft of MT-III. However, (in the fossils) the distal end of MT-III is deflected toward MT-II such that D-III actually deflects towards D-II (which is primitive for theropoda) rather than being straight forward (e.g. as seen in Emu; Ornithomimids, and to my knowledge, most ground birds). Compare the reconstruction of Ostrom 1969 with Fowler et al 2011 (I'll see if I can attach some images to this review). Curiously, our reconstruction makes more sense regarding the subtle lateral curvature of the D-III ungual tip wherein the tips of D-IV and D-III now point more or less anteriorly, rather than being inward pointing in Ostrom 1969.

So I suppose I am curious whether the supp info animations follow Ostrom's 1969 reconstruction of the Deinonychus foot. Mainly, I would be surprised because in a real fossil, PD-III-1 will not articulate with MT-III in a straight line.

(also note that the ball-shaped distal condyle of Deinonychus' MT-IV means that D-IV can vary in position somewhat, although this is less critical).

SUMMARY

I've talked a lot about Deinonychus & other paravian ecology. This is to draw attention to some differences I have with how Deinonychus' foot morphology and function are interpreted in the manuscript. Mainly however, I really like the manuscript and method.

What I would really like is if this new methodology was applied to different theropod taxa to see what difference the qualitative observations outlined above make to range of motion in these taxa. Fowler et al 2011 suggest that Troodontids and Dromaeosaurids basically evolve in opposite directions (cursor, grasper), so it would be really great to see this investigated. Of course, I'm not saying that this needs to be done in a revision of this manuscript, I'm just being enthusiastic about the potential for this method once this paper is published.

Deinonychus (Ostrom, 1969)

Deinonychus (Fowler et al, 2011)

Talos (troodontid)

Reviewer #2:

Remarks to the Author:

Thank you for inviting me to review Articular surface interactions distinguish dinosaurian locomotor poses by Manafzadeh and colleagues. I am aware of the work the Manafzadeh has been conducting on joint mobility and enjoyed reading this most recent work. It was well written, and I enjoyed learning about this new methodology. I have split my review into some more general comments at the beginning, followed by more specific comments later on.

General comments

Central message: I was a bit confused about what the central point of the paper was. From the title, I imagined I would be learning about new reconstructions for dinosaurian poses, but it unclear to me what poses were being distinguished. The method proposed here cannot, for example, identify the different poses involved in guineafowl walking or running, but instead provides an envelope of possible poses that may occur. The method also does not communicate between joints, so there is no way of creating a "locomotor pose", but rather can just create an array of possible locomotor poses. From the abstract, and this line in particular ("Without rigorous criteria for evaluating joint articulation, however, such analyses risk misleading reconstructions of in vivo motion"), I imagined the paper would be about rigorous methods for evaluating joint articulation, and how incorrect methods lead to misleading reconstructions of motion. But the paper neither presented a method for evaluating joint articulation or compared this method to others in terms of reconstructions (showing that this method is more accurate). The reason I say the paper did not present a method for evaluating joint articulation is that it provides a method for quantifying the likelihood of a joint engaging in a pose based on a measure of joint stability, but this does not quantify articulation. Nowhere in this metric can you determine if a joint is articulated or not, just if a joint is likely to engage in a pose during walking/running in a straight line.

What I believe the paper is about is a method for quantifying joint ROM based on a measure of stability, where relatively higher scores within a joint represent increased likelihood of the joint engaging in that pose. As such, it is a methodological paper, where the methodology has been applied to an extinct taxon to show how it can be used to reconstruct motion. Unfortunately, I do not believe anything new was learned about this extinct taxon by using this method, especially as the "whole foot" poses that reconstructed are random and largely informed by the author's opinions, and not this method, since this method does not reveal any information about covariation in joint positions across the foot.

The metric: I like the metric and think it could be useful but have some concerns about its generalizability to other joints in the vertebrate skeleton. For example, I imagine, when applied to vertebrate, you would find that the spine can spin 360° about its long axis! I also wonder what would happen with sliding joints, and whether you would find them being very "unstable" unless they are more-or-less completely overlapping. I do not think this is a general method for all articular joints, and may be limited in usefulness, e.g., to hinge joints.

The metric also seems to be limited by the lack of communication between body segments. This is less important for some joints and more important for others. For example, where there are many joints within the same region (e.g., the MTP or PP joints), it is possible that the position of one segment would constrain the ROM of another. Even in areas where joints are far away from each other, segments can constrain ROM (e.g., I can touch my elbows together, meaning the position of my left humerus can limit the ROM of my right shoulder).

This all means that this metric cannot be used to reconstruct locomotor poses, but provide an envelope for range of possible poses of a joint.

Ray casting: the use of ray casting is interesting, but I do not think entirely appropriate. By having a normal vector point from one articular surface to the other, it is suggesting that, if the joint were to move in the direction of the vector, the two articular surfaces would interact with each other. But what is the change of the joint actually moving in that direction? In many cases, extremely low. And even if it was high, the vectors on a curved surface point in a myriad of directions, only one of which can correlate to the motion of the joint. The underlying logic and biomechanical meaning of the ray casting is not clear to me, and seems more-or-less arbitrary.

One of the benefits of this method, as stated by the study, is that the ray casting takes the bony curvature of the joint into account, making it a better method for estimating joint mobility. The curvature of the joint and its relation to motion and mobility only matters in some, but not all joints. Take the human shoulder joint, for example. The humerus articulates with the glenoid cavity on the scapula, which has some curvature. The curvature of the cavity can be quite flat, leading to many positions of the humeral head which would be subluxations as being completely stable according to this method. I would similarly be surprised if the range of likely joint positions would be accurate for the (human) hip joint, as I imagine most scores would be near 5 (or 100%) regardless of if the joint position is realistic or not, given the surrounding soft tissue.

95-100% window: I discuss below how this could be improved upon, and would stress that this needs to be done, as the range of the window (here, 5%) will likely change depending on which joint is being analyzed. Again, take the human shoulder joint for example. The humerus can occupy a wide range of positions that are equally "stable" and therefore the appropriate window may be 60-100%, and not 95-100%. Determining the actual window for each joint is important, instead of using an arbitrary 95-100%. From Fig 2, it appears wider windows may be needed for some joints, and narrower windows may be needed for others (e.g., considering the ankle and metatarsophalangeal joints in Fig 2f).

I'm sorry my review is not more positive at this time. I very much enjoyed reading the manuscript, but have too many concerns to recommend for publication at this time.

Specific comments

Abstract

L18: "... reconstruction of in vivo motion." Is an oxymoron, as reconstruction implies motion is unknown, but in vivo would imply the motion was measured when the animal was alive. The abstract is additionally vague up until this point, as the authors have not specified they are only considering vertebrates. I would change to "... reconstructions of vertebrate motion."

Introduction

L32: This first sentence is not correct. Vertebrate animal motion does not rely predominantly on the mobility of the joint, but rather the neuromuscular and musculoskeletal system (muscle origins/insertions, activation patterns and co-contractions, etc.). Joint mobility affects the range of motion, and how many times that motion can be performed. E.g., hypermobile people have high joint mobility, but subluxed / disarticulated joints only cause minor changes in joint mobility compared to, e.g., removing all the flexor muscles that cross a joint.

Results

L78: It appears from Fig 2d and Fig 2e that this is only sometimes true. I cannot tell if in Fig 2e, abduction-adduction, how much this is true given the density of the black points. A statistic would be nice here (e.g., "92.3% of the locomotor poses fall within the highest-scoring...") Alternatively, and probably more usefully, what range would be needed to ensure all locomotor poses fall within the range? The 95-100% range seems arbitrary, and it would be more useful if information like "a 89.5-100% range is needed to reliably encompass all locomotor poses during walking and running..." This is somewhat done in L82, but a more exact range would be useful, because that could then be applied to other animals.

L88-90: This sentence is confusing, as it sounds like you can predict differences between guinea fowl and emu joints. I do not think that is what the authors meant, and if it is, I am unsure why they are looking at this (given the rest of the paper) or how they made this comparison.

L101: if better envelopes than 95-100 were constructed, using the suggestion of the above comment, this could lead to large changes in this reconstruction, as it opens up the range of possible joint positions.

L130: This method did not, unfortunately, "...enabled us to reconstruct a full potential six-degree-of-freedom stride cycle for a non-avian paravian dinosaur..." Multibody dynamic analysis enabled this reconstruction. What this method did was provide potentially more accurate input parameters (here,

constraints) on the model, which potentially lead to a more accurate reconstruction. But even so, this cannot be said until a reconstruction is done on an extant animal, and the accuracy of using traditional joint constraints and these joint constraints are compared to some ground-truth metric (which can be done with the data presented in this paper).

L156-62: The discussion in this paragraph is nice, but does highlight the limitations of the currently proposed method for very specific types of joints.

Methods

L257: what qualified as a high-scoring pose? 95-100%? It is not clear from the writing here

Reviewer #3:

Remarks to the Author:

The submitted manuscript describes a new approach to accurately determine the joint position and range-of-motion of limb bones groundtruthing the method using modern bird species and applying the findings to a fossil dinosaur. This is an interesting and well-written paper with lots of potential to elucidate locomotor the palaeobiology of extinct species (however this is only done in a limited capacity, presumably with more in-depth fossil work reserved for a follow-up paper).

I do have several questions/suggestions with regards to the methodology but these are all fairly minor.

Kind regards,
Stephan Lautenschlager

Specific points:

- I found the scoring quite difficult to understand beyond the initial three parameters (overlap, symmetry, congruence) and a more detailed explanation/example would be beneficial. Related to this, the scoring seems to evaluate facet overlap but not necessarily dislocation (e.g. a dislocated joint which is essentially translated so that the overlap and symmetry stay the same but distance between corresponding facets increases or decreases will still score highly). How is this taken into account?
- Considering the huge number of scenarios tested there is no account of the computing power necessary to run the analyses, used machine specs, time taken for a full run, etc. while raycasting itself is manageable on most standard computers, the visualisation, especially of high-res models, may push a graphic card to its limits.
- Data availability: I don't think "Availability on request" statements are appropriate anymore and the respective data should be deposited in a public repository

Line 57: I doubt many readers will be familiar with raycasting and a brief description how this works would be useful.

Line 233: I understand that a full 360 degree rotation in a hypothetical possibility but practically impossible. Related to the main comment above, would it make sense to constrain rotation in the first place to minimise computational effort?

Line 236: If I understand this correctly, translation is tested for seven discrete steps in each direction.

How/which magnitude was selected here?

Line 242: How were collisions/interpenetrations detected. Is this automatically possible in Maya or was this coded into the script (e.g. negative distances of the ray casts)?

Manafzadeh et al. (2023) *Nature Communications* Responses to Reviewers

As you will see from the reports copied below, the reviewers raise important concerns. We find that these concerns limit the strength of the study, and therefore we ask you to address them with additional work. Without substantial revisions, we will be unlikely to send the paper back to review. In particular, we would encourage you to incorporate Reviewer #1's suggestion to include additional theropod taxa, if possible.

We respond point-by-point to each Reviewer's comments and requests below, and indicate any actions we have taken in each case. All Line numbers listed refer to positions in the revised manuscript.

REVIEWER COMMENTS

Reviewer #1 (Remarks to the Author):

Manafzadeh et al. describe a new biomechanical analytical method for limb joints which determines range of motion and digit position during locomotion. They digitally captured bone joint surface morphology data for two walking extant birds and use this to constrain a stride cycle for the extinct dromaeosaurid theropod dinosaur *Deinonychus*, a close relative of the ancestor of birds.

As someone who investigated *Deinonychus*' range of motion using just "the fit" of foot bones, I am familiar with the fossil material and that of extinct paravian relatives. I am not practiced in computational biomechanics, so my comments on that aspect are limited. I saw the platform presentation for this paper at SVP2022, so was already familiar with the research.

The paper presents a creative new method and in my opinion it should be 'published with minor edits'. It's a great piece of work, so my few comments really only concern some language choices and the morphology of *Deinonychus*.

We thank Reviewer 1 for their thoughtful criticism and positive impression of our work.

WRITING

The writing is a bit dense and academic, which is understandable as this is a scientific publication, and it would represent a masterstroke of editing and dense writing if it were submitted to a specialized journal. However, for a more general journal it doesn't make it more readable, and I did not find it easy to follow sometimes. I think the readership for this paper is thus slightly narrowed by the writing style. This is, however, no major complaint and not a barrier to publication.

Thank you for this feedback. It is clear from all reviewers' comments that the concision alluded to here unfortunately came at the cost of obscuring our methodology. Our goal is of course to make our writing clear to the broadest possible audience, and our initial submission fell short in this regard. We have therefore unpacked our field-specific language across the entire manuscript, and added an additional, more accessible Methods subsection with corresponding Supplementary Figures to better illustrate our methodology, with the goal of making our study more broadly accessible.

ACTIONS TAKEN:

- *Unpacked field specific language throughout paper (e.g., Lines 40-41, 51, 98-100)*

- *Added a large new methods subsection to better explain our technical approach (Lines 223-347)*
- *Added Supplementary Figures 11-15 to better display our technical approach*

QUALITATIVE ROM STUDIES & DEINONYCHUS.

I caution the tone of the introduction in challenging previous work on range of motion. The proposed new methodology is an advance, but I do not think that previous range of motion studies are “called into question”, have their “validity challenged”, or are subject to a “severe obstacle” (lines 50-53). In using such language, others may be dismissive of qualitative range of motion, which can and has made legitimate observations and help formulate hypotheses.

We completely agree that qualitative ROM studies can serve as a valuable first step in formulating broad-scale hypotheses about organismal motion – in fact, we would go so far as to say that until the current paper, such studies even had some long unappreciated advantages over more quantitative work (we will return to this point later in this response). That said, we note that this Reviewer’s self-stated lack of familiarity with computational biomechanics may include a lack of familiarity with more recent progress in paleontological mobility reconstruction. Here we offer some important context for the present study and its position within this subfield. (N.B., this context is more fully detailed across the papers cited in our Introduction, but we summarize it briefly and explicitly here.)

As reviewed by Manafzadeh & Gatesy (2022), over the past decade, paleontological biomechanics has moved largely away from the qualitative physical manipulation of previous studies. This shift to computational, quantitative analyses of joint ROM occurred because biomechanical specialists within the field discovered that for effective application to and unification with computational reconstructions of organismal motion (e.g., including muscle simulation), such analyses required a more precise and reproducible foundation. We suggest, however, that this computational methodology was rapidly and enthusiastically adopted without careful consideration of the costs that may be associated with shifting manipulations of joints into a virtual world.

The earliest virtual ROM reconstructions (e.g., Mallison, 2010; Pierce et al., 2012) involved rotating fossil joints about a single rotational axis (i.e., flexion-extension, abduction-adduction, long-axis rotation) at a time and assessing the excursions possible about each. These results were presented as arcs drawn on figures of joints, or as a series of bar graphs.

If we consider a 3-D joint mobility space in which each dimension of the space is one of three rotational degrees of freedom, such studies would imply that joints can occupy a rectangular prism of joint pose space (see figure below). However, in 2017, Robert Kambic and colleagues published work on the joints of living birds (2017a,b) revealing that rotational degrees of freedom interact (in agreement with previous work on humans; Haering et al., 2014). In other words, extremes of rotation in all three degrees of freedom cannot actually be reached simultaneously by joints, and a joint's range of motion forms a complex polygonal shape in 3-D joint mobility space. As a result, the single-axis rotation standard at the time for virtual ROM reconstruction might have resulted in under- or overestimates of joint mobility, depending on the starting pose selected. (The example in the figure below represents an example of an overestimate.)

heights of bar graphs per degree of freedom are dimensions of implied 3-D mobility prism

[Manafzadeh et al., Responses to Reviewers 2023]

As a result, Manafzadeh and Padian (2018) recognized the need for a change to computational methodology and developed an approach for automatically and systematically sampling all possible joint rotational poses, or simultaneous triples of excursions in all three rotational degrees of freedom: (flexion-extension°, abduction-adduction°, long-axis rotation°). Each joint rotational pose at a given sampling resolution is represented by the blue points in 3-D joint mobility space below.

[Manafzadeh et al., Responses to Reviewers 2023]

Each pose could then be automatically checked by the computer to see whether it resulted in bone-on-bone interpenetration (i.e., bone meshes inside of each other), suggesting that the pose was not biologically possible. Although this work represented a methodological advance

over earlier studies and was widely adopted by the paleontological community (e.g., Demuth et al., 2020; Brocklehurst et al., 2022; Richards et al., 2021), it suffered from a critical flaw in its presentation. Manafzadeh and Padian had automated the sampling of joint rotations, but had manually imposed joint translations (see below figure) in their study. As a result, later workers adopting their methodology often did not allow joint translations, instead holding joints at a fixed rotational pivot in space (for example, at the hip joint, imagine holding the femoral head in a fixed location within the acetabulum and only allowing the femur to rotate, but not allowing the femoral head to move around -- or see the knee rotation examples to the left of the figure, below.)

[Manafzadeh et al., Responses to Reviewers 2023]

This secondary treatment of joint translation proved problematic when Manafzadeh and Gatesy (2021) discovered that without considering systematic combinations of both joint rotations AND translations, computational ROM analyses run a severe risk of excluding many of the joint rotational poses the animal could have used in life (based on cadaveric ROM studies of extant animals), even those used in key behaviors like locomotion (based on in vivo experimental studies of extant animals). This is why we state in the current paper that even the most methodologically advanced existing studies are “called into question,” have their “validity challenged,” etc. -- those studies adopting even the most recent computational approaches ran this risk of unjustifiably excluding perfectly reasonable joint poses, which presents a large problem for reconstructing organismal function.

The main reason our finding about joint translations revealed a real roadblock to paleontological analysis of organismal function is that even if we now knew we had to include joint translations in computational analyses, we had no good way of putting limits on them -- meaning that just

about any joint rotational pose would have to be considered possible. If you hold two bones infinitely far apart, you can spin them however you want! Obviously, infinitely far apart is “too far” -- and even in some more plausible configurations we can decide that to our eye, bones seem “too far” apart -- but how do we objectively decide what “too far” is across joints? And then how could we ever go about automating a detection of “too far” to allow a systematic and reproducible analysis of millions or billions of six-degree-of-freedom joint configurations?

This is when we realized that the qualitative ROM observations supported by this Reviewer did, in fact, have an advantage that was lost in the transition to the rigid computational world. Traditional paleontologists have a wonderful, intuitive sense for joint configurations “looking right” or “looking wrong,” and that intuition is what guides much of qualitative ROM interpretation – but it is lost in computational ROM reconstructions. That is what gave us the idea that if we could harness that traditional intuition -- if we could find a way to capture it, formalize it, measure it quantitatively, and ground-truth it by comparing it against real data from living animals -- then we could come up with a real measure of joints “looking right” (i.e., proper articulation). This analysis could be used to reel in joint translation, and in turn, joint rotation, allowing far more meaningful and targeted analyses of joint pose space occupation in extinct animals. Therein lies the goal of the current paper.

We hope that this summary offers some more context for this paper’s position within the broader literature. We wish we could include this level of background in every paper we write, but of course, space is limited. All of this said, we certainly do not mean to devalue this Reviewer’s work or that of others adopting similar methodology, and we likewise do not want nonspecialist readers to interpret our stance in that way. We have made some phrasing changes (below) to the Introduction and Discussion to clarify that we intend only to criticize (frankly our own, flawed) computational methodology, not traditional qualitative ROM estimates.

ACTIONS TAKEN:

- *Made clarifying phrasing changes in Lines 46, 53-54, 129-130*

For example (and these observations are especially valid for the current study, see later), Our paper (Fowler et al, 2011) describes notable differences in foot morphology between derived dromaeosauridae (specifically including Deinonychus), basal dromaeosauridae, and troodontidae (and other theropoda). Basal dromaeosauridae have an elongate subarctometatarsalian metatarsus (probably basal for Deinonychosauria: troodontidae + dromaeosauridae), which is lost in derived dromaeosauridae (which have a short broad non arctometatarsalian metatarsus), but is further developed into the full arctometatarsalian condition in troodontids. We suggest that this is due to increase in cursorial adaptation in troodontids, with derived dromaeosaurids conversely adapting towards greater overall foot strength. Dromaeosaurids have a more flexible ball-joint on MT-IV which allows more variable lateral positioning of D-IV (I note with interest that this is not mentioned in the current manuscript).

We of course carefully studied Fowler et al. (2011) in working on this project, and we found it to offer a treasure trove of interesting and relevant information -- we thank the Reviewer and their team for laying so much hypothesis-generating groundwork, setting the stage for studies like ours! We look forward to seeing future, computational biomechanical work on the joints of dromaeosaurids and troodontids and we are optimistic that the methodology we present here will be of use to that paper’s authors and their colleagues.

Regarding the specific point/concern about MT-IV: our analyses did not recover the “increased flexibility in lateral positioning” discussed by this Reviewer and in Fowler et al. (2011), which is why it is not mentioned in our manuscript. Given gross comparison of the dromaeosaurid MTP-IV joint with MTPs of extant birds and other reptiles (see Mitrovic, 1977; Tehrani et al., 2017; Schaeffer, 1941), and considering the general conservation of soft tissue structures (specifically general ligamentous architecture) in this region of the Bauplan across tetrapods, we suggest that the “ball-like” distal facet of metatarsal IV should not be interpreted as creating a true ball-and-socket joint. Rather, we suggest that as part of an (albeit derived) hinge joint, the fourth metatarsal must still maintain general symmetry between the medial and lateral halves of its articulation with the receiving facets on pedal phalanx IV-1.

In fact, we take this opportunity to point out that this disagreement inadvertently serves to highlight one advantage of our quantitative approach to mobility analysis over more qualitative interpretations -- if this Reviewer and/or future workers disagree with our results, they can identically reproduce our analyses and objectively point to very specific elements of our study design that they would have implemented differently. This level of precise and evidence-based argument about the interpretation of motion in extinct animals is exactly where we hope the approach we present here will help to shift paleontological discourse to going forward.

Troodontids by comparison have a MT-I with a ball joint (as do Caenagnathid oviraptorosaurians), whereas in dromaeosaurids, D-I has more restricted range of movement. Dromaeosaurids have greater hypertrophy of the claw on D-II, whereas it is not unusually enlarged in troodontids and basal avialans. Derived dromaeosaurids like *Deinonychus* have strongly ginglymoid articulations on D-II, moderate ginglymoidy on D-III and D-IV. Troodontids, by comparison, have ginglymoid articulations on all phalanges of D-II, but on D-III and D-IV the interphalangeal joints are only ginglymoid on the terminal nonungual phalanx where it articulates with the unguis (presumably strengthening the unguis joint).

We are in agreement about broad patterns of morphological variation among these animals. The exciting thing about the approach we present here is that it opens the door for functional implications of these differences to be investigated in greater detail.

In my experience, hypotheses can originate with such qualitative observations, not by starting with massive datasets of random measurements and hoping a pattern emerges out of the cloud (which is not to say that some people may meet success with such an approach).

I apologise for laboring the point, but it is important.

We completely agree that qualitative observations can lead to the generation of hypotheses, and as we discuss in our response above, we have clarified our writing to ensure that readers do not interpret our manuscript as suggesting that qualitative studies lack value.

However, we suggest that to effectively test such hypotheses – regardless of how they are generated – paleontologists require data that are of sufficient complexity to adequately reflect the true biomechanical complexity of in vivo organismal motion, and that is where we believe that qualitative inferences of potential functional differences can fall short. The measurements we take here to address these kinds of questions at the necessary level of precision are not random, but rather reflect known principles of articular function that underlie motion in animals extant and extinct -- meaning that ultimately, success (if we take it to be defined as agreement

with what is observed in extant animals) is not a happy accident, but rather an outcome that is both logical and probable.

ACTIONS TAKEN:

- *Revised Methods substantially to expand on and better explain the logic behind our measurements*

The above observations on Deinonychosaurian pedal morphology may contradict Line 136-141 in the current manuscript, which states:

“Although it is large for a paravian [34], Deinonychus is otherwise conservative in its hindlimb anatomy: it resembles other deinonychosaurs on the one hand (including primitive, flighted taxa such as Anchiornis) and 7 basal avialans, notably Archaeopteryx, on the other [35]. It is not unlikely that all of these taxa had stride cycles broadly similar to that of Deinonychus, including the habitually hyperextended posture of pedal digit II.”

I don't agree with this. Deinonychus is not really conservative and instead represents a close-to-endmember morphology for derived dromaeosaurids. Most notably, the metatarsals are not subarctometatarsalian (as in basal dromaeosaurs) and are actually rather short, especially compared to basal dromaeosaurids, troodontids, and even basal birds like Archaeopteryx. The D-II ungual is unusually large for Deinonychosauria as a whole, which may affect the way it is carried. D-IV is unusually long (being subequal to D-III; in cursors D-III is usually much longer). Various other proportions of the phalanges, orientations of the toes, and ginglymoids/joint shapes are representative of a derived dromaeosaurid (see above paragraph).

Deinonychus and other derived dromaeosaurids have feet adapted towards grasping and strength rather than being cursorial. Whether this affects their stride mechanics in a way that would impact the current study, I do not know.

We appreciate the reviewer's observations and agree that Deinonychus displays a number of apparent autapomorphies, to which we now make reference in the text. We have reworded our statement to indicate instead that Deinonychus is conservative in a number of ways while it is derived in others. Our initial statement was meant to be taken in a very general sense, more akin to that used in biomedical research than in detailed taxonomic study. In that general sense, Deinonychus is more conservative among paravians than, say, a hummingbird or a derived enantiornithine.

ACTIONS TAKEN:

- *Changed phrasing in Lines 69 and 147-154 to be more precise*

It has been assumed that the D-II claw is held in extension above the ground, but it's actually not all that well founded, and certainly wasn't very well founded in 1969 when suggested by Ostrom. The sheer size of the ungual (and the additional length of the sheath) would mean that the non-ungual phalanges would require some extension just so that the claw tip did not embed into the ground (which I was cheered to see noted in the manuscript). However, footprint data of extant *Seriema* (for example) have a small point where the tip of the claw touches the ground. I don't think people have looked for this in supposed dromaeosaurid didactyl footprints. This is part of a future mini project for me, revisiting assumptions of how the D-II claw is carried (with alternative explanation), but I thought I would bring it up here since it may affect the authors' work.

We certainly found that joint articulation data strongly support general hyperextension of digit II – we look forward to reading about the results of this interesting future work on footprints!

Also, note that Ostrom had the orientation of D-III and (to a lesser extent) D-IV incorrect in the classic figure of the Deinonychus foot in the 1969 monograph, and it looks like this reconstruction is being followed in the supp info animations. Ostrom shows D-III as being more or less straight, following the straightness of the main shaft of MT-III. However, (in the fossils) the distal end of MT-III is deflected toward MT-II such that D-III actually deflects towards D-II (which is primitive for theropoda) rather than being straight forward (e.g. as seen in Emu; Ornithomimids, and to my knowledge, most ground birds). Compare the reconstruction of Ostrom 1969 with Fowler et al 2011 (I'll see if I can attach some images to this review). Curiously, our reconstruction makes more sense regarding the subtle lateral curvature of the D-III ungual tip wherein the tips of D-IV and D-III now point more or less anteriorly, rather than being inward pointing in Ostrom 1969.

So I suppose I am curious whether the supp info animations follow Ostrom's 1969 reconstruction of the Deinonychus foot. Mainly, I would be surprised because in a real fossil, PD-III-1 will not articulate with MT-III in a straight line.

(also note that the ball-shaped distal condyle of Deinonychus' MT-IV means that D-IV can vary in position somewhat, although this is less critical).

We also picked up on this difference between Ostrom and Fowler et al.'s interpretations of Deinonychus MTIII while working on this study. Unfortunately, it appears the camera perspective on our supplementary animation was misleading, because we did actually find stronger support for an interpretation of digit III orientation more in line with that of Fowler et al. than that of Ostrom.

Of course, some variation in joint rotation is both possible and natural in true locomotion (see, for example, the 3-degree-of-freedom guineafowl ankle locomotor data we report in Figure 2, and our responses to Reviewer 2 below), but overall we did find that the joint morphology of MTP-III is suggestive of a general bias of deflection towards digit II as the Reviewer suggests. We again emphasize we think it is exciting that such conclusions can now be drawn and supported in an objective and reproducible manner!

(We already shared our view on the Reviewer's comments about MTP-IV in another response, above.)

ACTIONS TAKEN:

- *Created a new animation (Supplementary Movie 3) showing our reconstructed stride cycle from additional perspectives to resolve this ambiguity*

SUMMARY

I've talked a lot about Deinonychus & other paravian ecology. This is to draw attention to some differences I have with how Deinonychus' foot morphology and function are interpreted in the manuscript. Mainly however, I really like the manuscript and method.

Thank you again, and we appreciate the Reviewer taking the time to engage with our work so thoroughly and to write this all out.

What I would really like is if this new methodology was applied to different theropod taxa to see what difference the qualitative observations outlined above make to range of motion in these taxa. Fowler et al 2011 suggest that Troodontids and Dromaeosaurids basically evolve in opposite directions (cursor, grasper), so it would be really great to see this investigated. Of course, I'm not saying that this needs to be done in a revision of this manuscript, I'm just being enthusiastic about the potential for this method once this paper is published.

We agree! We think this approach has a lot of potential for immediate application to other dinosaurian taxa, and in the short-term future, further application to additional extinct animals. We are excited about getting this work out and into the hands of the community to make that possible. Thanks also for acknowledging that the inclusion of additional taxa here would be beyond the scope of this manuscript, and that this reflects enthusiasm for future work.

Reviewer #2 (Remarks to the Author):

Thank you for inviting me to review Articular surface interactions distinguish dinosaurian locomotor poses by Manafzadeh and colleagues. I am aware of the work the Manafzadeh has been conducting on joint mobility and enjoyed reading this most recent work. It was well written, and I enjoyed learning about this new methodology. I have split my review into some more general comments at the beginning, followed by more specific comments later on.

We thank Reviewer 2 for their thorough critique of our work. We regret that there seems to have been quite a bit of miscommunication between us and this Reviewer. However, Reviewer 2's comments were extremely helpful in revealing their interpretation of our writing, and where that interpretation strayed from our intent. As a result, we were able to identify aspects of our manuscript that could be modified to make our results and their value more clear.

General comments

Central message: I was a bit confused about what the central point of the paper was. From the title, I imagined I would be learning about new reconstructions for dinosaurian poses, but it unclear to me what poses were being distinguished. The method proposed here cannot, for example, identify the different poses involved in guineafowl walking or running, but instead provides an envelope of possible poses that may occur. The method also does not communicate between joints, so there is no way of creating a "locomotor pose", but rather can just create an array of possible locomotor poses.

We apologize for the confusion!

This comment suggests that we might be in semantic disagreement about what a "pose" is – and quite understandably so, given the word "pose" has been used in the orthopedic, anthropological, paleontological, zoological, computer animation, and biomechanical literature to refer to everything from a joint's excursion in a single rotational degree of freedom (e.g., Senter, 2009), to a joint's excursions in all three rotational degrees of freedom (e.g., Manafzadeh & Padian, 2018; this paper), to a joint's excursions in all six degrees of freedom (e.g., Bishop et al., 2023) to the combination of joint "poses" (defined as any of the previous) throughout a limb (e.g., Lauer et al., 2022) or even an entire individual/character. In this manuscript, we are careful to be consistent to use "pose" to refer to a single joint's simultaneous excursions in all three rotational degrees of freedom, and "configuration" to refer to a single joint's simultaneous excursions in all six (three rotational and three translational degrees of freedom). Our goal is to

allow improved reconstruction of dinosaurian locomotor joint poses, as we define joint poses, at individual joints (which of course is a necessary prerequisite to ever reconstructing coordinated series of poses among multiple joints; see additional responses to this Reviewer below).

The other half of this comment seems to suggest a fundamental miscommunication. In all animals, locomotor poses necessarily form an “envelope” or “array” of poses -- not just “possible” poses, but the poses truly used by the animal during steady, forward locomotion. This envelope is formed because (1) joints need to move during locomotion, meaning they will traverse a sequence of poses throughout rotational pose space with each stride/motion cycle, and (2) animals are not robots, meaning there exists a certain amount of natural variation in most joints’ pose usage from stride to stride and at different speeds. The envelope of black points plotted in Figure 2A-E is formed from real poses measured from living guineafowl and used throughout stride cycles. (We note that the majority are shared between walking and running strides.) As a result, we argue that it is not a logical or meaningful goal to search for any single “locomotor pose,” because such a pose does not exist no matter what one’s definition of “pose” is. To effectively reconstruct locomotion, then, we need to instead begin by reconstructing which region of joint pose space forms the very envelope the Reviewer alludes to. In this paper, we demonstrate that our methodology based on bones alone is able to target a small, high-scoring region of joint pose space -- and our experimental data from extant birds demonstrate that this is, in fact, the very same small region of joint pose space those joints occupy during locomotion.

ACTIONS TAKEN:

- We have revised the title to read “locomotor joint poses” instead of “locomotor poses” to emphasize this analysis is on a per-joint basis.*
- We have explicitly defined joint “pose” and joint “configuration” on first usage in the Introduction at Lines 40-41 and 51*

From the abstract, and this line in particular (“Without rigorous criteria for evaluating joint articulation, however, such analyses risk misleading reconstructions of in vivo motion”), I imagined the paper would be about rigorous methods for evaluating joint articulation, and how incorrect methods lead to misleading reconstructions of motion. But the paper neither presented a method for evaluating joint articulation or compared this method to others in terms of reconstructions (showing that this method is more accurate). The reason I say the paper did not present a method for evaluating joint articulation is that it provides a method for quantifying the likelihood of a joint engaging in a pose based on a measure of joint stability, but this does not quantify articulation. Nowhere in this metric can you determine if a joint is articulated or not, just if a joint is likely to engage in a pose during walking/running in a straight line.

We believe there has been a miscommunication of our intent. The articulation score we propose and measure here aims to evaluate the quality of joint articulation (this is now even more explicit in Lines 60-62). An articulation score of 0 represents no potential interaction between articular surfaces, but any articulation score greater than 0 up to 100 represents some level of potential interaction, suggesting that some articular raycasts have successfully reached the mating articular surface (which is exactly how the metric allows you to “determine if a joint is articulated or not”). All of the six-degree-of-freedom configurations of a guineafowl ankle in the figure below have articular surfaces placed such that they can interact, because rays from the tarsometatarsal articular surfaces are successfully hitting the tibiotarsal articular surfaces. However, their articulation scores will vary between just greater than 0 up to 100 based on the extent of their articular surface overlap, symmetry, and congruence.

We have written a large new section of Methods text and created associated supplementary figures specifically in response to this comment and other comments from this Reviewer to attempt to break down in greater detail the way the raycast is conducted and articulation score is calculated, which we are optimistic will make this process more straightforward, clarify its value, and resolve this confusion. We reproduce this new explanation in response to another comment from this Reviewer, below.

To the Reviewer's next point, we do not compare our method for evaluating articulation to existing methods because we unfortunately cannot do so. This is because to our knowledge, no other method exists for quantifying joint articulation at any possible 6-degree-of-freedom joint configuration based on 3-D interactions between bony articular surfaces alone. This lack of existing methodology was the direct motivation for this study and paper; this fieldwide problem is laid out rather explicitly at the end of Manafzadeh & Gatesy (2021), which the Reviewer might have read given their stated familiarity with our work. Please also see our responses to Reviewer 1 and accompanying figures in which we better contextualize this study, above.

Nowhere in the manuscript have we used the words "stable" or "stability," so we are admittedly not certain why the Reviewer has arrived at the impression that our articulation score is meant to capture some measure of stability, specifically. That said, we do think the Reviewer's general intuition is correct, and it is reasonable that joint configurations with a high articulation score may be those that are more stable. However, because such a claim would need to be tested experimentally and because making such a claim is not our goal, we avoid using these terms.

Finally, we clarify that this metric is not probabilistic, nor does it attempt to highlight "likely" poses. Articulation score can be calculated for a pair of bony articular surfaces in any viable, static six-degree-of-freedom configuration. What we find here is that the joint poses used during locomotion happen to consistently fall in the highest-scoring region of pose space -- but we find this using experimental data; the score itself contains no measure of probability and is not inherently tied to locomotion in any way. We again hope the new Supplementary Text and Figures we have added will make this more clear.

ACTIONS TAKEN:

- *Made goal of articulation score more explicit in introduction Lines 60-62*
- *Expanded Methods section to better explain what articulation score measures*
- *Added Supplementary Figures 11-15 to better display what articulation score measures*

What I believe the paper is about is a method for quantifying joint ROM based on a measure of stability, where relatively higher scores within a joint represent increased likelihood of the joint engaging in that pose. As such, it is a methodological paper, where the methodology has been applied to an extinct taxon to show how it can be used to reconstruct motion. Unfortunately, I do not believe anything new was learned about this extinct taxon by using this method, especially as the “whole foot” poses that reconstructed are random and largely informed by the author’s opinions, and not this method, since this method does not reveal any information about covariation in joint positions across the foot.

We again clarify that our articulation score does not measure “stability” or “likelihood” in any way. The score simply offers a quantifier of the quality of joint articulation for any viable, static six-degree-of-freedom configuration of mating bony articular surfaces – not “joint ROM”. Please see our response immediately above.

We do not dispute that our paper has a strong methodological component. The creation of this method is both a natural outcome of the conceptual advance we propose here, as well as a necessary prerequisite for conducting the analyses that reveal a link between articulation score and poses used during locomotion in dinosaurs. Beyond the specific application presented here, this method will also serve a broad community of researchers working to reconstruct vertebrate animal motion, as well as those aiming to evaluate articulation and joint pose usage in living taxa (including, in fact, humans and their joints, both in the general population and in orthopedic patient populations). The Reviewer is correct that after we discovered the correlation between articulation score and locomotor pose usage in dinosaurs, the methodology was applied to Deinonychus to demonstrate its utility in reconstructing motion; this is explicit in Introduction Line 66-67.

The Reviewer’s stated belief that nothing new was learned about the extinct taxon is incorrect. We hope the comments provided by Reviewer 1, a dinosaur expert who has published on Deinonychus, might make clear that much debate remains around the motion of this animal and also highlight that existing work has been largely qualitative. The most quantitative hypothesis of Deinonychus joint motion to date was published by Senter (2009), who relied on manual manipulation and subjective assessment of proper articulation in an attempt to identify each joint’s full ROM (critically, not its locomotor ROM, which is a subset of full ROM [see Manafzadeh et al., 2021]). Notably, Senter’s hypotheses sometimes underestimate even the locomotor ROM we recover here! To make more explicit in the paper that we are contributing new data to our understanding of Deinonychus, we have created new Supplementary Figure 8 to directly compare our results with Senter’s.

The whole-foot combinations of joint poses reconstructed in Figure 3 and Supplementary Movies 2-4 are neither “informed by [our] opinions” nor are they “random.” The Reviewer is absolutely correct in stating that the method we present here does not reveal any new information about temporal covariation in joint positions across the foot. Covariation in joint poses is definitely an important line of evidence to consider when reconstructing locomotion -- two of us have published on this previously (Gatesy et al., 2009; Herbst & Manafzadeh et al., 2022). However, models of covariation are only as useful as the potential locomotor joint poses they are given. Here the hypothesis of covariation we chose to employ, as stated in Line 362, is based on that of avian feet. This is one reasonable hypothesis for non-avian dinosaur intra-pes coordination and is regularly applied by dinosaur paleontologists (e.g., Falkingham & Gatesy, 2014; Gatesy et al., 1999). All joint poses throughout the entire foot, throughout the entire stride cycle reconstructed here, are consistent with and strongly supported by the articulation data we

obtain. Alternative hypotheses of intra-foot coordination could and should certainly be tested in the future; however, the information about what locomotor poses are used at each joint that we uncover here is a necessary prerequisite for any such analysis.

ACTIONS TAKEN:

- *Created new Supplementary Figure 8 comparing our Deinonychus articulation score results to the mobility estimates of Senter (2009)*
- *Created new Supplementary Movie 4 to demonstrate that joints maintain high-scoring raycasts throughout the entire reconstructed stride*

The metric: I like the metric and think it could be useful but have some concerns about its generalizability to other joints in the vertebrate skeleton. For example, I imagine, when applied to vertebrate, you would find that the spine can spin 360° about its long axis! I also wonder what would happen with sliding joints, and whether you would find them being very “unstable” unless they are more-or-less completely overlapping. I do not think this is a general method for all articular joints, and may be limited in usefulness, e.g., to hinge joints.

We are glad the Reviewer likes our metric and understands its future utility, but we find their stated concerns about its generalizability to be overly pessimistic. We are hopeful that the new Methods text might resolve this Reviewer’s concerns about articulation score, but we will also respond to these criticisms directly.

We find it very difficult to imagine how most pairs of vertebrae could ever be found to spin 360 degrees given the existence of interzygapophyseal joints. At many possible joint spacings, zygapophyses would interpenetrate with each other (i.e., bang into each other) at numerous long-axis rotation values. Even among viable (non-interpenetrating) poses, any reasonable articulation analysis of an intervertebral joint would require the maintenance of high articulation score for interactions not only between vertebral centra but also between zygapophyses. Articulation score would quickly decrease as articular facets on the zygapophyses long-axis rotate away from each other and lose overlap. This is shown in new Supplementary Figure 9.

Again reminding the Reviewer that there is no element of “stability” being considered, we agree that sliding joints would suffer in articulation score as joint overlap decreases – as would any other type of joint. We believe this accurately reflects a lower quality of articulation. Please see our new Methods text.

We do not suggest that articulation score analysis will prove to be equally useful for all joints in all taxa, and as we state in Lines 172-175 and 367-371, further work on extant animals will be necessary to identify additional correlations between articulation score and in vivo joint motion and biologically meaningful score thresholds in different clades and regions of the body. However, the basic idea that raycast-based articulation analysis could be applied to other types of joints, and that parameters of overlap, symmetry, and congruence could be measured, should not be uncontroversial. New Supplementary Figure 9 should make this easier to envision.

ACTIONS TAKEN:

- *Created new Supplementary Figure 9 to demonstrate generalizability of articular raycasting, including the specific intervertebral example brought up by this Reviewer.*

The metric also seems to be limited by the lack of communication between body segments. This is less important for some joints and more important for others. For example, where there are

many joints within the same region (e.g., the MTP or PP joints), it is possible that the position of one segment would constrain the ROM of another. Even in areas where joints are far away from each other, segments can constrain ROM (e.g., I can touch my elbows together, meaning the position of my left humerus can limit the ROM of my right shoulder).

This all means that this metric cannot be used to reconstruct locomotor poses, but provide an envelope for range of possible poses of a joint.

We direct the Reviewer to our responses above about “envelopes,” reconstructing a single locomotor pose, and coordination.

Ray casting: the use of ray casting is interesting, but I do not think entirely appropriate. By having a normal vector point from one articular surface to the other, it is suggesting that, if the joint were to move in the direction of the vector, the two articular surfaces would interact with each other. But what is the change of the joint actually moving in that direction? In many cases, extremely low. And even if it was high, the vectors on a curved surface point in a myriad of directions, only one of which can correlate to the motion of the joint. The underlying logic and biomechanical meaning of the ray casting is not clear to me, and seems more-or-less arbitrary.

This comment unfortunately reflects a fundamental misunderstanding of our approach on the part of this Reviewer. We take full responsibility for the lack of clarity of our original text. Above all else, we note that vectors as they are used here do not suggest or imply motion, but rather capture morphological interactions within a static joint configuration, following numerous previous orthopedic workers (e.g., Ateshian et al., 1992; Concorn & Castelli, 2010; Connolly et al., 2009; Lenz et al., 2021).

We aim to make our paper accessible to the broadest possible audience, and we regret that it misled the Reviewer in its original form. We have therefore written new Methods text and created new Supplementary Figures to more clearly explain normal vectors and raycasting. We thank the Reviewer for the motivation to add this section, and we hope they will find it helpful.

ACTIONS TAKEN:

- *Added extensive new Methods text in Articular Raycasting section.*
- *Created associated new Supplementary Figures 11-15.*

One of the benefits of this method, as stated by the study, is that the ray casting takes the bony curvature of the joint into account, making it a better method for estimating joint mobility. The curvature of the joint and its relation to motion and mobility only matters in some, but not all joints. Take the human shoulder joint, for example. The humerus articulates with the glenoid cavity on the scapula, which has some curvature. The curvature of the cavity can be quite flat, leading to many positions of the humeral head which would be subluxations as being completely stable according to this method. I would similarly be surprised if the range of likely joint positions would be accurate for the (human) hip joint, as I imagine most scores would be near 5 (or 100%) regardless of if the joint position is realistic or not, given the surrounding soft tissue. 95-100% window: I discuss below how this could be improved upon, and would stress that this needs to be done, as the range of the window (here, 5%) will likely change depending on which joint is being analyzed. Again, take the human shoulder joint for example. The humerus can occupy a wide range of positions that are equally “stable” and therefore the appropriate window may be 60-100%, and not 95-100%. Determining the actual window for each joint is important, instead of using an arbitrary 95-100%. From Fig 2, it appears wider windows may be needed for

some joints, and narrower windows may be needed for others (e.g., considering the ankle and metatarsophalangeal joints in Fig 2f).

We again clarify that our approach does not claim or aim to measure stability in particular.

We certainly agree that as this method is expanded to different joints from different animals and regions of the body plan, in vivo data from those types of joints will need to have their articulation and pose space occupation studied (as we do here for guineafowl and emu ankles and toes) to determine what biologically meaningful articulation score thresholds will look like – we definitely do not think 95 is a universal solution! Our goal is to demonstrate the type of inferential framework that researchers can adopt when expanding this method to other joints.

We have included a human shoulder joint in Supplementary Figure 9 with the hope of demonstrating that even though the glenoid is comparatively flat, its osteological curvature certainly still affects its articulation, and is more than sufficient to result in different articulation score parameters in different six-degree-of-freedom joint configurations. However, as stated above, articulation analysis and pose space occupation during different behaviors for the human shoulder would have to be conducted on living individuals to understand how articulation score relates to in vivo joint pose usage, and what score threshold is meaningful for any particular research question.

Regarding the Reviewer's comment on Figure 2, we emphasize that the positioning of the black in vivo data points in the vertical axis is arbitrary (see Figure caption), and the fit in the horizontal flexion-extension dimension is maintained consistently across all joints.

ACTIONS TAKEN:

- *Added human shoulder joint to Supplementary Figure 9 to make clear that it is also amenable to articulation analysis.*

I'm sorry my review is not more positive at this time. I very much enjoyed reading the manuscript, but have too many concerns to recommend for publication at this time.

We are optimistic that the substantial additional work we have done to make this manuscript more clear will resolve this Reviewer's concerns.

Specific comments

Abstract

L18: "... reconstruction of in vivo motion." Is an oxymoron, as reconstruction implies motion is unknown, but in vivo would imply the motion was measured when the animal was alive. The abstract is additionally vague up until this point, as the authors have not specified they are only considering vertebrates. I would change to "... reconstructions of vertebrate motion."

We accept this change.

ACTIONS TAKEN:

- *Change made on Line 18*

Introduction

L32: This first sentence is not correct. Vertebrate animal motion does not rely predominantly on the mobility of the joint, but rather the neuromuscular and musculoskeletal system (muscle

origins/insertions, activation patterns and co-contractions, etc.). Joint mobility affects the range of motion, and how many times that motion can be performed. E.g., hypermobile people have high joint mobility, but subluxed / disarticulated joints only cause minor changes in joint mobility compared to, e.g., removing all the flexor muscles that cross a joint.

We have rephrased to make this less contentious.

ACTIONS TAKEN:

- *Cut “predominantly” on Line 32*

Results

L78: It appears from Fig 2d and Fig 2e that this is only sometimes true. I cannot tell if in Fig 2e, abduction-adduction, how much this is true given the density of the black points. A statistic would be nice here (e.g., “92.3% of the locomotor poses fall within the highest-scoring...”) Alternatively, and probably more usefully, what range would be needed to ensure all locomotor poses fall within the range? The 95-100% range seems arbitrary, and it would be more useful if information like “a 89.5-100% range is needed to reliably encompass all locomotor poses during walking and running...” This is somewhat done in L82, but a more exact range would be useful, because that could then be applied to other animals.

For 1.9% of all sampled poses -- critically exclusively during the highly flexed mid-swing phase of a single experimental trial of highest-speed running -- articulation score deviates below the 95-100 range because joints long-axis rotate by up to 5 degrees beyond the region of pose space with articulation score of 95-100 (see Figure 2). 98.1% of measured poses fall within the 95-100 range. This information is now included in the text.

The 95-100 range selected for application to Deinonychus is not arbitrary; its contour matches the region of pose space occupied by remainder of locomotor poses remarkably well, as visible in Figure 2, and includes all walking as well as the vast majority of running poses. We reiterate that natural variation in locomotion is also to be expected, and the same highly flexed flexion-extension excursions, when reached at other times during locomotion with more internal rotation, are included within the set of joint poses in the 95-100 set.

ACTIONS TAKEN:

- *Analysis conducted and additional detail added as requested in Lines 86-90.*

L88-90: This sentence is confusing, as it sounds like you can predict differences between guineafowl and emu joints. I do not think that is what the authors meant, and if it is, I am unsure why they are looking at this (given the rest of the paper) or how they made this comparison.

We reiterate that articulation score is calculated based on the morphology and configuration of static bony articular surfaces alone. When calculating this score for guineafowl and emu joints, we found differences in the pattern of articulation score distribution throughout joint pose space between the two species. When we plotted independently obtained in vivo pose data for these joints in joint pose space (Figure 2d-e), we found that a match between the highest-scoring region of pose space and true locomotor poses was maintained in both guineafowl and emu. In other words, the highest-scoring regions of pose space differed in the same way that the regions of pose space occupied by locomotor poses differed. Therefore, articulation score analysis based on bony shape alone was effectively predictive of differences in locomotor poses used by the two animals, as these lines state.

ACTIONS TAKEN:

- *This confusion should be resolved by the new Methods text and Supplementary Figures 11-15.*

L101: if better envelopes than 95-100 were constructed, using the suggestion of the above comment, this could lead to large changes in this reconstruction, as it opens up the range of possible joint positions.

We contend that it would not change the reconstruction, because these poses would not be ruled out even if the envelope were expanded as the Reviewer suggests. 95-100 is a conservative range in that we are confident these poses are used in locomotion. Additionally, 95-100 was found to capture all walking poses for guineafowl (deviation from this region was only found during the swing phase of highest-speed running, see Lines 86-90 and responses above), and the stride reconstructed for Deinonychus is a walking stride.

L130: This method did not, unfortunately, "...enabled us to reconstruct a full potential six-degree-of-freedom stride cycle for a non-avian paravian dinosaur..." Multibody dynamic analysis enabled this reconstruction. What this method did was provide potentially more accurate input parameters (here, constraints) on the model, which potentially lead to a more accurate reconstruction. But even so, this cannot be said until a reconstruction is done on an extant animal, and the accuracy of using traditional joint constraints and these joint constraints are compared to some ground-truth metric (which can be done with the data presented in this paper).

We clarify we never mention "multibody dynamic analysis" nor do we use that approach here. The reconstructed animation was generated through a combination of inverse and forward kinematic rigging; see Lines 361-364.

We have altered our phrasing in Line 141 to remove "full," emphasizing that this is a potential stride cycle. We agree with the Reviewer's statement that our method provided more accurate input constraints on our reconstruction, which when combined with a hypothesis of coordination (see our responses above), yielded the stride cycle.

However, the final sentence of this comment again reflects some miscommunication. We calculated articulation score for extant avian joints based on data from their osteological morphology alone. We also independently collected experimental data for the poses these same joints use during locomotion. Upon comparing these two datasets, we found that locomotor poses consistently fell within the highest-scoring regions of pose space as identified based on osteological morphology alone. This is the very ground-truthing the Reviewer appears to be asking for, and we have now made this more clear in text Lines 78-90. We reiterate that we are unaware of any other existing approach for quantifying joint articulation in any six-degree-of-freedom configuration and therefore cannot conduct a comparative analysis with existing approaches.

ACTIONS TAKEN:

- *Change to phrasing in Line 141 and 78-90.*

L156-62: The discussion in this paragraph is nice, but does highlight the limitations of the currently proposed method for very specific types of joints.

Thank you! And we are hopeful new Supplementary Figure 9 and our response above will assuage the Reviewer's concerns about generalizability.

ACTIONS TAKEN:

- *Created new Supplementary Figure 9 to make it more explicit that this method is not limited to very specific types of joints.*

Methods

L257: what qualified as a high-scoring pose? 95-100%? It is not clear from the writing here

The original phrasing of "within 15% of overall maximum score" might have been confusing, so we have modified phrasing to make clear that this is equivalent to a score of 85-100.

ACTIONS TAKEN:

- *Changed phrasing to clarify in Line 331.*

Reviewer #3 (Remarks to the Author):

The submitted manuscript describes a new approach to accurately determine the joint position and range-of-motion of limb bones groundtruthing the method using modern bird species and applying the findings to a fossil dinosaur. This is an interesting and well-written paper with lots of potential to elucidate locomotor the palaeobiology of extinct species (however this is only done in a limited capacity, presumably with more in-depth fossil work reserved for a follow-up paper). I do have several questions/suggestions with regards to the methodology but these are all fairly minor.

Kind regards,
Stephan Lautenschlager

We thank Dr. Lautenschlager for this positive review of our work.

Specific points:

- I found the scoring quite difficult to understand beyond the initial three parameters (overlap, symmetry, congruence) and a more detailed explanation/example would be beneficial. Related to this, the scoring seems to evaluate facet overlap but not necessarily dislocation (e.g. a dislocated joint which is essentially translated so that the overlap and symmetry stay the same but distance between corresponding facets increases or decreases will still score highly). How is this taken into account?

Thank you for this feedback. We have written an extended portion of text with accompanying supplementary figures to make the process of raycasting and score calculation more clear. In brief, what you term "dislocation" here (translating two surfaces farther apart) results in a change in congruence (and therefore articular score) because the raycast hit locations change. We have included an example demonstrating this in new Supplementary Figure 14.

ACTIONS TAKEN:

- *Added new Methods text with Supplementary Figures 11-15 to offer a more detailed explanation.*

- Considering the huge number of scenarios tested there is no account of the computing power necessary to run the analyses, used machine specs, time taken for a full run, etc. while raycasting itself is manageable on most standard computers, the visualisation, especially of high-res models, may push a graphic card to its limits.

Perhaps surprisingly, all analyses described here were conducted on a Macbook Pro laptop and behaved remarkably well -- likely because after interpenetration is evaluated, the raycasting analysis requires consideration only of small articular patches rather than entire bone models. It is a great idea to make this more clear so that readers realize no specialized equipment is required.

ACTIONS TAKEN:

- *Added detailed information about computer specifications and average analysis time in Lines 326-328.*

- Data availability: I don't think "Availability on request" statements are appropriate anymore and the respective data should be deposited in a public repository

This is a very fair point. Unfortunately, the emu CT data we were kindly sent by John Hutchinson for use in this study are still being studied by his group and cannot be made publicly available at this time. That said, our sensitivity analyses of comparison among guineafowl individuals suggests that any emu specimen, widely available in global museum collections, should yield comparable results.

We are very happy to make this paper more useful to the paleontological community by depositing the Deinonychus meshes in Morphosource and have now received Yale Peabody Museum approval to do so; we will do so immediately upon acceptance of this manuscript.

ACTIONS TAKEN:

- *Will deposit Deinonychus meshes in Morphosource immediately upon manuscript acceptance.*

Line 57: I doubt many readers will be familiar with raycasting and a brief description how this works would be useful.

We agree. The new Methods text and Supplementary Figure 11 now address this.

ACTIONS TAKEN:

- *Added new Methods text and Supplementary Figure 11*

Line 233: I understand that a full 360 degree rotation in a hypothetical possibility but practically impossible. Related to the main comment above, would it make sense to constrain rotation in the first place to minimise computational effort?

In this paper, we were exhaustive in our search to show how much of full joint pose space must be considered without an articulation constraint. With this foundation now established, moving forward, future researchers could certainly set limits on long-axis rotation for some types of joints (e.g., bicondylar ones) to minimize computational effort. However, we hesitate to make

this recommendation explicitly in the text, because making decisions about how to constrain rotation requires a certain degree of preliminary exploration with the specific joint system at hand.

Line 236: If I understand this correctly, translation is tested for seven discrete steps in each direction. How/which magnitude was selected here?

That is correct! The magnitude is selected as indicated in Supplementary Table 1, based on multiples of the dimensions of cylinders fit to the distal condyles of the proximal bone in each studied joint. We have added a new Supplementary Figure to display this process more clearly.

ACTIONS TAKEN:

- *Created new Supplementary Figure 10.*

Line 242: How were collisions/interpenetrations detected. Is this automatically possible in Maya or was this coded into the script (e.g. negative distances of the ray casts)?

Thank you for catching this; we should include this in our methods. Interpenetration was detected here based on presence of at least one positive dot product between (1) the vector connecting a point on one mesh and its closest point on the other and (2) the surface normal of that closest point (i.e., indicating that the two vectors form an obtuse angle and that the vertex of mesh #2 is therefore within mesh #1.) We note that this can be replaced with any other interpenetration detection method as desired, including the Boolean-based approach we have applied previously.

ACTIONS TAKEN:

Detail added to Methods in Lines 252-255

REFERENCES FOR ALL RESPONSES

*Ateshian, G., Rosenwasser, M. P., & Mow, V. C. (1992). Curvature characteristics and congruence of the thumb carpometacarpal joint: differences between female and male joints. *Journal of biomechanics*, 25(6), 591-607.*

*Bishop, P. J., Brocklehurst, R. J., & Pierce, S. E. (2023). Intelligent sampling of high-dimensional joint mobility space for analysis of articular function. *Methods in Ecology and Evolution*, 14(2), 569-582.*

*Brocklehurst, R. J., Fahn-Lai, P., Regnault, S., & Pierce, S. E. (2022). Musculoskeletal modeling of sprawling and parasagittal forelimbs provides insight into synapsid postural transition. *iScience*, 25(1), 103578.*

*Concorn, M., & Castelli, V. P. (2010). A kinematic model of the tibio-talar joint using a minimum energy principle. In *ROMANSY 18 Robot Design, Dynamics and Control: Proceedings of The Eighteenth CISM-IFTOMM Symposium* (pp. 347-356). Springer Vienna.*

Connolly, K. D., Ronsky, J. L., Westover, L. M., Küpper, J. C., & Frayne, R. (2009). Analysis techniques for congruence of the patellofemoral joint.

Demuth, O. E., Rayfield, E. J., & Hutchinson, J. R. (2020). 3D hindlimb joint mobility of the stem-archosaur *Euparkeria capensis* with implications for postural evolution within Archosauria. *Scientific reports*, 10(1), 15357.

Falkingham, P. L., & Gatesy, S. M. (2014). The birth of a dinosaur footprint: subsurface 3D motion reconstruction and discrete element simulation reveal track ontogeny. *Proceedings of the National Academy of Sciences*, 111(51), 18279-18284.

Fowler, D. W., Freedman, E. A., Scannella, J. B., & Kambic, R. E. (2011). The predatory ecology of *Deinonychus* and the origin of flapping in birds. *PLoS One*, 6(12), e28964.

Gatesy, S. M., Middleton, K. M., Jr, F. A. J., & Shubin, N. H. (1999). Three-dimensional preservation of foot movements in Triassic theropod dinosaurs. *Nature*, 399(6732), 141-144.

Gatesy, S. M., Bäker, M., & Hutchinson, J. R. (2009). Constraint-based exclusion of limb poses for reconstructing theropod dinosaur locomotion. *Journal of Vertebrate Paleontology*, 29(2), 535-544.

Haering, D., Raison, M., & Begon, M. (2014). Measurement and description of three-dimensional shoulder range of motion with degrees of freedom interactions. *Journal of biomechanical engineering*, 136(8).

Herbst, E. C., Manafzadeh, A. R., & Hutchinson, J. R. (2022). Multi-joint analysis of pose viability supports the possibility of salamander-like hindlimb configurations in the Permian tetrapod *Eryops megacephalus*. *Integrative and Comparative Biology*, 62(2), 139-151.

Kambic, R. E., Roberts, T. J., & Gatesy, S. M. (2017). 3-D range of motion envelopes reveal interacting degrees of freedom in avian hind limb joints. *Journal of Anatomy*, 231(6), 906-920.

Kambic, R. E., Biewener, A. A., & Pierce, S. E. (2017). Experimental determination of three-dimensional cervical joint mobility in the avian neck. *Frontiers in Zoology*, 14(1), 1-15.

Lenz, A. L., Krähenbühl, N., Peterson, A. C., Lisonbee, R. J., Hintermann, B., Saltzman, C. L., ... & Anderson, A. E. (2021). Statistical shape modeling of the talocrural joint using a hybrid multi-articulation joint approach. *Scientific Reports*, 11(1), 7314.

Lauer, J., Zhou, M., Ye, S., Menegas, W., Schneider, S., Nath, T., ... & Mathis, A. (2022). Multi-animal pose estimation, identification and tracking with DeepLabCut. *Nature Methods*, 19(4), 496-504.

Mallison, H. (2010). The digital Plateosaurus II: an assessment of the range of motion of the limbs and vertebral column and of previous reconstructions using a digital skeletal mount. *Acta Palaeontologica Polonica*, 55(3), 433-458.

Manafzadeh, A. R., & Gatesy, S. M. (2021). Paleobiological reconstructions of articular function require all six degrees of freedom. *Journal of Anatomy*, 239(6), 1516-1524.

Manafzadeh, A. R., Kambic, R. E., & Gatesy, S. M. (2021). A new role for joint mobility in reconstructing vertebrate locomotor evolution. *Proceedings of the National Academy of Sciences*, 118(7), e2023513118.

Manafzadeh, A. R., & Gatesy, S. M. (2022). Advances and challenges in paleobiological reconstructions of joint mobility. *Integrative and Comparative Biology*, 62(5), 1369-1376.

Manafzadeh, A. R., & Padian, K. (2018). ROM mapping of ligamentous constraints on avian hip mobility: implications for extinct ornithomirans. *Proceedings of the Royal Society B: Biological Sciences*, 285(1879), 20180727.

Mitrovic, D. R. (1977). Development of the metatarsophalangeal joint of the chick embryo: morphological, ultrastructural and histochemical studies. *American Journal of Anatomy*, 150(2), 333-347.

Ostrom, J. H., & Gauthier, J. A. (2019). *Osteology of Deinonychus antirrhopus, an unusual theropod from the Lower Cretaceous of Montana*. Yale University Press.

Pierce, S. E., Clack, J. A., & Hutchinson, J. R. (2012). Three-dimensional limb joint mobility in the early tetrapod *Ichthyostega*. *Nature*, 486(7404), 523-526.

Richards, H. L., Bishop, P. J., Hocking, D. P., Adams, J. W., & Evans, A. R. (2021). Low elbow mobility indicates unique forelimb posture and function in a giant extinct marsupial. *Journal of Anatomy*, 238(6), 1425-1441.

Schaeffer, B. (1941). The morphological and functional evolution of the tarsus in amphibians and reptiles. *Bulletin of the AMNH*; v. 78, article 6.

Senter, P. (2009). Pedal function in deinonychosaurs (*Dinosauria: Theropoda*): a comparative study. *Bulletin of the Gunma Museum of Natural History*, 13, 1-14.

Tehrani, P. R., Gilanpour, H., & Veshkini, A. (2017). Radiographic anatomy of the Metatarsophalangeal joint and digits of the ostrich (*Struthio camelus*). *Journal of avian medicine and surgery*, 31(3), 198-205.

Reviewers' Comments:

Reviewer #1:

Remarks to the Author:

Firstly I apologise for taking a little longer than usual for this review. I'm in the field and so my access to a laptop or the internet is very limited.

I have read through the revised manuscript and especially commend the authors on their well explained reply to reviewers document.

I did not have many comments on the original manuscript, so I do not have too many things to follow up on.

Mainly I will focus on this passage in the reply letter:

"Regarding the specific point/concern about MT-IV: our analyses did not recover the "increased flexibility in lateral positioning" discussed by this Reviewer and in Fowler et al. (2011), which is why it is not mentioned in our manuscript. Given gross comparison of the dromaeosaurid MTP-IV joint with MTPs of extant birds and other reptiles (see Mitrovic, 1977; Tehrani et al., 2017; Schaeffer, 1941), and considering the general conservation of soft tissue structures (specifically general ligamentous architecture) in this region of the Bauplan across tetrapods, we suggest that the "ball-like" distal facet of metatarsal IV should not be interpreted as creating a true ball-and-socket joint. Rather, we suggest that as part of an (albeit derived) hinge joint, the fourth metatarsal must still maintain general symmetry between the medial and lateral halves of its articulation with the receiving facets on pedal phalanx IV-1.

"

I am not convinced that the MT-IV and D-IV-1 proximal articulation are correctly determined.

Ostrom (1969) notes (p131), "Metatarsal IV, however, has a broadly rounded (transversely and longitudinally) distal articular facet, quite unlike the extremities of II and III, without the slightest suggestion of a fore-aft groove".

And (bottom of p131)

"The articular facets of all phalangeal elements are well-formed and highly finished. With the exception of the proximal surface of the first phalanx of digit IV, all proximal phalanges feature prominent lateral and medial concavities, separated by a vertical ridge."

Ostrom does not comment further. I will not include images of the elements but they are figured adequately (though not in great detail) in Ostrom (1969).

As with our own work (Fowler et al 2011), Ostrom's passage contradicts the above statement from the reply letter in that there is only one receiving facet on D-IV-1, it is cup shaped, and unlike all other facets on the pedal phalanges. Given the shape of this facet and the corresponding distal end of MT-IV, I doubt very much that it was as restricted in its movement as that of the ginglymoid facets of D-II, II, and all the other phalanges of D-IV.

The joint between MT-IV and D-IV-1 is unusual and unlike the other joints. Indeed, in other theropods which have ginglymoid MT-II and III, MT-IV similarly has a more rounded joint. I would certainly not suggest that it had a similar mobility to a true ball and socket joint (e.g. the hip), but rather (based on physical manipulation) it has more lateral flexibility than D-II or D-III which, due to the ginglymoidy, are restricted to movement in a single plane. I can imagine this lateral flexibility may assist in a stabilizing function when walking, or maybe when manipulating food. It is hard to see how the current

(new) analytical method failed to find any difference between MT/D-IV and MT-II and III. So I suppose on this point I disagree with the authors' findings.

I appreciate the suggestion that their method is repeatable... in reality, repetition is generally only available to those with expertise and access to technology. That's fine, I'm not going to reject the manuscript, it can be published as is, but I do not think that the analysis of the MT-IV D-IV-1 joint is likely to be correct (or maybe there is a nuance that I am missing since I am not a computational biomechanist). This is important as there are many curious differences among theropod clades regarding the articular surfaces of pedal joints and the shapes of the phalanges and unguals. In Fowler et al (2011) we note the stark differences between basal dromaeosaurs, derived dromaeosaurs, and troodontids (since these groups form a single clade). However, there are even more small and large differences within other clades around the origin of birds, and indeed within birds. I reject the idea that feet are just for walking (which has been a general assumption in dinosaur paleobiology), and even that walking cycles are broadly similar across taxa. Walking may be a daily function for legs, but not necessarily strongly selected for when compared with more critical and derived functions (at least in some clades). Differences in phalangeal morphology may be more indicative of diet, foraging, or other more strongly selected behaviors, and I am just cautious about the potential for narrowing of colleagues' interpretation and expectations based on this result.

Overall the paper and method are interesting and a good piece of research. Publish as is.

Reviewer #2:

Remarks to the Author:

I thank the authors for making edits to the manuscript, and the editor for inviting me to re-review this submission. I began by reading all reviewer's comments, as well as the responses to the reviewers, before re-reading the manuscript, so I could have these edits in mind. I found it helpful to better understand the edits the authors have made, and it gave me several points to look out for while reading the manuscript.

There is a point R1 made which matched some of my comments before, and as such I would like to address some of the author's response to R1, and mix it in with some comments from my review (page 2-5, 13):

I agree with reviewer 1 about taking caution in the novelty in this advance. As pointed out by R1, some of the language used in the manuscript (and response to reviewers here) is too strong. I have pointed out some aspects of this throughout my second review.

Page 2: this is tangential to the paper, but the earliest ROM reconstructions from skeletal material were completed decades before 2010. For example, the work on masticatory biomechanics where gape angle is estimated from skeletal collections (e.g., Bill Hylander, Callum Ross, Chris Vinyard, Elizabeth Dumont). These studies generally investigated mandibular rotation (i.e., opening the mouth and twisting about the superior/inferior axis) and translation of the TMJ (usually in an anterior/posterior direction).

Page 4: the idea that translation and rotation of a joint must be considered together, simultaneously, is not new to (paleo)biomechanics. In addition to the mandible example (discussed above) this has also been considered at length in the human knee.

Page 4/5: I agree with the reviewers that a) the inclusion of a six DoF joint into biomechanical models and b) the inclusion of more complex joints into paleontology is unique (for some subfields), but agree with the reviewer that the authors do not have the data to make such authoritative statements.

Page 5: the author's express a desire to have the response to the reviewers included in their manuscripts, and I agree, such information is needed in this one. This was one of the big issues I had with this manuscript the first time I read it: I do not believe Nat Comms provides the space necessary for the authors to properly discuss the topic or methodology due to space limitations of the journal.

Page 13: I understand the author's feel I am being overly pessimistic about the generalizability of this metric, but, unfortunately, the author's have not offered sufficient evidence to the alternative, and my

concerns are supported by the evidence presented in supp. Fig 9 (with regards to the glenohumeral joint). My issue was not whether or not you could use the raycasting method on other joints (of course you can!), it's about whether the results will be useful. The authors have expanded their limitations discussion in the manuscript, which is appreciated, but I believe you should have a better understanding of the limitations of a method in a manuscript which is stating, quite conclusively "this is the way forward."

This speaks to the "dual nature" of the manuscript: one being the reconstruction of the pedal pose of an extinct animal, and the other presenting the new method. It is, essentially, two studies wrapped into one, and being presented in a short communication.

Page 9: I thank the authors for their clarification, there was indeed a miscommunication. Pose certainly does need to be defined, and early on in the manuscript (but see comments on this manuscript below). In its current form, joint poses (line 51), is now defined as rotational combinations of joint articulations, which I do not believe is what the author's mean the definition to mean. I believe a proper definition would be something closer to "position of body segments, relative to each other, that are connected by a joint". This is closer to the definition given in L40-41 for a different term (joint configurations). As an aside, the authors also misunderstood my comment, as I was not suggesting in my comment that motion in animals was deterministic (and neither is motion in robots, not the least bit because of the inclusion of AI and machine learning, but also because of things like vibrations and inertial momentum, making precision manufacturing by robots difficult/impossible today 😊) For the title, I would recommend changing "locomotor joint poses" to "joint poses during locomotion" for clarity.

Page 11: In response to there being no other methods present, It does not need to be another 6 DoF method, it just needs to be another method for estimating joint poses, and the authors have actually discussed an alternative methodology previously in this review (i.e., the "prism method" for estimated ROM). While imperfect, it would be useful to see how results from that method, or the "qualitative method" of joint poses discussed later in the response to R1, compared to results from this method. If results are similar in terms of locomotor reconstruction (forward dynamic simulations), then does this more complicated ray casting method of estimation actually need to be used? If not, it creates a good argument for using this model (which has its own inaccuracies) over other models. I stand by my earlier comment that a comparison is needed to provide tangible evidence for how this method is superior and provides superior estimates for extinct animals.

If higher articulation scores do not represent stability, what do they biologically represent? This needs to be explicitly stated in the manuscript, to prevent others from making the same mistake I have. A definition of articulation early on would clear up some of this confusion, as the authors are using terms like "disarticulation", "misalignment", and "subluxation," which compromise the stability of the joint, to explain improper articulation, but are also stating that articulation scores (and thereby, articulation) do not represent stability.

While the authors state that the articulation metric is not probabilistic, they are treating it like a probability in their analysis. I.e., scores above 95 were used further means the probability that scores less than 95 would be engaged in is low. They also discuss it in a probabilistic manner in the manuscript (see comments about current manuscript below). They are not using it as a 95% CI, but they are using it as a probability (i.e., the probability that the joint will be in that pose, due to an articulation score which has been validated with a subset of joints from two species)

Page 12: The authors write "The score simply offers a quantifier of the quality of joint articulation for any viable, static six-degree-of-freedom configuration of mating bony articular surfaces – not "joint ROM"." The authors are, in fact, quantifying joint range of motion based on their articulation scores: high articulation scores = inclusion in ROM, low articulation scores = exclusion. In their study, ROM is defined as having an articulation score of 95+.

With regards to the whole-foot combinations of joint poses, part of the reason I used the words "opinion" and "random" is due to the writing of the author's on L290 (L362 in the resubmission). The authors write "An inverse kinematic animation rig was used to animate a rough stride cycle for the pes of Deinonychus using inspiration from published extant avian toe tip motion" To me, this sentence comes across as the author's constructed a model of extant avian toe tip motion, and then – combined with the joint poses – used it as inspiration to create the combined joint poses presented in the paper.

In this case, inspiration would be similar to an artist using a set of trees in a forest as inspiration to create one, composite tree. In this case, the composite tree is influenced by the author's opinion heavily, with no necessary protocol (i.e., random) as to why some features were chosen, and others were not. If this is what the author's did, then my comment stands true. If not, then this again speaks towards not enough detail being given in the paper for the study to be properly explained or replicated.

Page 13: as stated above, the authors have misunderstood my comment. It is not whether this method can be applied to all joints, but rather if the results would be useful or not. From supplementary Figure 9, it certainly seems it would be less than useful for the human glenohumeral joint, as it appears some positions that are subluxations would come across as having high articulation scores, which is a fundamental shortcoming of this method.

Page 14: Thank you for clearing up explanations in the review – what the author's are trying to accomplish is much clearer in the response to the reviewers. My (simplified) understanding of ray casting is now as follows

- The more rays that connect one articular surface to the other, the more symmetric ray lengths are... etc... the more "articulated" the joint is, and therefore the higher the articulation score

- Higher articulation scores are correlated with joint poses in extant animals, and therefore are correlated to the way the body segments move relative to each other. Now, this implies one of two things

- 1) this is a random correlation, in which case it is biologically meaningless

- 2) this is a causative relationship, whereby articulation scores are related to joint motion (rotation and translation)

The authors believe this is a causative relationship (e.g., see wording in abstract, discussed below), which is why they can use it to reconstruct extinct vertebrate locomotion.

Assuming this is true, my question from before still stands: what is the underlying logic and biomechanical meaning of ray casting? I.e., why, functionally, was ray casting work? It is touched on in the paper, but not explained. E.g., why does asymmetry in ray length matter? Why does having more rays connecting the two surfaces when there is space between them biomechanically matter? Otherwise it appears to be more-or-less arbitrary why ray casting was chosen, and why it works in this paper.

Page 16: If the point of this method is to create an envelope of possible joint poses, the authors do not need to identify what percentage of sampled poses fall within the 95-100 articulation score envelope, but rather find the articulation score envelope that fits 100% of the joint poses. Otherwise, the authors are contradicting themselves (or believe 1.9% of the in vivo data is incorrect). I stand by my comment that articulation scores from 95-100 should not be used, but rather this window needs to be calculated from the in vivo data, and then used to re-reconstruct extinct animal locomotion.

Page 17: "We contend that it would not change the reconstruction, because these poses would not be ruled out even if the envelope were expanded as the Reviewer suggests." The authors have misunderstood my comment. Of course, the poses would not be ruled out, but new poses that were ruled out before could now be chosen.

Page 17: I misunderstood the previous read of the manuscript, and had thought the authors had run forward dynamic simulations, using the joint envelopes as boundary conditions. It is now my understanding the authors used the extant data to roughly position the bones of the extinct animal in the correct position, then manually altered joint positions so they fit within this envelope of possible positions. If this is correct, there is certainly a degree of artistic flair in the reconstruction that needs to be discussed, and the sensitivity of these reconstructions (inter- and intra-observer error) needs to be looked at, given this method is being advertised as a "new way forward" (my words, not the authors) for reconstructing joint poses and locomotor poses of extinct animals.

Page 17: "However, the final sentence of this comment..." the authors have misread my comment. What the authors have "ground-truthed" was whether they could put joints in realistic poses. What I was saying what you cannot say your method has led to a more accurate reconstruction of the gait of the extinct animals because 1) it was not compared to results obtained using traditional joint constraints, and 2) it was not compared to how this extinct animal actually moved. If you wanted to use extant models for this purpose, you would have to take the data from one of the extant animals

and do a similar reconstruction that you did with the extinct animal (potentially with bones from another individual of that extant species – e.g., another guinea fowl or emu) using both traditional joint constraints and these new constraints (estimated using ray casting) and see which one produces results closer to your experimental data. Again, if both produce similar results, the traditional method is easier and takes less time, so why should we bother with raycasting?

Resubmission

Abstract: "... locomotor joint poses consistently have high articulation scores. We then exploited this predictive relationship to constrain reconstruction of a pedal stride cycle..." These lines state that it is possible to predict joint poses based on articulation scores, but the authors failed to show (or statistically test) any type of predictive relationship between articulation score and pedal stride cycle in their extant individuals. That is to say, they showed pedal stride cycles were encompassed in the polygonal space created by articulation scores (you can predict the number of legs a zebra has), but never that articulation scores can predict pedal stride cycle (you can predict an species by knowing the organism has four legs).

L33: I must have missed it the last time – a joint is not an organ

L40-41: this definition of joint configurations (the excursion of all three rotational and translational DoF) are what the authors stated was "poses" in the response to the reviewers.

L48-49: much related to gape in primates has been ground-truthed, so this statement is incorrect

L81-82: In their response, the authors stressed that they were not discussing the "probability" of a joint ending up in a pose, but the writing here ("... mapping our articulation scores onto this mobility estimate enhances our knowledge of each pose beyond a binary "possible" or "impossible"..."") implies they are bringing joint articulation out of a deterministic, and into a probabilistic space. The authors are contradicting themselves.

L86-90: This is a major issue with the 95-100 window, as it does not encompass all possible ranges of motion. As these envelopes of possible joint poses are being used as boundary constraints for models of locomotion for extinct animals, they must encompass the entire possible ROM for the extant animals, otherwise they are poorly defined constraints.

L95-98: Text needs to be added to the supplementary material explaining the sensitivity analyses. As it stands, it is not clear what sensitivity analysis was done from the figures and captions alone. Results from the sensitivity analysis should be explained as well.

L140-143: It was not the findings in this paper that enabled the reconstruction of the stride cycle, but "using the findings of this paper as constraints for XXX simulations..."

I very much appreciate the expanded limitations discussion, and the expanded materials and methods. The additional information on the method has helped immensely.

Reviewer #3:

Remarks to the Author:

I would like to thank the authors for thoroughly addressing my comments and implementing suggestions. From my perspective all points have been addressed and the updated manuscript has been greatly improved. I would consequently recommend publication now.

Manafzadeh et al. (2023) *Nature Communications* Responses to Reviewers (Round 2)

We are grateful that all three Reviewers kindly took the time to re-review our manuscript. We are also grateful that the Editor continues to evaluate our submission favorably and has allowed us the opportunity to respond to the Reviewers once more.

In the initial round of review, all three Reviewers raised concerns related to clarity and the broader accessibility of our manuscript. We responded to these concerns with extended responses as well as substantial revision and expansion of our manuscript and supplementary information.

Following these changes, in the second round of review, both R1 and R3 expressed satisfaction with our revisions and stated that they are now happy for the manuscript to be published as is. However, R2 has again raised a number of concerns, many of which reiterate concerns they already voiced and to which we had already responded in the previous round.

Here we respond to all comments point by point with the hope that this additional clarification will have made our manuscript suitable for publication in its current form. Additional changes we have made to our submission as a result of these comments are **highlighted** below.

Reviewer #1 (Remarks to the Author):

Firstly I apologise for taking a little longer than usual for this review. I'm in the field and so my access to a laptop or the internet is very limited.

I have read through the revised manuscript and especially commend the authors on their well explained reply to reviewers document.

We thank Reviewer 1 for taking the time to re-review our manuscript, especially from the field.

I did not have many comments on the original manuscript, so I do not have too many things to follow up on.

Mainly I will focus on this passage in the reply letter:

“Regarding the specific point/concern about MT-IV: our analyses did not recover the “increased flexibility in lateral positioning” discussed by this Reviewer and in Fowler et al. (2011), which is why it is not mentioned in our manuscript. Given gross comparison of the dromaeosaurid MTP-IV joint with MTPs of extant birds and other reptiles (see Mitrovic, 1977; Tehrani et al., 2017; Schaeffer, 1941), and considering the general conservation of soft tissue structures (specifically general ligamentous architecture) in this region of the Bauplan across tetrapods, we suggest that the “ball-like” distal facet of metatarsal IV should not be interpreted as creating a true ball-and-socket joint. Rather, we suggest that as part of an (albeit derived) hinge joint, the fourth metatarsal must still maintain general symmetry between the medial and lateral halves of its articulation with the receiving facets on pedal phalanx IV-1.

“

I am not convinced that the MT-IV and D-IV-1 proximal articulation are correctly determined.

Ostrom (1969) notes (p131), “Metatarsal IV, however, has a broadly rounded (transversely and longitudinally) distal articular facet, quite unlike the extremities of II and III, without the slightest suggestion of a fore-aft groove”.

And (bottom of p131)

“The articular facets of all phalangeal elements are well-formed and highly finished. With the exception of the proximal surface of the first phalanx of digit IV, all proximal phalanges feature prominent lateral and medial concavities, separated by a vertical ridge.”

Ostrom does not comment further. I will not include images of the elements but they are figured adequately (though not in great detail) in Ostrom (1969).

As with our own work (Fowler et al 2011), Ostrom’s passage contradicts the above statement from the reply letter in that there is only one receiving facet on D-IV-1, it is cup shaped, and unlike all other facets on the pedal phalanges. Given the shape of this facet and the corresponding distal end of MT-IV, I doubt very much that it was as restricted in its movement as that of the ginglymoid facets of D-II, II, and all the other phalanges of D-IV.

The joint between MT-IV and D-IV-1 is unusual and unlike the other joints. Indeed, in other theropods which have ginglymoid MT-II and III, MT-IV similarly has a more rounded joint. I would certainly not suggest that it had a similar mobility to a true ball and socket joint (e.g. the hip), but rather (based on physical manipulation) it has more lateral flexibility than D-II or D-III which, due to the ginglymoidy, are restricted to movement in a single plane. I can imagine this lateral flexibility may assist in a stabilizing function when walking, or maybe when manipulating food. It is hard to see how the current (new) analytical method failed to find any difference between MT/D-IV and MT-II and III. So I suppose on this point I disagree with the authors’ findings.

I appreciate the suggestion that their method is repeatable... in reality, repetition is generally only available to those with expertise and access to technology. That’s fine, I’m not going to reject the manuscript, it can be published as is, but I do not think that the analysis of the MT-IV D-IV-1 joint is likely to be correct (or maybe there is a nuance that I am missing since I am not a computational biomechanist). This is important as there are many curious differences among theropod clades regarding the articular surfaces of pedal joints and the shapes of the phalanges and unguals. In Fowler et al (2011) we note the stark differences between basal dromaeosaurs, derived dromaeosaurs, and troodontids (since these groups form a single clade). However, there are even more small and large differences within other clades around the origin of birds, and indeed within birds. I reject the idea that feet are just for walking (which has been a general assumption in dinosaur paleobiology), and even that walking cycles are broadly similar across taxa. Walking may be a daily function for legs, but not necessarily strongly selected for when compared with more critical and derived functions (at least in some clades). Differences in phalangeal morphology may be more indicative of diet, foraging, or other more strongly selected behaviors, and I am just cautious about the potential for narrowing of colleagues’ interpretation and expectations based on this result.

Overall the paper and method are interesting and a good piece of research. Publish as is.

We thank the Reviewer for their interest in our work, and thank them for making clear that they are happy for the paper to be published as is.

We certainly agree with Ostrom and the Reviewer's assessment of the distal articular surface of MT IV; these accounts align with our own physical examination of the YPM 5205 specimen. We agree that the morphology discussed here *does* play a role in the function of the joint, and that our analysis in fact captures this -- we direct the Reviewer to a comparison of this MTP joint with the others in the middle column of Supplementary Figure 6. The green-colored region of mid-range articulation scores is clearly much larger in the ABAD for MTP IV, resulting from the difference in morphology the Reviewer discusses. We now call attention to this interesting difference in the Supplementary Figure 6 caption.

However, we note that this expanded range is not maintained when considering only articulation scores 95-100, which we found to be the relevant range for reconstructing steady forward locomotion. Therefore, although the modified morphology of MT IV perhaps allowed a larger overall range of motion at MTP IV in life (perhaps during food manipulation as the Reviewer suggests above), our analyses suggest that a larger range of excursions would not be used during locomotion. We are optimistic that future articulation analyses involving *in vivo* studies of articulation score in non-locomotor behaviors in extant dinosaurs will be able to shed more light on the broader functional relevance of this interesting anatomy!

Reviewer #2 (Remarks to the Author):

I thank the authors for making edits to the manuscript, and the editor for inviting me to re-review this submission. I began by reading all reviewer's comments, as well as the responses to the reviewers, before re-reading the manuscript, so I could have these edits in mind. I found it helpful to better understand the edits the authors have made, and it gave me several points to look out for while reading the manuscript.

We thank the Reviewer for thoroughly re-reviewing our manuscript.

This first set of comments from the Reviewer concerns our Responses to Reviewers rather than our manuscript itself. We note that many of these comments have no connection to text or figures present in the manuscript, but nonetheless respond to each of them for clarity.

There is a point R1 made which matched some of my comments before, and as such I would like to address some of the author's response to R1, and mix it in with some comments from my review (page 2-5, 13):

I agree with reviewer 1 about taking caution in the novelty in this advance. As pointed out by R1, some of the language used in the manuscript (and response to reviewers here) is too strong. I have pointed out some aspects of this throughout my second review.

We clarify that R1 never suggested we should "[take] caution in the novelty of this advance"; their concern was with whether previous paleontological work would be disregarded as a result of our statements, stating "I caution the tone of the introduction in challenging previous work on range of motion" (R1 round 1). In response, we already took efforts to ameliorate this concern in the previous round of review by making clarifying phrasing changes in our previous submission's Lines 51-52, 64, 158. R1 expressed satisfaction with these revisions.

Page 2: this is tangential to the paper, but the earliest ROM reconstructions from skeletal material were completed decades before 2010. For example, the work on masticatory biomechanics where gape angle is estimated from skeletal collections (e.g., Bill Hylander,

Callum Ross, Chris Vinyard, Elizabeth Dumont). These studies generally investigated mandibular rotation (i.e., opening the mouth and twisting about the superior/inferior axis) and translation of the TMJ (usually in an anterior/posterior direction).

We thank the Reviewer for pointing this out; we are well aware of this and have written on this previously (for example, in the review by Manafzadeh & Gatesy, 2022 *ICB*). ROM reconstruction from bones has been going on for centuries, and quantitatively at least as early as the early 1900s, far before the masticatory examples cited here (see Manafzadeh & Gatesy, 2022 *ICB* for specific citations). The 2010 date stated in our response – it does not appear in the manuscript – was in specific reference to computational, virtual ROM reconstructions, as made clear by our existing phrasing in the response: “The earliest virtual ROM reconstructions (e.g., Mallison, 2010; Pierce et al., 2012)” (emphasis added here).

Manafzadeh, A. R., & Gatesy, S. M. (2022). Advances and challenges in paleobiological reconstructions of joint mobility. *Integrative and Comparative Biology*, 62(5), 1369-1376.

Page 4: the idea that translation and rotation of a joint must be considered together, simultaneously, is not new to (paleo)biomechanics. In addition to the mandible example (discussed above) this has also been considered at length in the human knee.

We humbly suggest that this is, contrary to the Reviewer’s statement, a very new idea to paleobiomechanics and computational ROM reconstructions. Although the mandible examples discussed above as well as studies on numerous other joints have cursorily alluded to or to some extent investigated both rotation and translation, a systematic and reproducible study algorithmically investigating combinations of rotations and translations had **never** been conducted in paleobiomechanics.

This is why we were both motivated and able to publish an entire paper on the subject two years ago (Manafzadeh & Gatesy, 2021 *J. Anat.*), highlighting the shortcomings of previous approaches that had not engaged with combinations of rotation and translation as systematically as necessary to capture biological reality. Subsequent studies considering systematic combinations of rotations and translations (e.g., Bishop et al., 2023 *Methods in Ecology and Evolution*; Wiseman et al., 2022 *IOB*) have explicitly drawn inspiration from our advance.

Bishop, P. J., Brocklehurst, R. J., & Pierce, S. E. (2023). Intelligent sampling of high-dimensional joint mobility space for analysis of articular function. *Methods in Ecology and Evolution*, 14(2), 569-582.

Manafzadeh, A. R., & Gatesy, S. M. (2021). Paleobiological reconstructions of articular function require all six degrees of freedom. *Journal of Anatomy*, 239(6), 1516-1524.

Wiseman, A. L., Demuth, O. E., Pomeroy, E., & De Groote, I. (2022). Reconstructing articular cartilage in the *Australopithecus afarensis* hip joint and the need for modeling six degrees of freedom. *Integrative Organismal Biology*, 4(1), obac031.

Page 4/5: I agree with the reviewers that a) the inclusion of a six DoF joint into biomechanical models and b) the inclusion of more complex joints into paleontology is unique (for some subfields), but agree with the reviewer that the authors do not have the data to make such authoritative statements.

The Reviewer has repeated the same concern within this round of review. As noted above, we clarify that R1’s only concern expressed in Pages 4/5 was with whether previous paleontological work would be disregarded as a result of our statements. In the previous round of review we

already clarified our phrasing in our previous submission's Lines 51-52, 64,158 to be as explicit as possible about the scope of our advance and to better prevent this unintended interpretation. We reiterate that R1 expressed satisfaction with these revisions.

Page 5: the author's express a desire to have the response to the reviewers included in their manuscripts, and I agree, such information is needed in this one. This was one of the big issues I had with this manuscript the first time I read it: I do not believe Nat Comms provides the space necessary for the authors to properly discuss the topic or methodology due to space limitations of the journal.

Appropriateness for the journal is at the discretion of the Editor, who has communicated that our manuscript is a good fit for publication in *Nature Communications*.

Page 13: I understand the author's feel I am being overly pessimistic about the generalizability of this metric, but, unfortunately, the author's have not offered sufficient evidence to the alternative, and my concerns are supported by the evidence presented in supp. Fig 9 (with regards to the glenohumeral joint). My issue was not whether or not you could use the raycasting method on other joints (of course you can!), it's about whether the results will be useful. The authors have expanded their limitations discussion in the manuscript, which is appreciated, but I believe you should have a better understanding of the limitations of a method in a manuscript which is stating, quite conclusively "this is the way forward." This speaks to the "dual nature" of the manuscript: one being the reconstruction of the pedal pose of an extinct animal, and the other presenting the new method. It is, essentially, two studies wrapped into one, and being presented in a short communication.

We are gratified to learn that the Reviewer does understand that our raycasting approach can easily be applied to other joints. As stated in our Discussion since our original submission:

"comparing articulation score distributions against *in vivo* kinematics for further joints and taxa will illuminate the broader utility of articulation analysis, facilitating the widespread application of these data in vertebrate functional reconstruction" (previous submission Lines 172-174).

In stating this, we have made clear to the reader that the only way to determine "whether the results will be useful" (Reviewer's phrasing) is through comparisons with *in vivo* data that are specific to the research question at hand.

We cannot possibly test the full diversity of vertebrate joints in one manuscript, and neither we nor the Reviewer can evaluate how "useful" articulation analysis will prove for the reconstruction of any particular behavior at any particular joint without conducting such tests. However, our paper demonstrates the potential great utility of this approach using our selected case study, while also laying the necessary groundwork in conceptual and methodological advances for its future expansion.

To the Reviewer's concern about the glenohumeral joint in Supplementary Figure 9, then, we emphasize that the exemplar configurations we have displayed will successfully receive different articulation scores – we have now added quantitative articulation subscores to the Supplementary Figure 9 caption to make this as clear as possible – and determining how "useful" these differences in score are will rely on future collection of and comparison with *in vivo* shoulder kinematics.

Dedicated analysis of the human glenohumeral joint is far beyond the intended scope of this manuscript and this example was added to assuage this Reviewer's concerns in the previous round about the broader generalizability of this approach, an addition which is hopefully also of interest to the journal's broad readership. As of interest to the Reviewer, we offer the following screenshots of extremes of *in vivo* glenohumeral rotation kindly provided by an orthopedic biomechanics colleague to emphasize that in a wide range of rotational poses, joint translation is limited such that articular congruence (informally assessed here based on apparent matching of joint surface curvature) appears to stay much higher than in our low-scoring 6 dof configuration figured in Supplementary Figure 9. This would tentatively suggest that raycasting analysis of this joint will in fact have future value for functional reconstruction of this joint, though we emphasize the need for thorough, formal articulation analysis and comparison with *in vivo* kinematics. If the Reviewer has a particular interest in the human glenohumeral joint, we hope they will implement our new approach to investigate shoulder articulation in more detail!

To concern about the “dual nature” of our manuscript: the Reviewer has repeated the same concern within this round of review. We again, as above, defer to the judgment of the Editor, who has communicated that our manuscript as written is a good fit for *Nature Communications*.

Page 9: I thank the authors for their clarification, there was indeed a miscommunication. Pose certainly does need to be defined, and early on in the manuscript (but see comments on this

manuscript below). In its current form, joint poses (line 51), is now defined as rotational combinations of joint articulations, which I do not believe is what the author's mean the definition to mean. I believe a proper definition would be something closer to "position of body segments, relative to each other, that are connected by a joint". This is closer to the definition given in L40-41 for a different term (joint configurations). As an aside, the authors also misunderstood my comment, as I was not suggesting in my comment that motion in animals was deterministic (and neither is motion in robots, not the least bit because of the inclusion of AI and machine learning, but also because of things like vibrations and inertial momentum, making precision manufacturing by robots difficult/impossible today 😊)

For the title, I would recommend changing "locomotor joint poses" to "joint poses during locomotion" for clarity.

The Reviewer is repeating a concern from the previous round of review. We clarify that we have defined joint poses as combinations of excursions in all three rotational degrees of freedom (not "rotational combinations of joint articulations;" we are unsure what combinations of articulations would mean and therefore do not use this language). Likewise, we have defined joint configurations as combinations of excursions in all six (all three rotational and all three translational) degrees of freedom. These definitions were added explicitly to the manuscript in last submission's Lines 40-41 and Line 51 during the last round of review to increase clarity. We re-emphasize as discussed in the previous round of review that there is no "proper definition" for joint poses, and that different fields use the word pose differently – we offered several citations to this effect in the previous round of review, and repeat that response here:

the word "pose" has been used in the orthopedic, anthropological, paleontological, zoological, computer animation, and biomechanical literature to refer to everything from a joint's excursion in a single rotational degree of freedom (e.g., Senter, 2009), to a joint's excursions in all three rotational degrees of freedom (e.g., Manafzadeh & Padian, 2018; this paper), to a joint's excursions in all six degrees of freedom (e.g., Bishop et al., 2023) to the combination of joint "poses" (defined as any of the previous) throughout a limb (e.g., Lauer et al., 2022) or even an entire individual/character.

Bishop, P. J., Brocklehurst, R. J., & Pierce, S. E. (2023). Intelligent sampling of high-dimensional joint mobility space for analysis of articular function. *Methods in Ecology and Evolution*, 14(2), 569-582.
Lauer, J., Zhou, M., Ye, S., Menegas, W., Schneider, S., Nath, T., ... & Mathis, A. (2022). Multi-animal pose estimation, identification and tracking with DeepLabCut. *Nature Methods*, 19(4), 496-504.
Manafzadeh, A. R., & Padian, K. (2018). ROM mapping of ligamentous constraints on avian hip mobility: implications for extinct ornithomirans. *Proceedings of the Royal Society B: Biological Sciences*, 285(1879), 20180727.
Senter, P. (2009). Pedal function in deinonychosaurs (Dinosauria: Theropoda): a comparative study. *Bulletin of the Gunma Museum of Natural History*, 13, 1-14.

Therefore, other readers would certainly disagree with both us and this Reviewer about their intuition for the "proper definition" of "pose," and it is most important for clarity and accessibility of our work that we are explicit about what we mean by this word for this specific manuscript. Again, this is a change we already implemented in the previous round.

We already changed the title of our submission to "...locomotor joint poses" in the last round of review in response to this Reviewer's concerns that "...locomotor poses" did not clearly enough reflect that our definition of pose was on a per-joint basis. The title the Reviewer now suggests is fully synonymous with our current title and purely a difference of semantic preference, so we respectfully decline the suggested change. Instead, to increase clarity as fully as possible, we have added a parenthetical to the Introduction in new Line 66 (joint poses actually used during terrestrial locomotion (i.e., locomotor joint poses)), explicitly drawing the connection the Reviewer requests, and have changed two instances of "locomotor pose" to "locomotor joint pose" (new Lines 85 and 89) in the Results for better consistency with the existing title.

Page 11: In response to there being no other methods present, It does not need to be another 6 DoF method, it just needs to be another method for estimating joint poses, and the authors have actually discussed an alternative methodology previously in this review (i.e., the “prism method” for estimated ROM). While imperfect, it would be useful to see how results from that method, or the “qualitative method” of joint poses discussed later in the response to R1, compared to results from this method. If results are similar in terms of locomotor reconstruction (forward dynamic simulations), then does this more complicated ray casting method of estimation actually need to be used? If not, it creates a good argument for using this model (which has its own inaccuracies) over other models. I stand by my earlier comment that a comparison is needed to provide tangible evidence for how this method is superior and provides superior estimates for extinct animals.

The Reviewer is repeating a concern from the previous round of review. The Reviewer, in the previous round, stated “But the paper neither presented a method for evaluating joint articulation or compared this method to others in terms of reconstructions (showing that this method is more accurate),” so their desired comparison was of methods for “evaluating joint articulation.” We reiterate, as stated in our previous round of responses, that there are no other reproducible methods in existence to accomplish the comparison that the Reviewer requests.

The Reviewer now raises suggestions for two potential “alternative methodolog[ies].”

First, they bring up the “prism method.” We invented the prism method two years ago (Manafzadeh & Gatesy, 2021 *J Anat*). The prism method in no way evaluates articulation. It is simply a method for sampling joint translations in hinge joints. In fact, we already implement the prism method in the present study when sampling 6 dof joint configurations, and then we evaluate articulation at each viable (non-interpenetrating) joint configuration using our new raycast-based approach. This has been explicit since the initial submission in the Methods section, where we state:

“At each rotational pose, 343 potential translation combinations were allowed (7×7×7 translations sampled using the prism-based hinge joint translation method proposed by [26])” (new Lines 245-247)

and

“We then conducted an articular raycast at each viable (i.e., non-interpenetrating) configuration.” (new Lines 256-257).

Manafzadeh, A. R., & Gatesy, S. M. (2021). Paleobiological reconstructions of articular function require all six degrees of freedom. Journal of Anatomy, 239(6), 1516-1524.

Second, they bring up the “qualitative method ... discussed in the response to R1.” The qualitative method of evaluating joint articulation is just that – qualitative – and is therefore subjective based on the interpretations of the specific researcher implementing the method. That said, we already include the results of this method as applied to the foot of *Deinonychus* as Supplementary Figure 8; we added this in the last round of review in response to a different comment by this same Reviewer. It appears the Reviewer may have missed this addition. We reiterate our response from the last round:

The most quantitative hypothesis of Deinonychus joint motion to date was published by Senter (2009), who relied on manual manipulation and subjective assessment of proper articulation in an attempt to identify each joint's full ROM (critically, not its locomotor ROM, which is a subset of full ROM [see Manafzadeh et al., 2021]). Notably, Senter's hypotheses sometimes underestimate even the locomotor ROM we recover here! To make more explicit in the paper that we are contributing new data to our understanding of Deinonychus, we have created new Supplementary Figure 8 to directly compare our results with Senter's.

Therefore, we posit that the raycast-based approach we use here is not “more complicated,” but that it is the only objective method currently available to the field for quantitatively evaluating joint articulation. We cannot claim that our method is “superior,” nor do we intend to – we intend to provide the **first and only** method for accomplishing this goal. We are indeed hopeful that this will serve as the foundation for the future development of methods superior to ours, such that future comparison is possible!

If higher articulation scores do not represent stability, what do they biologically represent? This needs to be explicitly stated in the manuscript, to prevent others from making the same mistake I have. A definition of articulation early on would clear up some of this confusion, as the authors are using terms like “disarticulation”, “misalignment”, and “subluxation,” which compromise the stability of the joint, to explain improper articulation, but are also stating that articulation scores (and thereby, articulation) do not represent stability.

The Reviewer is repeating a concern from the previous round of review. In the previous round, the Reviewer stated:

“The reason I say the paper did not present a method for evaluating joint articulation is that it provides a method for quantifying the likelihood of a joint engaging in a pose based on a measure of joint stability, but this does not quantify articulation. Nowhere in this metric can you determine if a joint is articulated or not, just if a joint is likely to engage in a pose during walking/running in a straight line.”

In response to this miscommunication of our intent, we already made the goal of our articulation analysis (“measuring the quality of joint articulation for any viable, static six-degree-of-freedom joint configuration”) more explicit in Lines 60-62 of the previous submission. Also in response, we also already clarified in our Methods (previous submission Lines 257-264) that:

“The articular raycasting approach we propose here aims to capture information about the morphological relationship of a pair of mating articular surfaces in any viable (i.e., non-interpenetrating), static six-degree-of-freedom joint configuration. Conceptually, our approach provides data about the relationship between these surfaces’ 3-D curvatures. We select this emphasis here because interactions between articular surface curvatures are fundamental to joint function (see [36, 64]). The relationship between the curvatures of mating articular surfaces in any given configuration dictates the paths of minimum work along which joints habitually move, as well as the capacity of a joint to effectively distribute and evenly transmit loads.”

And later in lines 280-287 that:

“We propose that the formation of a successful ray hit means that articular curvatures are aligned such that there is a capacity for meaningful biomechanical interaction between the pair of articular surfaces in vivo. Therefore, if even a single ray hit successfully, we assigned the joint configuration an articulation score of greater than zero (see formula, below). If no rays succeeded in hitting the mating articular surface, we considered the joint to be unscorable, and gave the joint configuration an articulation score of zero. Articulation score was then calculated using three

parameters – overlap, symmetry, and congruence – each of which receives an individual subscore from 0 to 1.”

These sections of text already explicitly address our understanding of the biological meaning of joint articulation.

We disagree that “disarticulation,” “misalignment,” and “subluxation,” necessarily communicate something about joint stability. These words, like “pose,” are highly fraught, having been taken to mean very different things in different papers across different fields. It is possible the Reviewer has encountered them largely in the context of stability, leading to their impression, but in the absence of any citations provided by them we cannot be sure. We reiterate that the word “stability” has never appeared, and still never appears, in our manuscript. We also already acknowledged in the previous round of responses that:

*“we do think the Reviewer’s general intuition is correct, and it is reasonable that joint configurations with a high articulation score may be those that are more stable. **However, because such a claim would need to be tested experimentally and because making such a claim is not our goal, we avoid using these terms.**” (Emphasis added here.)*

While the authors state that the articulation metric is not probabilistic, they are treating it like a probability in their analysis. I.e., scores above 95 were used further means the probability that scores less than 95 would be engaged in is low. They also discuss it in a probabilistic manner in the manuscript (see comments about current manuscript below). They are not using it as a 95% CI, but they are using it as a probability (i.e., the probability that the joint will be in that pose, due to an articulation score which has been validated with a subset of joints from two species)

The Reviewer is repeating a concern from the previous round of review. Again, we have already addressed this concern in the previous round:

“we clarify that this metric is not probabilistic, nor does it attempt to highlight “likely” poses. Articulation score can be calculated for a pair of bony articular surfaces in any viable, static six-degree-of-freedom configuration. What we find here is that the joint poses used during locomotion happen to consistently fall in the highest-scoring region of pose space -- but we find this using experimental data; the score itself contains no measure of probability and is not inherently tied to locomotion in any way. We again hope the new Supplementary Text and Figures we have added will make this more clear.”

We are certainly identifying a consistent correlation in extant animals and applying it to new data from extinct animals within a phylogenetically and morphologically informed framework. Perhaps we differ with the Reviewer in terms of our understanding of the meaning of “probabilistic,” because if this general approach is viewed as probabilistic, we would argue that all application of data from extant animals to the fossil record is similarly inherently probabilistic – this is the foundation of modern functional morphology in paleontology. Regardless, the Reviewer does not suggest here that they view this approach as problematic, even if “probabilistic,” nor do they propose a constructive path forward for revision.

Page 12: The authors write “The score simply offers a quantifier of the quality of joint articulation for any viable, static six-degree-of-freedom configuration of mating bony articular surfaces – not “joint ROM.” The authors are, in fact, quantifying joint range of motion based on their articulation scores: high articulation scores = inclusion in ROM, low articulation scores = exclusion. In their study, ROM is defined as having an articulation score of 95+.

The Reviewer is repeating a concern from the previous round of review. We reiterate our previous statement that we are not quantifying range of motion based on our scores – the Reviewer has kindly quoted that previous statement for us. Our score offers information about the quality of joint articulation across all of joint pose space. We have identified that locomotor poses consistently fall within the subset of pose space with articulation scores of 95-100. The score itself does not quantify range of motion. The subset of pose space identified through articulation score does not reflect the joint's full range of motion, but rather a locomotor subset of range of motion.

This initially came up because the Reviewer stated in the previous round of review that “What I believe the paper is about is a method for quantifying joint ROM based on a measure of stability.” We have already clarified in several ways why this belief is incorrect – we have reiterated our disagreements about both “quantifying joint ROM” and “a measure of stability,” already discussed once in the last round of review, here and above.

With regards to the whole-foot combinations of joint poses, part of the reason I used the words “opinion” and “random” is due to the writing of the author’s on L290 (L362 in the resubmission). The authors write “An inverse kinematic animation rig was used to animate a rough stride cycle for the pes of Deinonychus using inspiration from published extant avian toe tip motion” To me, this sentence comes across as the author’s constructed a model of extant avian toe tip motion, and then – combined with the joint poses – used it as inspiration to create the combined joint poses presented in the paper. In this case, inspiration would be similar to an artist using a set of trees in a forest as inspiration to create one, composite tree. In this case, the composite tree is influenced by the author’s opinion heavily, with no necessary protocol (i.e., random) as to why some features were chosen, and others were not. If this is what the author’s did, then my comment stands true. If not, then this again speaks towards not enough detail being given in the paper for the study to be properly explained or replicated.

The Reviewer is repeating a concern from the previous round of review. They include our text that “An inverse kinematic animation rig was used to animate a rough stride cycle for the pes of Deinonychus using inspiration from published extant avian toe tip motion.” We pointed out in the previous round of review that this is common practice in the field of paleontology, stating in the last round that:

Here the hypothesis of covariation we chose to employ, as stated in Line 362, is based on that of avian feet. This is one reasonable hypothesis for non-avian dinosaur intra-pes coordination and is regularly applied by dinosaur paleontologists (e.g., Falkingham & Gatesy, 2014; Gatesy et al., 1999).

We have now edited our text in new Lines 361-365 to include reference to Falkingham & Gatesy (new reference 70) and to make more reproducible the specific way in which this inspiration was implemented in the current study. We reiterate that such practice is regularly applied in paleontology and that the goal of the present study was to reconstruct one of many possible well-supported stride cycles for *Deinonychus*, reminding the Reviewer as we discussed extensively in the previous round that natural variation is to be expected in the joint poses and configurations used in locomotor stride cycles. The edited text is as follows:

An inverse kinematic animation rig was used to animate a rough stride cycle for the pes of Deinonychus using inspiration from published extant avian toe tip motion by importing published videos and reconstructed animations of avian toe tip motion into Maya and aligning the

metatarsus and distal phalanges of *Deinonychus* to the avian metatarsus and distal phalanges [69-70].

We also re-emphasize to the Reviewer that this inspiration was used simply to generate a starting point, first-pass “rough stride cycle” that was then modified into alignment with our articulation data, as explicitly described in the remainder of the Methods section.

Page 13: as stated above, the authors have misunderstood my comment. It is not whether this method can be applied to all joints, but rather if the results would be useful or not. From supplementary Figure 9, it certainly seems it would be less than useful for the human glenohumeral joint, as it appears some positions that are subluxations would come across as having high articulation scores, which is a fundamental shortcoming of this method.

The Reviewer has repeated the same concern within this round of review. We have already responded to this concern extensively above, but restate here that the exemplar configurations we have displayed will successfully receive different articulation scores – we have now added quantitative articulation subscores to the Supplementary Figure 9 caption to make this as clear as possible – and determining how “useful” these differences in score are will rely on future collection of and comparison with *in vivo* kinematics.

Page 14: Thank you for clearing up explanations in the review – what the author’s are trying to accomplish is much clearer in the response to the reviewers. My (simplified) understanding of ray casting is now as follows

- The more rays that connect one articular surface to the other, the more symmetric ray lengths are... etc... the more “articulated” the joint is, and therefore the higher the articulation score
- Higher articulation scores are correlated with joint poses in extant animals, and therefore are correlated to the way the body segments move relative to each other.

We are gratified to see that the Reviewer now has a better understanding of raycasting. All information included about raycasting in the previous round of review was also incorporated into the manuscript’s Methods section through substantial revision, making it available to all readers.

Now, this implies one of two things

- 1) this is a random correlation, in which case it is biologically meaningless
- 2) this is a causative relationship, whereby articulation scores are related to joint motion (rotation and translation)

The authors believe this is a causative relationship (e.g., see wording in abstract, discussed below), which is why they can use it to reconstruct extinct vertebrate locomotion.

Assuming this is true, my question from before still stands: what is the underlying logic and biomechanical meaning of ray casting? I.e., why, functionally, was ray casting work? It is touched on in the paper, but not explained. E.g., why does asymmetry in ray length matter? Why does having more rays connecting the two surfaces when there is space between them biomechanically matter? Otherwise it appears to be more-or-less arbitrary why ray casting was chosen, and why it works in this paper.

The Reviewer has repeated the same concern within this round of review. We reiterate, as above, that we already implemented substantial revisions during the previous round in response to this Reviewer’s concerns about this. Again, in our revised Methods (previous submission Lines 257-264) we stated that:

*“The articular raycasting approach we propose here aims to capture information about the morphological relationship of a pair of mating articular surfaces in any viable (i.e., non-interpenetrating), static six-degree-of-freedom joint configuration. **Conceptually, our approach provides data about the relationship between these surfaces’ 3-D curvatures. We select this emphasis here because interactions between articular surface curvatures are fundamental to joint function (see [36, 64]). The relationship between the curvatures of mating articular surfaces in any given configuration dictates the paths of minimum work along which joints habitually move, as well as the capacity of a joint to effectively distribute and evenly transmit loads.**” (emphasis added here)*

And later in lines 280-287 that:

***“We propose that the formation of a successful ray hit means that articular curvatures are aligned such that there is a capacity for meaningful biomechanical interaction between the pair of articular surfaces in vivo.** Therefore, if even a single ray hit successfully, we assigned the joint configuration an articulation score of greater than zero (see formula, below). If no rays succeeded in hitting the mating articular surface, we considered the joint to be unscorable, and gave the joint configuration an articulation score of zero. Articulation score was then calculated using three parameters – overlap, symmetry, and congruence – each of which receives an individual subscore from 0 to 1.” (emphasis added here)*

These sections of text already discuss our explicit justification for selecting raycasting, a choice which was not arbitrary, and explain its biomechanical meaning.

Page 16: If the point of this method is to create an envelope of possible joint poses, the authors do not need to identify what percentage of sampled poses fall within the 95-100 articulation score envelope, but rather find the articulation score envelope that fits 100% of the joint poses. Otherwise, the authors are contradicting themselves (or believe 1.9% of the in vivo data is incorrect). I stand by my comment that articulation scores from 95-100 should not be used, but rather this window needs to be calculated from the in vivo data, and then used to re-reconstruct extinct animal locomotion.

We note that we calculated and included the percentage of measured *in vivo* locomotor poses that fall within the 95-100 envelope in direct response to a request from this Reviewer. We think this is a useful addition that has improved our manuscript and thank them for the motivation.

The Reviewer is repeating a concern from the previous round of review. A goal of this study was to reconstruct one of many possible locomotor stride cycles for the extinct animal *Deinonychus* using support from joint articulation data. Because **ALL avian walking data fell within the 95-100 envelope**, we maintain our judgment that it is reasonable to use this range to reconstruct **one of many possible walking stride cycles for *Deinonychus***, even if a small number of the poses used during swing phase of highest-speed avian running were found to exit the the 95-100 score range.

Page 17: “We contend that it would not change the reconstruction, because these poses would not be ruled out even if the envelope were expanded as the Reviewer suggests.” The authors have misunderstood my comment. Of course, the poses would not be ruled out, but new poses that were ruled out before could now be chosen.

The Reviewer has repeated the same concern within this round of review. As described directly above, the stride cycle we reconstruct here would not be ruled out, which is what is relevant for the goal of this study.

Page 17: I misunderstood the previous read of the manuscript, and had thought the authors had run forward dynamic simulations, using the joint envelopes as boundary conditions. It is now my understanding the authors used the extant data to roughly position the bones of the extinct animal in the correct position, then manually altered joint positions so they fit within this envelope of possible positions. If this is correct, there is certainly a degree of artistic flair in the reconstruction that needs to be discussed, and the sensitivity of these reconstructions (inter- and intra-observer error) needs to be looked at, given this method is being advertised as a “new way forward” (my words, not the authors) for reconstructing joint poses and locomotor poses of extinct animals.

The Reviewer has repeated the same concern within this round of review. We addressed this above, and as stated, edited our text in new Lines 361-365 to include reference to Falkingham & Gatesy (new reference 70) and to make more reproducible the specific way in which this inspiration was implemented in the current study. As discussed in the two responses immediately above this one, our goal was to reconstruct one of many possible walking stride cycles for *Deinonychus* using support from joint articulation data. The aspect of our approach that we suggest is a “new way forward” (we will agree with the Reviewer’s words) is the ability to test a reconstructed stride – no matter how it is generated – using data from joint articulation. Our joint articulation results have, of course, been heavily tested using the sensitivity analyses presented in the manuscript, because that is the extent of the advance we are proposing.

Page 17: “However, the final sentence of this comment...” the authors have misread my comment. What the authors have “ground-truthed” was whether they could put joints in realistic poses. What I was saying what you cannot say your method has led to a more accurate reconstruction of the gait of the extinct animals because 1) it was not compared to results obtained using traditional joint constraints, and 2) it was not compared to how this extinct animal actually moved. If you wanted to use extant models for this purpose, you would have to take the data from one of the extant animals and do a similar reconstruction that you did with the extinct animal (potentially with bones from another individual of that extant species – e.g., another guinea fowl or emu) using both traditional joint constraints and these new constraints (estimated using ray casting) and see which one produces results closer to your experimental data. Again, if both produce similar results, the traditional method is easier and takes less time, so why should we bother with raycasting?

The Reviewer has repeated the same concern within this round of review. We extensively responded to the Reviewer’s concerns about comparison of our method above; this comment appears to be a reiteration of the same concern. We repeat the final portion of our response here:

“we posit that the raycast-based approach we use here is not “more complicated,” but that it is the only objective method currently available to the field for quantitatively evaluating joint articulation. We cannot claim that our method is “superior,” nor do we intend to – we intend to provide the **first and only** method for accomplishing this goal. We are indeed hopeful that this will serve as the foundation for the future development of methods superior to ours, such that future comparison is possible!”

Resubmission

This next set of comments from the Reviewer engages with our manuscript itself, reiterating many of the concerns we have addressed above.

Abstract: "... locomotor joint poses consistently have high articulation scores. We then exploited this predictive relationship to constrain reconstruction of a pedal stride cycle..." These lines state that it is possible to predict joint poses based on articulation scores, but the authors failed to show (or statistically test) any type of predictive relationship between articulation score and pedal stride cycle in their extant individuals. That is to say, they showed pedal stride cycles were encompassed in the polygonal space created by articulation scores (you can predict the number of legs a zebra has), but never that articulation scores can predict pedal stride cycle (you can predict an species by knowing the organism has four legs).

The Reviewer has repeated the same concern within this round of review. We have already responded to the Reviewer's concerns about probability above. We reiterate that the kind of reasoning the Reviewer appears to take issue with is the basis of all application of data from extant animals to the fossil record. We engage in this application here as responsibly as possible, within a phylogenetically and morphologically informed framework. The approach we take is standard to the field of paleontology.

L33: I must have missed it the last time – a joint is not an organ

Many sources would disagree. The *Biology of the Synovial Joint* book cited in that sentence states that "This unique collection of reviews has arisen due to the belief of the Editors that joints need to be studied as a whole organ." We align with this viewpoint and would similarly argue that a joint is an organ. This point has no bearing on the paper's data or conclusions.

L40-41: this definition of joint configurations (the excursion of all three rotational and translational DoF) are what the authors stated was "poses" in the response to the reviewers.

We cannot identify the claimed location in the response to reviewers. Our definitions of poses and configurations have, to our knowledge, stayed consistent, and are certainly consistent within the manuscript. Poses = 3 rotational dof, configurations = 3 rotational and 3 translational dof.

L48-49: much related to gape in primates has been ground-truthed, so this statement is incorrect

We are certain that the Reviewer's statement is generally true, however, to our knowledge and despite substantial searching, we suggest that it is false in the context of what is discussed in lines 48-49 (joint articulation criteria for virtual ROM analyses). Metrics such as "maximum bony gape," while very interesting, have not involved analysis of articulation between joint surfaces in living animals, which is how we define ground-truthing articulation criteria. This task would require the use of XROMM-derived data or rigid body marker tracking. Given that one of the co-developers of XROMM is an author on this paper, we question whether such analyses have been conducted. Published studies could always exist that we are unaware of, but the Reviewer did not provide specific citations.

L81-82: In their response, the authors stressed that they were not discussing the "probability" of a joint ending up in a pose, but the writing here ("... mapping our articulation scores onto this mobility estimate enhances our knowledge of each pose beyond a binary "possible" or "impossible"...") implies they are bringing joint articulation out of a deterministic, and into a probabilistic space. The authors are contradicting themselves.

The Reviewer has repeated the same concern within this round of review. An “[enhanced] knowledge” is just that -- increased data. There is no allusion to probability, and we suspect that the relationship between articulation score and in vivo pose utilization is highly non-linear. We have already responded extensively above (and in the previous round of review) to the reviewer’s concerns about probability.

L86-90: This is a major issue with the 95-100 window, as it does not encompass all possible ranges of motion. As these envelopes of possible joint poses are being used as boundary constraints for models of locomotion for extinct animals, they must encompass the entire possible ROM for the extant animals, otherwise they are poorly defined constraints.

The Reviewer has repeated the same concern within this round of review. We have already responded to this concern above. As stated, our goal was to reconstruct one of many possible walking stride cycles for *Deinonychus* using support from joint articulation data.

We also point out that experimental data have revealed that locomotor poses occupy a small subset of a joint’s full possible ROM (see Manafzadeh et al., 2021 *PNAS*). Therefore, contrary to the Reviewer’s statement that “envelopes... must encompass the entire possible ROM for the extant animals, otherwise they are poorly defined constraints,” we would argue that identifying the entire possible ROM instead of what we have done here would be a very poor constraint.

Manafzadeh, A. R., Kambic, R. E., & Gatesy, S. M. (2021). A new role for joint mobility in reconstructing vertebrate locomotor evolution. Proceedings of the National Academy of Sciences, 118(7), e2023513118.

L95-98: Text needs to be added to the supplementary material explaining the sensitivity analyses. As it stands, it is not clear what sensitivity analysis was done from the figures and captions alone. Results from the sensitivity analysis should be explained as well.

The methods already describe what was done, stating in current Lines 334-339 that:

“Translational sensitivity analysis was performed for guineafowl individual 1 by allowing 1,331 translational combinations over a larger range (two additional increments of equal size at each end of the X, Y, and Z translation ranges) at five-degree angular resolution, and sensitivity to articulation score formula was evaluated by conducting an analysis for this individual at the original rotational and translational resolution, weighting congruence by single-condyle overlap rather than full-joint overlap.”

The results already summarize the results, stating in current Lines 95-98 that:

“We found that our articulation score distributions are robust to sensitivity analyses conducted using additional individuals, increased translational allowance, and alternative score formulas (Supplementary Fig. 3-5).”

The captions of Supplementary Figures 3-5 offer both sets of information with additional detail, already stating that:

Supplementary Figure 3: “Results from the guineafowl individual figured in the main text, displayed at five-degree angular resolution in (a) and one-degree angular resolution in (d); a second individual, displayed at five-degree angular resolution in (b) and one-degree angular resolution in (e); and a third individual, displayed at five-degree angular resolution in (c) and one-

degree angular resolution in (f), **are grossly similar, especially within the highest-scoring region.**” (emphasis added here)

Supplementary Figure 4: “Results from both the translational allowance throughout this paper, displayed at five-degree angular resolution in (a) and (b) and one-degree angular resolution in (c), and an increased translational allowance (see Methods), displayed at five-degree angular resolution in (d) and (e) and one-degree angular resolution in (f), demonstrate that **although increasing translational allowance increases the region of pose space coded as viable (and therefore the region of pose space receiving articulation scores), the highest-scoring region of pose space remains identical.**” (emphasis added here)

Supplementary Figure 5: “Results from both the articulation score formula implemented throughout this paper, displayed at five-degree angular resolution in (a) and one-degree angular resolution in (c), and an alternative formula that instead weights each condyle’s congruence by only its own overlap (rather than average overlap; see Methods), displayed at five-degree angular resolution in (b) and one-degree angular resolution in (d), **are grossly similar.**”

In the absence of more specific requests from the Reviewer, we are not sure what additional detail would be helpful. The details already provided allow full reproduction of the sensitivity analyses and offer our interpretation of their results.

L140-143: It was not the findings in this paper that enabled the reconstruction of the stride cycle, but “using the findings of this paper as constraints for XXX simulations...”

We suggest this is semantic disagreement and respectfully decline the suggestion.

I very much appreciate the expanded limitations discussion, and the expanded materials and methods. The additional information on the method has helped immensely.

We thank the Reviewer for their time and for their recognition of our substantial effort to improve our manuscript in the previous round of review.

Reviewer #3 (Remarks to the Author):

I would like to thank the authors for thoroughly addressing my comments and implementing suggestions. From my perspective all points have been addressed and the updated manuscript has been greatly improved. I would consequently recommend publication now.

We thank Reviewer 3 for taking the time to re-review our manuscript, and for recommending publication at this time.

Reviewers' Comments:

Reviewer #4:

Remarks to the Author:

Thank you for the opportunity to contribute to the peer review process of this interesting paper, proposing a novel method to improve objectivity in our field. Where there is scope for progress and improvement, there will also likely be lively debate. I hope that my opinion is useful to all parties involved.

I appreciate that I have been asked primarily as a referee/additional opinion on the discussion between Reviewer 2 (R2) and the authors, to assess how the concerns of R2 have been addressed, rather than review the manuscript. However, the Editor notes that feedback on the manuscript more generally is welcomed. I thought it sensible to read the manuscript and record my own thoughts independently before diving into the R2/author discussion, for the purposes of objectivity. I do not intend for these comments to contribute further to the author workload (as I understand the manuscript has already been seen by three prior reviewers), and I understand if my comments are taken only as context to the R2/author discussion rather than additional actionable tasks.

I have formatted my opinion as three sections. First, specific point-by-point comments on the 26 points of concern in the R2/author discussion. Then my own comments on the manuscript. Finally, since the resulting doc is quite long and has a fair amount of repetition, a summary of main points.

I have uploaded the comments as an attachment due to length.

Opinion on Response to Reviewers (Round 2)

In the first part of the Response discussion, R2 and the authors refer to the previous rounds of review (both of R2 and of other reviewers). I cannot see these previous rounds, and nor do I think I need to, however I note that I may be missing the full context of these discussions. I have done my best to express an opinion, but apologise if I have misunderstood any of the earlier context. I have duplicated the Response discussion and added my comments in yellow text for clarity.

Point 1:

R2:

There is a point R1 made which matched some of my comments before, and as such I would like to address some of the author's response to R1, and mix it in with some comments from my review (page 2-5, 13): I agree with reviewer 1 about taking caution in the novelty in this advance. As pointed out by R1, some of the language used in the manuscript (and response to reviewers here) is too strong. I have pointed out some aspects of this throughout my second review.

Authors:

We clarify that R1 never suggested we should “[take] caution in the novelty of this advance”; their concern was with whether previous paleontological work would be disregarded as a result of our statements, stating “I caution the tone of the introduction in challenging previous work on range of motion” (R1 round 1). In response, we already took efforts to ameliorate this concern in the previous round of review by making clarifying phrasing changes in our previous submission's Lines 51-52, 64, 158. R1 expressed satisfaction with these revisions.”

R4:

These comments refer to R1's previous round of comments. If R1 is now satisfied, I think this matter should be considered closed – unless, R2 expresses specific concerns re: the revised manuscript that has been submitted.

Point 2:

R2:

Page 2: this is tangential to the paper, but the earliest ROM reconstructions from skeletal material were completed decades before 2010. For example, the work on masticatory biomechanics where gape angle is estimated from skeletal collections (e.g., Bill Hylander, Callum Ross, Chris Vinyard, Elizabeth Dumont). These studies generally investigated mandibular rotation (i.e., opening the mouth and twisting about the superior/inferior axis) and translation of the TMJ (usually in an anterior/posterior direction).

Authors:

We thank the Reviewer for pointing this out; we are well aware of this and have written on this previously (for example, in the review by Manafzadeh & Gatesy, 2022 ICB). ROM reconstruction from bones has been going on for centuries, and quantitatively at least as early as the early 1900s, far before the masticatory examples cited here (see Manafzadeh & Gatesy, 2022 ICB for specific citations). The 2010 date stated in our response – it does not appear in the manuscript – was in specific reference to computational, virtual ROM reconstructions, as made clear by our existing phrasing in the response: “The earliest virtual ROM reconstructions (e.g., Mallison, 2010; Pierce et al., 2012)” (emphasis added here).

Manafzadeh, A. R., & Gatesy, S. M. (2022). Advances and challenges in paleobiological reconstructions of joint mobility. *Integrative and Comparative Biology*, 62(5), 1369-1376.

R4:

This point is acknowledged as tangential to the paper, and not part of the manuscript. I suggest that this matter should be considered closed.

Point 3:

R2:

Page 4: the idea that translation and rotation of a joint must be considered together, simultaneously, is not new to (paleo)biomechanics. In addition to the mandible example (discussed above) this has also been considered at length in the human knee.

Authors:

We humbly suggest that this is, contrary to the Reviewer’s statement, a very new idea to paleobiomechanics and computational ROM reconstructions. Although the mandible examples discussed above as well as studies on numerous other joints have cursorily alluded to or to some extent investigated both rotation and translation, a systematic and reproducible study algorithmically investigating combinations of rotations and translations had never been conducted in paleobiomechanics.

This is why we were both motivated and able to publish an entire paper on the subject two years ago (Manafzadeh & Gatesy, 2021 *J. Anat.*), highlighting the shortcomings of previous approaches that had not engaged with combinations of rotation and translation as systematically as necessary to capture biological reality. Subsequent studies considering systematic combinations of rotations and translations (e.g., Bishop et al., 2023 *Methods in Ecology and Evolution*; Wiseman et al., 2022 *IOB*) have explicitly drawn inspiration from our advance.

Bishop, P. J., Brocklehurst, R. J., & Pierce, S. E. (2023). Intelligent sampling of high-dimensional jointmobility space for analysis of articular function. *Methods in Ecology and Evolution*, 14(2), 569-582.

Manafzadeh, A. R., & Gatesy, S. M. (2021). Paleobiological reconstructions of articular function require all six degrees of freedom. *Journal of Anatomy*, 239(6), 1516-1524.

Wiseman, A. L., Demuth, O. E., Pomeroy, E., & De Groote, I. (2022). Reconstructing articular cartilage in the *Australopithecus afarensis* hip joint and the need for modeling six degrees of freedom. *Integrative Organismal Biology*, 4(1), obac031.

R4:

I agree with the authors that translation and simultaneous rotations have, at the very least, traditionally been underused in estimating ROM of extinct animals. Until recently (and including studies that continue to be published), it seems usual to explore single rotational degrees of freedom at a time. Translation may or may not be explored. Because this has been the norm in our field, I think emphasising the idea that translations and rotations need to be considered together in a reproducible and objective way is defensible, even if it is not a new concept.

Point 4:

R2:

Page 4/5: I agree with the reviewers that a) the inclusion of a six DoF joint into biomechanical models and b) the inclusion of more complex joints into paleontology is unique (for some subfields), but agree with the reviewer that the authors do not have the data to make such authoritative statements.

Authors:

The Reviewer has repeated the same concern within this round of review. As noted above, we clarify that R1's only concern expressed in Pages 4/5 was with whether previous paleontological work would be disregarded as a result of our statements. In the previous round of review we already clarified our phrasing in our previous submission's Lines 51-52, 64,158 to be as explicit as possible about the scope of our advance and to better prevent this unintended interpretation.

We reiterate that R1 expressed satisfaction with these revisions.

R4:

As with Point 1, these comments appear to primarily refer to R1's previous round of comments. If R1 is now satisfied, I think this matter should be considered closed – unless, R2 expresses specific concerns re: the revised manuscript that has been re-submitted.

Point 5:

R2:

Page 5: the author's express a desire to have the response to the reviewers included in their manuscripts, and I agree, such information is needed in this one. This was one of the big issues I had with this manuscript the first time I read it: I do not believe Nat Comms provides the space necessary for the authors to properly discuss the topic or methodology due to space limitations of the journal.

Authors:

Appropriateness for the journal is at the discretion of the Editor, who has communicated that our manuscript is a good fit for publication in Nature Communications.

R4:

The reviews are routinely available/published, I understand, due to the Journal guidelines, which sounds like it would satisfy all parties. From the reviewer email: “Nature Communications uses a transparent peer review system for published work, and therefore publishes reviewer comments and the authors' rebuttal alongside the paper. By submitting a reviewer report you agree to the publication of the comments made to the authors (any confidential comments to the editor will not be published).”

My understanding is that the appropriateness of the Journal (in terms of space limitations/formatting) is of Editorial rather than reviewer concern. Appropriateness due to content/subject matter would be of reviewer concern.

Point 6:

R2:

Page 13: I understand the author's feel I am being overly pessimistic about the generalizability of this metric, but, unfortunately, the author's have not offered sufficient evidence to the alternative, and my concerns are supported by the evidence presented in supp. Fig 9 (with regards to the glenohumeral joint). My issue was not whether or not you could use the raycasting method on other joints (of course you can!), it's about whether the results will be useful. The authors have expanded their limitations discussion in the manuscript, which is appreciated, but I believe you should have a better understanding of the limitations of a method in a manuscript which is stating, quite conclusively “this is the way forward.” This speaks to the “dual nature” of the manuscript: one being the reconstruction of the pedal pose of an extinct animal, and the other presenting the new method. It is, essentially, two studies wrapped into one, and being presented in a short communication.

Authors:

We are gratified to learn that the Reviewer does understand that our raycasting approach can easily be applied to other joints. As stated in our Discussion since our original submission: “comparing articulation score distributions against in vivo kinematics for further joints and taxa will illuminate the broader utility of articulation analysis, facilitating the widespread application of these data in vertebrate functional reconstruction” (previous submission Lines 172-174).

In stating this, we have made clear to the reader that the only way to determine “whether the results will be useful” (Reviewer's phrasing) is through comparisons with in vivo data that are specific to the research question at hand. We cannot possibly test the full diversity of vertebrate joints in one manuscript, and neither we nor the Reviewer can evaluate how “useful” articulation analysis will prove for the reconstruction of any particular behavior at any particular joint without conducting such tests. However, our paper demonstrates the potential great utility of this approach using our selected case study, while also laying the necessary groundwork in conceptual and methodological advances for its future expansion.

To the Reviewer's concern about the glenohumeral joint in Supplementary Figure 9, then, we emphasize that the exemplar configurations we have displayed will successfully receive different articulation scores – we have now added quantitative articulation subscores to the Supplementary Figure 9 caption to make this as clear as possible – and determining how “useful” these differences in score are will rely on future collection of and comparison with in vivo shoulder kinematics.

Dedicated analysis of the human glenohumeral joint is far beyond the intended scope of this manuscript and this example was added to assuage this Reviewer's concerns in the previous round about the broader generalizability of this approach, an addition which is hopefully also of interest to the journal's broad readership. As of interest to the Reviewer, we offer the following screenshots of extremes of in vivo glenohumeral rotation kindly provided by an orthopedic biomechanics colleague to emphasize that in a wide range of rotational poses, joint translation is limited such that articular congruence (informally assessed here based on apparent matching of joint surface curvature) appears to stay much higher than in our low-scoring 6 dof configuration figured in Supplementary Figure 9. This would tentatively suggest that raycasting analysis of this joint will in fact have future value for functional reconstruction of this joint, though we emphasize the need for thorough, formal articulation analysis and comparison with in vivo kinematics. If the Reviewer has a particular interest in the human glenohumeral joint, we hope they will implement our new approach to investigate shoulder articulation in more detail!

To concern about the "dual nature" of our manuscript: the Reviewer has repeated the same concern within this round of review. We again, as above, defer to the judgment of the Editor, who has communicated that our manuscript as written is a good fit for Nature Communications.

R4:

My reading of the manuscript is that this method is being proposed as a way forward (rather than the way forward), the need and usefulness of which is being presented to be critiqued and tested by others. This intention is clarified by the authors' response above "we hope they will implement our new approach to investigate shoulder articulation in more detail". Perhaps the back-and-forth between R2 and the authors indicates that this intention could be made clearer in the manuscript for the benefit of other readers who may similarly interpret the paper's offering ("why not this?") as imperative ("you must do this").

I do think there is some concern in developing the method and validating/ground-truthing it against predominantly one joint (the ankle of guineafowl), then applying the method to different joints (IP & MTP joints of *Deinonychus*; see my comments/feedback on the manuscript itself). This concern may be separate from R2's point; it's a little difficult to tell – apologies if so.

But as a short communication pitching a new method and its application, or the concept of a dual manuscript, it not problematic in of itself. The authors could choose to do a longer form manuscript and/or split it into two studies with multiple joint/joint types. Indeed, it may make the conclusions more robust and "sell" the method more effectively to do so. But in my opinion it is the authors' choice how to put this manuscript into the world and receive scrutiny from others working in the field. Carefully caveated (e.g. clear methods signalled, as in my comments/feedback on the manuscript) and perhaps some care with the language describing impact/previous study, I see no reason why the short communication or combined method + application is inappropriate.

Point 7:

R2:

Page 9: I thank the authors for their clarification, there was indeed a miscommunication. Pose certainly does need to be defined, and early on in the manuscript (but see comments on this manuscript below). In its current form, joint poses (line 51), is now defined as rotational combinations of joint articulations, which I do not believe is what the author's mean the definition to mean. I believe a proper definition would be something closer to "position of body segments,

relative to each other, that are connected by a joint". This is closer to the definition given in L40- 41 for a different term (joint configurations). As an aside, the authors also misunderstood my comment, as I was not suggesting in my comment that motion in animals was deterministic (and neither is motion in robots, not the least bit because of the inclusion of AI and machine learning, but also because of things like vibrations and inertial momentum, making precision manufacturing by robots difficult/impossible today)

For the title, I would recommend changing "locomotor joint poses" to "joint poses during locomotion" for clarity.

Authors:

The Reviewer is repeating a concern from the previous round of review. We clarify that we have defined joint poses as combinations of excursions in all three rotational degrees of freedom (not "rotational combinations of joint articulations;" we are unsure what combinations of articulations would mean and therefore do not use this language). Likewise, we have defined joint configurations as combinations of excursions in all six (all three rotational and all three translational) degrees of freedom. These definitions were added explicitly to the manuscript in last submission's Lines 40-41 and Line 51 during the last round of review to increase clarity. We re-emphasize as discussed in the previous round of review that there is no "proper definition" for joint poses, and that different fields use the word pose differently – we offered several citations to this effect in the previous round of review, and repeat that response here:

the word "pose" has been used in the orthopedic, anthropological, paleontological, zoological, computer animation, and biomechanical literature to refer to everything from a joint's excursion in a single rotational degree of freedom (e.g., Senter, 2009), to a joint's excursions in all three rotational degrees of freedom (e.g., Manafzadeh & Padian, 2018; this paper), to a joint's excursions in all six degrees of freedom (e.g., Bishop et al., 2023) to the combination of joint "poses" (defined as any of the previous) throughout a limb (e.g., Lauer et al., 2022) or even an entire individual/character.

Bishop, P. J., Brocklehurst, R. J., & Pierce, S. E. (2023). Intelligent sampling of high -dimensional joint mobility space for analysis of articular function. *Methods in Ecology and Evolution*, 14(2), 569-582.

Lauer, J., Zhou, M., Ye, S., Menegas, W., Schneider, S., Nath, T., ... & Mathis, A. (2022). Multi-animal pose estimation, identification and tracking with DeepLabCut. *Nature Methods*, 19(4), 496-504.

Manafzadeh, A. R., & Padian, K. (2018). ROM mapping of ligamentous constraints on avian hip mobility: implications for extinct ornithomirans. *Proceedings of the Royal Society B: Biological Sciences*, 285(1879), 20180727.

Senter, P. (2009). Pedal function in deinonychosaurs (Dinosauria: Theropoda): a comparative study. *Bulletin of the Gunma Museum of Natural History*, 13, 1-14

Therefore, other readers would certainly disagree with both us and this Reviewer about their intuition for the "proper definition" of "pose," and it is most important for clarity and accessibility of our work that we are explicit about what we mean by this word for this specific manuscript.

Again, this is a change we already implemented in the previous round. We already changed the title of our submission to "...locomotor joint poses" in the last round of review in response to this Reviewer's concerns that "...locomotor poses" did not clearly enough reflect that our definition of pose was on a per-joint basis. The title the Reviewer now suggests is fully synonymous with our current title and purely a difference of semantic preference, so we respectfully decline the suggested

change. Instead, to increase clarity as fully as possible, we have added a parenthetical to the Introduction in new Line 66 (joint poses actually used during terrestrial locomotion (i.e., locomotor joint poses)), explicitly drawing the connection the Reviewer requests, and have changed two instances of “locomotor pose” to “locomotor joint pose” (new Lines 85 and 89) in the Results for better consistency with the existing title.

R4:

The manuscript’s current use/definition of ‘joint pose’ and ‘locomotor joint pose’ is both clear and appropriate, in my opinion.

Point 8:

R2:

Page 11: In response to there being no other methods present, It does not need to be another 6 DoF method, it just needs to be another method for estimating joint poses, and the authors have actually discussed an alternative methodology previously in this review (i.e., the “prism method” for estimated ROM). While imperfect, it would be useful to see how results from that method, or the “qualitative method” of joint poses discussed later in the response to R1, compared to results from this method. If results are similar in terms of locomotor reconstruction (forward dynamic simulations), then does this more complicated ray casting method of estimation actually need to be used? If not, it creates a good argument for using this model (which has its own inaccuracies) over other models. I stand by my earlier comment that a comparison is needed to provide tangible evidence for how this method is superior and provides superior estimates for extinct animals.

Authors:

The Reviewer is repeating a concern from the previous round of review. The Reviewer, in the previous round, stated “But the paper neither presented a method for evaluating joint articulation or compared this method to others in terms of reconstructions (showing that this method is more accurate),” so their desired comparison was of methods for “evaluating joint articulation.” We reiterate, as stated in our previous round of responses, that there are no other reproducible methods in existence to accomplish the comparison that the Reviewer requests.

The Reviewer now raises suggestions for two potential “alternative methodolog[ies].” First, they bring up the “prism method.” We invented the prism method two years ago (Manafzadeh & Gatesy, 2021 J Anat). The prism method in no way evaluates articulation. It is simply a method for sampling joint translations in hinge joints. In fact, we already implement the prism method in the present study when sampling 6 dof joint configurations, and then we evaluate articulation at each viable (non-interpenetrating) joint configuration using our new raycast-based approach. This has been explicit since the initial submission in the Methods section, where we state:

“At each rotational pose, 343 potential translation combinations were allowed ($7 \times 7 \times 7$ translations sampled using the prism-based hinge joint translation method proposed by [26])” (new Lines 245-247)

and

“We then conducted an articular raycast at each viable (i.e., non-interpenetrating) configuration.” (new Lines 256-257).

Manafzadeh, A. R., & Gatesy, S. M. (2021). Paleobiological reconstructions of articular function require all six degrees of freedom. *Journal of Anatomy*, 239(6), 1516-1524.

Second, they bring up the “qualitative method ... discussed in the response to R1.” The qualitative method of evaluating joint articulation is just that – qualitative – and is therefore subjective based on the interpretations of the specific researcher implementing the method. That said, we already include the results of this method as applied to the foot of *Deinonychus* as Supplementary Figure 8; we added this in the last round of review in response to a different comment by this same Reviewer. It appears the Reviewer may have missed this addition. We reiterate our response from the last round:

The most quantitative hypothesis of *Deinonychus* joint motion to date was published by Senter (2009), who relied on manual manipulation and subjective assessment of proper articulation in an attempt to identify each joint’s full ROM (critically, not its locomotor ROM, which is a subset of full ROM [see Manafzadeh et al., 2021]). Notably, Senter’s hypotheses sometimes underestimate even the locomotor ROM we recover here!

To make more explicit in the paper that we are contributing new data to our understanding of *Deinonychus*, we have created new Supplementary Figure 8 to directly compare our results with Senter’s. Therefore, we posit that the raycast-based approach we use here is not “more complicated,” but that it is the only objective method currently available to the field for quantitatively evaluating joint articulation. We cannot claim that our method is “superior,” nor do we intend to – we intend to provide the first and only method for accomplishing this goal. We are indeed hopeful that this will serve as the foundation for the future development of methods superior to ours, such that future comparison is possible!

R4:

My understanding/awareness of previous methods is that they give an idea of ‘possible’ vs ‘not possible’ poses, i.e. binary yes/no. One could compare these ROM estimates to the ROMs actually used in locomotion – and in fact the manuscript does this, in Fig 2b.

This manuscript’s method, using articulation score, proposes a quantitative refinement of the binary yes/no.

In this sense, I would tend to agree with the authors. Although previous methods have made substantial efforts towards objectivity, these methods are not quantifiable in the same way as the articulation score (to my knowledge & understanding). Given the caveat above, I personally can’t see how a meaningful comparison could be made.

Point 9:

R2:

If higher articulation scores do not represent stability, what do they biologically represent? This needs to be explicitly stated in the manuscript, to prevent others from making the same mistake I have. A definition of articulation early on would clear up some of this confusion, as the authors are using terms like “disarticulation”, “misalignment”, and “subluxation,” which compromise the stability of the joint, to explain improper articulation, but are also stating that articulation scores (and thereby, articulation) do not represent stability.

Authors:

The Reviewer is repeating a concern from the previous round of review. In the previous round, the Reviewer stated:

“The reason I say the paper did not present a method for evaluating joint articulation is that it provides a method for quantifying the likelihood of a joint engaging in a pose based on a measure of joint stability, but this does not quantify articulation. Nowhere in this metric can you determine if a joint is articulated or not, just if a joint is likely to engage in a pose during walking/running in a straight line.”

In response to this miscommunication of our intent, we already made the goal of our articulation analysis (“measuring the quality of joint articulation for any viable, static six-degree-of-freedom joint configuration”) more explicit in Lines 60-62 of the previous submission. Also in response, we also already clarified in our Methods (previous submission Lines 257-264) that:

“The articular raycasting approach we propose here aims to capture information about the morphological relationship of a pair of mating articular surfaces in any viable (i.e., non-interpenetrating), static six-degree-of-freedom joint configuration. Conceptually, our approach provides data about the relationship between these surfaces’ 3-D curvatures. We select this emphasis here because interactions between articular surface curvatures are fundamental to joint function (see [36, 64]). The relationship between the curvatures of mating articular surfaces in any given configuration dictates the paths of minimum work along which joints habitually move, as well as the capacity of a joint to effectively distribute and evenly transmit loads.”

And later in lines 280-287 that:

“We propose that the formation of a successful ray hit means that articular curvatures are aligned such that there is a capacity for meaningful biomechanical interaction between the pair of articular surfaces in vivo. Therefore, if even a single ray hit successfully, we assigned the joint configuration an articulation score of greater than zero (see formula, below). If no rays succeeded in hitting the mating articular surface, we considered the joint to be unscorable, and gave the joint configuration an articulation score of zero. Articulation score was then calculated using three parameters – overlap, symmetry, and congruence – each of which receives an individual subscore from 0 to 1.”

These sections of text already explicitly address our understanding of the biological meaning of joint articulation.

We disagree that “disarticulation,” “misalignment,” and “subluxation,” necessarily communicate something about joint stability. These words, like “pose,” are highly fraught, having been taken to mean very different things in different papers across different fields. It is possible the Reviewer has encountered them largely in the context of stability, leading to their impression, but in the absence of any citations provided by them we cannot be sure. We reiterate that the word “stability” has never appeared, and still never appears, in our manuscript. We also already acknowledged in the previous round of responses that: “we do think the Reviewer’s general intuition is correct, and it is reasonable that joint configurations with a high articulation score may be those that are more stable. However, because such a claim would need to be tested experimentally and because making such a claim is not our goal, we avoid using these terms.” (Emphasis added here.)

R4:

Stability, in the sense used by R2, is a bit of a woolly term that could mean different things to different people. To me, stability depends equally on the surrounding muscles, ligaments, menisci etc as it does the bony morphology, and implies specific loading regimes. I disagree that the articulation

score, as defined in the manuscript, makes implications about a joint's stability: different joints (ball-and-socket, condylar, planar and so on) all have different inherent stabilities but I suspect that they would have similar articulation scores / ranges of articulation scores.

In my opinion, this confusion is not an issue in the current manuscript; articulation score is not implied to represent 'stability'. It seems clear enough to me that articulation score is a 'plausibility' measure of pose occurrence in *in vivo* locomotor behaviours.

Point 10:

R2:

While the authors state that the articulation metric is not probabilistic, they are treating it like a probability in their analysis. I.e., scores above 95 were used further means the probability that scores less than 95 would be engaged in is low. They also discuss it in a probabilistic manner in the manuscript (see comments about current manuscript below). They are not using it as a 95% CI, but they are using it as a probability (i.e., the probability that the joint will be in that pose, due to an articulation score which has been validated with a subset of joints from two species)

Authors:

The Reviewer is repeating a concern from the previous round of review. Again, we have already addressed this concern in the previous round: "we clarify that this metric is not probabilistic, nor does it attempt to highlight "likely" poses. Articulation score can be calculated for a pair of bony articular surfaces in any viable, static six- degree-of-freedom configuration. What we find here is that the joint poses used during locomotion happen to consistently fall in the highest-scoring region of pose space -- but we find this using experimental data; the score itself contains no measure of probability and is not inherently tied to locomotion in any way. We again hope the new Supplementary Text and Figures we have added will make this more clear."

We are certainly identifying a consistent correlation in extant animals and applying it to new data from extinct animals within a phylogenetically and morphologically informed framework. Perhaps we differ with the Reviewer in terms of our understanding of the meaning of "probabilistic," because if this general approach is viewed as probabilistic, we would argue that all application of data from extant animals to the fossil record is similarly inherently probabilistic – this is the foundation of modern functional morphology in paleontology. Regardless, the Reviewer does not suggest here that they view this approach as problematic, even if "probabilistic," nor do they propose a constructive path forward for revision.

R4:

In my reading of the current manuscript, I didn't see anything that claims a score of 95+ implying the likelihood (as a percentage) that the pose is used. Semantically, I suppose that low-scoring poses are less likely (less often) used in locomotion, and one could argue that they should/could be eliminated when reconstructing animal movement. But I don't see an issue with that logic, accepting that some *in vivo* poses did fall below 95 score and so would be incorrectly eliminated. The probability is discussed in a qualitative way, which to me did not seem problematic.

I do think the manuscript would benefit from more clarity on how the 95 score cutoff was chosen, and how future researchers can decide articulation score cut-offs for their own datasets.

Point 11:

R2:

Page 12: The authors write “The score simply offers a quantifier of the quality of joint articulation for any viable, static six-degree-of-freedom configuration of mating bony articular surfaces – not “joint ROM”.” The authors are, in fact, quantifying joint range of motion based on their articulation scores: high articulation scores = inclusion in ROM, low articulation scores = exclusion. In their study, ROM is defined as having an articulation score of 95+.

Authors:

The Reviewer is repeating a concern from the previous round of review. We reiterate our previous statement that we are not quantifying range of motion based on our scores – the Reviewer has kindly quoted that previous statement for us. Our score offers information about the quality of joint articulation across all of joint pose space. We have identified that locomotor poses consistently fall within the subset of pose space with articulation scores of 95-100. The score itself does not quantify range of motion. The subset of pose space identified through articulation score does not reflect the joint’s full range of motion, but rather a locomotor subset of range of motion.

This initially came up because the Reviewer stated in the previous round of review that “What I believe the paper is about is a method for quantifying joint ROM based on a measure of stability.” We have already clarified in several ways why this belief is incorrect – we have reiterated our disagreements about both “quantifying joint ROM” and “a measure of stability,” already discussed once in the last round of review, here and above.

R4:

It seems to me that there is scope for misunderstanding on an important point of the manuscript. The authors clarify here “The subset of pose space identified through articulation score does not reflect the joint’s full range of motion, but rather a locomotor subset of range of motion.”

This is worth making more clear in the manuscript, particularly early on and for parts I have commented separately in my own feedback as being general/minimising of prior work. I think it would be easy to take a similar interpretation as R2 does from the manuscript alone – and I similar initially misunderstood until reading the ‘Response to reviewers’ – that the raycasting method is not primarily a method to more objectively estimate ROM, but rather that refining the ROMs likely/commonly used by a joint during locomotion.

I feel that the generalisations in the manuscript which I mention in my own comments/feedback are working against the authors’ intentions, by making it sound as if the method is about general ROM. Even if later clarified to be more specifically about locomotion, this can be missed by readers.

Point 12:

R2:

With regards to the whole-foot combinations of joint poses, part of the reason I used the words “opinion” and “random” is due to the writing of the author’s on L290 (L362 in the resubmission). The authors write “An inverse kinematic animation rig was used to animate a rough stride cycle for the pes of Deinonychus using inspiration from published extant avian toe tip motion” To me, this sentence comes across as the author’s constructed a model of extant avian toe tip motion, and then – combined with the joint poses – used it as inspiration to create the combined joint poses presented in the paper. In this case, inspiration would be similar to an artist using a set of trees in a forest as inspiration to create one, composite tree. In this case, the composite tree is influenced by the

author's opinion heavily, with no necessary protocol (i.e., random) as to why some features were chosen, and others were not. If this is what the author's did, then my comment stands true. If not, then this again speaks towards not enough detail being given in the paper for the study to be properly explained or replicated.

Authors:

The Reviewer is repeating a concern from the previous round of review. They include our text that "An inverse kinematic animation rig was used to animate a rough stride cycle for the pes of *Deinonychus* using inspiration from published extant avian toe tip motion." We pointed out in the previous round of review that this is common practice in the field of paleontology, stating in the last round that:

Here the hypothesis of covariation we chose to employ, as stated in Line 362, is based on that of avian feet. This is one reasonable hypothesis for non-avian dinosaur intra-pes coordination and is regularly applied by dinosaur paleontologists (e.g., Falkingham & Gatesy, 2014; Gatesy et al., 1999).

We have now edited our text in new Lines 361-365 to include reference to Falkingham & Gatesy (new reference 70) and to make more reproducible the specific way in which this inspiration was implemented in the current study. We reiterate that such practice is regularly applied in paleontology and that the goal of the present study was to reconstruct one of many possible well-supported stride cycles for *Deinonychus*, reminding the Reviewer as we discussed extensively in the previous round that natural variation is to be expected in the joint poses and configurations used in locomotor stride cycles. The edited text is as follows:

An inverse kinematic animation rig was used to animate a rough stride cycle for the pes of *Deinonychus* using inspiration from published extant avian toe tip motion by importing published videos and reconstructed animations of avian toe tip motion into Maya and aligning the metatarsus and distal phalanges of *Deinonychus* to the avian metatarsus and distal phalanges [69-70].

We also re-emphasize to the Reviewer that this inspiration was used simply to generate a starting point, first-pass "rough stride cycle" that was then modified into alignment with our articulation data, as explicitly described in the remainder of the Methods section.

R4:

I don't know enough about inverse kinematic animation or forward kinematic animation to critique this as a 'starting point'. But the description of the method seems justifiable for readers to judge and critique plausibility of the stride cycle for themselves, in my opinion.

The manuscript could ameliorate concerns by flagging this as 'starting point' more clearly in the Results (I appreciate this is covered in Methods, but this comes later in the manuscript due to the Journal format, and due to brevity could be missed). I think it is important to be clear that the starting point for the extinct dinosaur's locomotor pose 'reconstruction' was an existing kinematic dataset of bird locomotion stride cycle.

I would also query whether it could be considered circular (or otherwise self-reinforcing) if the articulation scores are validated/ground-truthed from a bird's locomotor kinematics (predominantly guineafowl), and the starting point *Deinonychus* kinematics are also derived from birds (presumably also guineafowl, although it's not stated in the manuscript which bird 'inspires' the *Deinonychus* stride cycle)? Does this really tell us something new about *Deinonychus*, or are these essentially *Deinonychus* bones animated with guineafowl kinematics?

Is there a method of doing this first step without the bird kinematics, to alleviate concerns of circularity and/or prior assumptions of guineafowl-like stride kinematics?

Point 13:

R2:

Page 13: as stated above, the authors have misunderstood my comment. It is not whether this method can be applied to all joints, but rather if the results would be useful or not. From supplementary Figure 9, it certainly seems it would be less than useful for the human glenohumeral joint, as it appears some positions that are subluxations would come across as having high articulation scores, which is a fundamental shortcoming of this method.

Authors:

The Reviewer has repeated the same concern within this round of review. We have already responded to this concern extensively above, but restate here that the exemplar configurations we have displayed will successfully receive different articulation scores – we have now added quantitative articulation subscores to the Supplementary Figure 9 caption to make this as clear as possible – and determining how “useful” these differences in score are will rely on future collection of and comparison with in vivo kinematics.

R4:

In my opinion, this point is not relevant to this paper. Determining utility for other joints and species depends on future research for the species and joints in question. It would be a huge study needed to claim that the raycasting/articulation scoring method was valid for all joint and species. The authors are not making this claim.

Point 14:

R2:

Now, this implies one of two things

- 1) this is a random correlation, in which case it is biologically meaningless
- 2) this is a causative relationship, whereby articulation scores are related to joint motion (rotation and translation)

The authors believe this is a causative relationship (e.g., see wording in abstract, discussed below), which is why they can use it to reconstruct extinct vertebrate locomotion. Assuming this is true, my question from before still stands: what is the underlying logic and biomechanical meaning of ray casting? I.e., why, functionally, was ray casting work? It is touched on in the paper, but not explained. E.g., why does asymmetry in ray length matter? Why does having more rays connecting the two surfaces when there is space between them biomechanically matter? Otherwise it appears to be more-or-less arbitrary why ray casting was chosen, and why it works in this paper.

Authors:

The Reviewer has repeated the same concern within this round of review. We reiterate, as above, that we already implemented substantial revisions during the previous round in response to this Reviewer’s concerns about this. Again, in our revised Methods (previous submission Lines 257-264) we stated that:

“The articular raycasting approach we propose here aims to capture information about the morphological relationship of a pair of mating articular surfaces in any viable (i.e., non-interpenetrating), static six-degree-of-freedom joint configuration. Conceptually, our approach provides data about the relationship between these surfaces’ 3-D curvatures. We select this emphasis here because interactions between articular surface curvatures are fundamental to joint function (see [36, 64]). The relationship between the curvatures of mating articular surfaces in any given configuration dictates the paths of minimum work along which joints habitually move, as well as the capacity of a joint to effectively distribute and evenly transmit loads.” (emphasis added here)

And later in lines 280-287 that:

“We propose that the formation of a successful ray hit means that articular curvatures are aligned such that there is a capacity for meaningful biomechanical interaction between the pair of articular surfaces *in vivo*. Therefore, if even a single ray hit successfully, we assigned the joint configuration an articulation score of greater than zero (see formula, below). If no rays succeeded in hitting the mating articular surface, we considered the joint to be unscorable, and gave the joint configuration an articulation score of zero. Articulation score was then calculated using three parameters – overlap, symmetry, and congruence – each of which receives an individual subscore from 0 to 1.” (emphasis added here)

These sections of text already discuss our explicit justification for selecting raycasting, a choice which was not arbitrary, and explain its biomechanical meaning.

R4:

In my opinion the manuscript, as quoted here by the authors, gives sufficient explanation for why raycasting was chosen.

Further, even if the articular scores are “randomly correlated” with *in vivo* locomotor pose use, as long as that correlation can be proved reliable (particularly in future studies of other joints/species), the method can still be useful even if it’s biological meaning is not fully understood.

Point 15:

R2:

Page 16: If the point of this method is to create an envelope of possible joint poses, the authors do not need to identify what percentage of sampled poses fall within the 95-100 articulation score envelope, but rather find the articulation score envelope that fits 100% of the joint poses. Otherwise, the authors are contradicting themselves (or believe 1.9% of the *in vivo* data is incorrect). I stand by my comment that articulation scores from 95-100 should not be used, but rather this window needs to be calculated from the *in vivo* data, and then used to re-reconstruct extinct animal locomotion.

Authors:

We note that we calculated and included the percentage of measured *in vivo* locomotor poses that fall within the 95-100 envelope in direct response to a request from this Reviewer. We think this is a useful addition that has improved our manuscript and thank them for the motivation. The Reviewer is repeating a concern from the previous round of review. A goal of this study was to reconstruct one of many possible locomotor stride cycles for the extinct animal *Deinonychus* using support from joint articulation data. Because ALL avian walking data fell within the 95-100 envelope, we maintain our judgment that it is reasonable to use this range to reconstruct one of many possible walking stride

cycles for Deinonychus, even if a small number of the poses used during swing phase of highest-speed avian running were found to exit the the 95- 100 score range.

R4:

The authors' judgement for using the 95-100 window does seem reasonable. However, it could be made more clear why the 95-100 score envelope was chosen in the first place. I can see R2's point that it would be useful to define the articulation score cutoff/window from the experimental data (it was only because of R2 and careful re-reading of the manuscript that I realised the 95-100 window is not defined by the locomotor experimental data as I had assumed).

If doing so creates an unhelpfully large window, could other weightings of the combined articulation score 'fit' the data better?

Point 16:

R2:

Page 17: "We contend that it would not change the reconstruction, because these poses would not be ruled out even if the envelope were expanded as the Reviewer suggests." The authors have misunderstood my comment. Of course, the poses would not be ruled out, but new poses that were ruled out before could now be chosen.

Authors:

The Reviewer has repeated the same concern within this round of review. As described directly above, the stride cycle we reconstruct here would not be ruled out, which is what is relevant for the goal of this study.

R4:

Whether or not poses are ruled out of Deinonychus' reconstructed stride cycle specifically is less interesting than the wider point R2 asks here (and above with point 15) of how an appropriate score cut-off is decided.

For future studies of other joints or species, how will the authors (or others) decide what is an appropriate cut-off window for reconstructions? The authors' decision to cut off at 95 and only eliminate 1.9% of running poses does seem reasonable. But what is 'reasonable judgement'? What if it were 5% of poses? 10%? What is (or could be) the basis for making the decision? Was there any bias in the kind of joints that fell outside the 95-100 window (mentioned in my comments on the manuscript itself, later)?

The Response does not address this wider point. And related to the my comment on circularity above, it doesn't seem a revelation that using the starting kinematics from bird data and the articulation scores validated against bird data result in a reconstructed stride cycle that is tolerant of more relaxed articulation scores. (If anything, the expanded envelope would result in a less tweaked/more faithful replication of the bird's stride cycle.) I don't see this as a good argument against more clearly defining the articulation score cutoff from the *in vivo* data. Even if the solution is just clearer acknowledgment within the manuscript that 95-100 was judged by the authors for this specific dataset (joint(s) and species), as balance between including all walking poses and only eliminating a subjectively acceptable minority of running poses.

Point 17:

R2:

Page 17: I misunderstood the previous read of the manuscript, and had thought the authors had run forward dynamic simulations, using the joint envelopes as boundary conditions. It is now my understanding the authors used the extant data to roughly position the bones of the extinct animal in the correct position, then manually altered joint positions so they fit within this envelope of possible positions. If this is correct, there is certainly a degree of artistic flair in the reconstruction that needs to be discussed, and the sensitivity of these reconstructions (inter- and intra-observer error) needs to be looked at, given this method is being advertised as a “new way forward” (my words, not the authors) for reconstructing joint poses and locomotor poses of extinct animals.

Authors:

The Reviewer has repeated the same concern within this round of review. We addressed this above, and as stated, edited our text in new Lines 361-365 to include reference to Falkingham & Gatesy (new reference 70) and to make more reproducible the specific way in which this inspiration was implemented in the current study. As discussed in the two responses immediately above this one, our goal was to reconstruct one of many possible walking stride cycles for *Deinonychus* using support from joint articulation data. The aspect of our approach that we suggest is a “new way forward” (we will agree with the Reviewer’s words) is the ability to test a reconstructed stride – no matter how it is generated – using data from joint articulation. Our joint articulation results have, of course, been heavily tested using the sensitivity analyses presented in the manuscript, because that is the extent of the advance we are proposing.

R4:

As far as I can tell, this seems to be related / the same points as raised Point 12, 15 and 16. If so, then the same comments stand as for those points.

Point 18:

R2:

Page 17: “However, the final sentence of this comment...” the authors have misread my comment. What the authors have “ground-truthed” was whether they could put joints in realistic poses. What I was saying what you cannot say your method has led to a more accurate reconstruction of the gait of the extinct animals because 1) it was not compared to results obtained using traditional joint constraints, and 2) it was not compared to how this extinct animal actually moved. If you wanted to use extant models for this purpose, you would have to take the data from one of the extant animals and do a similar reconstruction that you did with the extinct animal (potentially with bones from another individual of that extant species – e.g., another guinea fowl or emu) using both traditional joint constraints and these new constraints (estimated using ray casting) and see which one produces results closer to your experimental data. Again, if both produce similar results, the traditional method is easier and takes less time, so why should we bother with raycasting?

Authors:

The Reviewer has repeated the same concern within this round of review. We extensively responded to the Reviewer’s concerns about comparison of our method above; this comment appears to be a reiteration of the same concern. We repeat the final portion of our response here:

“we posit that the raycast-based approach we use here is not “more complicated,” but that it is the only objective method currently available to the field for quantitatively evaluating joint articulation.

We cannot claim that our method is “superior,” nor do we intend to – we intend to provide the first and only method for accomplishing this goal. We are indeed hopeful that this will serve as the foundation for the future development of methods superior to ours, such that future comparison is possible!”

R4:

As far as I can tell, this seems to be related / the same point as raised in Point 8 (i.e. why not compare to a previous method). If so, then the same comment stands as for Point 8.

Additionally, from my understanding, the approach suggested by R2 seems to me that it would be completely circular. Because of the way the ‘starting point’ would use extant data, ‘reconstructing’ the stride using the kinematics and articulation scores from the same species would (should) result in the starting kinematics being essentially unchanged in the reconstruction. A better suggestion for ground-truthing would be to reconstruct locomotor kinematics a different extant species (i.e. other bird, not guineafowl or emu) with similar joint morphology.

To me, the significance of this concern depends on whether the paper is primarily intended to:

i) propose a method as proof-of-concept, demo an application (the conclusions of which are less important than it being an example of how application could work), and let others validate (play with it until it breaks)

or

ii) conclusively showcase a new, finalised method as the next way forward in the field, illustrated with a robust application generating novel, concrete conclusions

If the latter, then I would tend to agree with R2 that validation against another extant species would be a valuable step to consider including. But I think R2, myself and the authors are probably all seeing the different intentions/purposes of the study, causing disparity of opinion on what is necessary for the manuscript to be published. I do not think it has to be a completely finished, polished and fully validated method to still be a valuable contribution to scientific discourse (i.e. option (i) above is very much of interest to the readership and worth publishing) – as long as this is clearly signalled and communicated in the manuscript. The solution may lie in tweaking the messaging of the manuscript’s purpose, rather than necessarily needing additional analyses.

In the second part of the Response discussion, R2 and the authors discuss the revised manuscript itself.

Point 19:

R2:

Abstract: “... locomotor joint poses consistently have high articulation scores. We then exploited this predictive relationship to constrain reconstruction of a pedal stride cycle...” These lines state that it is possible to predict joint poses based on articulation scores, but the authors failed to show (or statistically test) any type of predictive relationship between articulation score and pedal stride cycle in their extant individuals. That is to say, they showed pedal stride cycles were encompassed in the polygonal space created by articulation scores (you can predict the number of legs a zebra has), but never that articulation scores can predict pedal stride cycle (you can predict an species by knowing the organism has four legs).

Authors:

The Reviewer has repeated the same concern within this round of review. We have already responded to the Reviewer's concerns about probability above. We reiterate that the kind of reasoning the Reviewer appears to take issue with is the basis of all application of data from extant animals to the fossil record. We engage in this application here as responsibly as possible, within a phylogenetically and morphologically informed framework. The approach we take is standard to the field of paleontology.

R4:

The main issue here seems to be the manuscript's use of "predictive". It may just be the term "predictive" being perceived differently, as either:

- i) can 'predict' locomotor poses that are more likely to be used by ranking them via an articulation score window (my understanding is yes, that is what is being done here)
- ii) can predict how an animal must move based on articulation scores (no; the method can only refine the window of likely from possible joint poses)

I think this be resolved for both parties, without changing the meaning of the sentence, by removing the term "predictive". (i.e. changing the sentence to "we then exploited this relationship")

Point 20:

R2:

L33: I must have missed it the last time – a joint is not an organ

Authors:

Many sources would disagree. The Biology of the Synovial Joint book cited in that sentence states that "This unique collection of reviews has arisen due to the belief of the Editors that joints need to be studied as a whole organ." We align with this viewpoint and would similarly argue that a joint is an organ. This point has no bearing on the paper's data or conclusions.

R4:

This appears to be a minor point of personal disagreement. But I would agree with the authors that a joint can be considered an organ. Most common definitions of an organ are some permutation of "collection of tissues, as a structural unit, performing a function", which a joint can be considered to satisfy.

Point 21:

R2:

L40-41: this definition of joint configurations (the excursion of all three rotational and translational DoF) are what the authors stated was "poses" in the response to the reviewers.

Authors:

We cannot identify the claimed location in the response to reviewers. Our definitions of poses and configurations have, to our knowledge, stayed consistent, and are certainly consistent within the manuscript. Poses = 3 rotational dof, configurations = 3 rotational and 3 translational dof.

R4:

To me it seemed clear that ‘configuration’ and ‘poses’ were used synonymously in the manuscript. I don’t see the need to change anything on this point in the current form of the manuscript (if that is what is being suggested by R2 here).

Point 22:

R2:

L48-49: much related to gape in primates has been ground-truthed, so this statement is incorrect

Authors:

We are certain that the Reviewer’s statement is generally true, however, to our knowledge and despite substantial searching, we suggest that it is false in the context of what is discussed in lines 48-49 (joint articulation criteria for virtual ROM analyses). Metrics such as “maximum bony gape,” while very interesting, have not involved analysis of articulation between joint surfaces in living animals, which is how we define ground-truthing articulation criteria. This task would require the use of XROMM-derived data or rigid body marker tracking. Given that one of the co-developers of XROMM is an author on this paper, we question whether such analyses have been conducted. Published studies could always exist that we are unaware of, but the Reviewer did not provide specific citations.

R4:

I picked up on this statement (line 48-49) in my own first-pass manuscript comments, concerning generalising/narrowing statements that are made. Because 2/4 reviewers take issue with how this statement is constructed, I suggest this means it should be revisited. As currently worded, this statement excludes other studies that come to mind to the reviewers and therefore probably also the wider readership (primate gape as mentioned by R2, echidna shoulder ROM as explored in Regnault, Lai & Pierce 2021).

Point 23:

R2:

L81-82: In their response, the authors stressed that they were not discussing the “probability” of a joint ending up in a pose, but the writing here (“... mapping our articulation scores onto this mobility estimate enhances our knowledge of each pose beyond a binary “possible” or “impossible”...”) implies they are bringing joint articulation out of a deterministic, and into a probabilistic space. The authors are contradicting themselves.

Authors:

The Reviewer has repeated the same concern within this round of review. An “[enhanced] knowledge” is just that -- increased data. There is no allusion to probability, and we suspect that the relationship between articulation score and in vivo pose utilization is highly non-linear. We have already responded extensively above (and in the previous round of review) to the reviewer’s concerns about probability.

R4:

I agree that this discussion on probably has been covered in earlier points. My opinion as in those earlier points; I do not share the concern on probability.

Point 24:

R2:

L86-90: This is a major issue with the 95-100 window, as it does not encompass all possible ranges of motion. As these envelopes of possible joint poses are being used as boundary constraints for models of locomotion for extinct animals, they must encompass the entire possible ROM for the extant animals, otherwise they are poorly defined constraints.

Authors:

The Reviewer has repeated the same concern within this round of review. We have already responded to this concern above. As stated, our goal was to reconstruct one of many possible walking stride cycles for *Deinonychus* using support from joint articulation data. We also point out that experimental data have revealed that locomotor poses occupy a small subset of a joint's full possible ROM (see Manafzadeh et al., 2021 PNAS). Therefore, contrary to the Reviewer's statement that "envelopes... must encompass the entire possible ROM for the extant animals, otherwise they are poorly defined constraints," we would argue that identifying the entire possible ROM instead of what we have done here would be a very poor constraint.

Manafzadeh, A. R., Kambic, R. E., & Gatesy, S. M. (2021). A new role for joint mobility in reconstructing vertebrate locomotor evolution. *Proceedings of the National Academy of Sciences*, 118(7), e2023513118.

R4:

This discussion has been covered in earlier points (Points 15 & 16). To save repeating, my opinion & concerns are the same as in those earlier points.

Point 25:

R2:

L95-98: Text needs to be added to the supplementary material explaining the sensitivity analyses. As it stands, it is not clear what sensitivity analysis was done from the figures and captions alone. Results from the sensitivity analysis should be explained as well.

Authors:

The methods already describe what was done, stating in current Lines 334-339 that:

"Translational sensitivity analysis was performed for guineafowl individual 1 by allowing 1,331 translational combinations over a larger range (two additional increments of equal size at each end of the X, Y, and Z translation ranges) at five-degree angular resolution, and sensitivity to articulation score formula was evaluated by conducting an analysis for this individual at the original rotational and translational resolution, weighting congruence by single-condyle overlap rather than full-joint overlap."

The results already summarize the results, stating in current Lines 95-98 that:

"We found that our articulation score distributions are robust to sensitivity analyses conducted using additional individuals, increased translational allowance, and alternative score formulas (Supplementary Fig. 3-5)."

The captions of Supplementary Figures 3-5 offer both sets of information with additional detail, already stating that:

Supplementary Figure 3: “Results from the guineafowl individual figured in the main text, displayed at five-degree angular resolution in (a) and one-degree angular resolution in (d); a second individual, displayed at five-degree angular resolution in (b) and one-degree angular resolution in (e); and a third individual, displayed at five-degree angular resolution in (c) and one-degree angular resolution in (f), are grossly similar, especially within the highest-scoring region.” (emphasis added here)

Supplementary Figure 4: “Results from both the translational allowance throughout this paper, displayed at five-degree angular resolution in (a) and (b) and one-degree angular resolution in (c), and an increased translational allowance (see Methods), displayed at five-degree angular resolution in (d) and (e) and one-degree angular resolution in (f), demonstrate that although increasing translational allowance increases the region of pose space coded as viable (and therefore the region of pose space receiving articulation scores), the highest-scoring region of pose space remains identical.” (emphasis added here)

Supplementary Figure 5: “Results from both the articulation score formula implemented throughout this paper, displayed at five-degree angular resolution in (a) and one-degree angular resolution in (c), and an alternative formula that instead weights each condyle’s congruence by only its own overlap (rather than average overlap; see Methods), displayed at five-degree angular resolution in (b) and one-degree angular resolution in (d), are grossly similar.”

In the absence of more specific requests from the Reviewer, we are not sure what additional detail would be helpful. The details already provided allow full reproduction of the sensitivity analyses and offer our interpretation of their results.

R4:

The information appears to be provided in various places. However, perhaps R2’s concern could be addressed by having this information collected together in one place (e.g. within Supplementary info, a section on sensitivity analyses that collates & repeats the methods and results) for the interested reader’s ease of access?

Point 26:

R2:

L140-143: It was not the findings in this paper that enabled the reconstruction of the stride cycle, but “using the findings of this paper as constraints for XXX simulations...”

Authors:

We suggest this is semantic disagreement and respectfully decline the suggestion.

R4:

I can see both sides of this point. Some would not consider this a reconstruction in the truest sense because existing avian kinematics were used as the starting point. The avian kinematic starting point is worth making clear here in my opinion, because the Methods come after the Results / Discussion.

A suggestion for compromise between both views could be insertion of a short clarifier at line 140-143 (e.g. suggested insertion underlined here: “Applying this finding to the foot of *Deinonychus* thus enabled us to reconstruct a potential six DOF stride cycle for a non-avian paravian dinosaur, from

an avian kinematic start point, refining qualitative inferences made by previous workers based on observation and physical manipulation of fossil bones”).

This concludes the points of concern discussed by R2 and the authors in the Rebuttal document.

I have added my thoughts and comments below on the manuscript, from the first reading. Some of these comments are minor; others pick up on similar concerns to R2. As mentioned, I do not intend for these to saddle the authors with an additional review to address, but provide context for my opinions in the Points above. I summarise areas of major or common concern at the end of this document.

My (Reviewer 4) feedback on manuscript

- 1) The method proposed in the manuscript is novel and impactful. I agree with the authors that the methods for defining articulation in joints can be made more objective, and applaud their innovative development of a new method for doing so. However, in parts of the manuscript it does feel as if existing work (which this method builds on) is downplayed or generalised, and/or the topic at hand has been narrowed to a degree, in order to emphasise its impact.
 - An example is the text from line 46 onwards (“Current articulation criteria for virtual ROM analysis...”). Re: the critique of subjectivity on whether a joint “looks right”: arguably the use of shape primitives and anatomical joint axes to articulate digital joints are less subjective than this statement implies, and from my understanding this concept is in fact employed in this study too (in Methods, line 248-252, to set joint translation limits). Similarly, “have not been ground-truthed using data from extant animals”: I am aware of at least one other study beyond R2’s example, that evaluates how digital joint articulation choices affect ROM in an extant species (Point 22, above).
 - Line 130; “existing computational articulation criteria, which rely on simple closest-point distances”. References for this statement? My understanding, perhaps outdated, is that many studies set articulation via shape primitives paired to bone surfaces. That seems like it is also “exploiting information about local articular surface curvature”, albeit in a different way to this study. If I haven’t misunderstood and that is what is referred to here, then ‘simple closest-point distances’ might need some unpacking or referencing, to avoid coming across as minimising other studies’ efforts towards objectivity.
 - Line 52 presents an opinion as established fact or widespread concern “This sobering conclusion calls into question [all studies to date]...” The conclusion follows from the authors’ previous work [26]. Is this questioning of all studies to date solely the authors’ opinion, based on their previous study? If so, consider clarifying (“In our opinion, this sobering conclusion calls into question...”) If it is an agreed concern that others have also expressed, can that be referenced?
- 2) I appreciate the pressure for concision in the manuscript, as well as the journal format having much of the detail (Methods) last, but this sometimes results in details that are very pertinent feeling obscured.
 - Line 62-68 talks about analysing joints in two birds, comparing (validating) results against *in vivo* locomotor kinematics, and then using the validated method to reconstruct locomotor joint angles in an extinct dinosaur. However, in the Methods, the raycasting method seems to be only validated against the ankle and MTP of guineafowl (and actually predominantly the ankle) in locomotion. The emu ankle and

MTP and most of the guineafowl MTP joint angles are analysed using still 2D photographs, where the bones are not visible. No IP joints are analysed or validated in extant birds. Then the raycasting method is applied to the MTP and IP joints of the extinct dinosaur (despite these joints not being the focus of the method development or validation). I am concerned that the lack of clarifying detail early on unintentionally misleads the reader and inflates the actual scope of the study. Rather than robustly analysing 3D/6 DoF movements of three joints in two extant species and application to a third extinct species (as implicit throughout earlier parts of the manuscript), the Methods reveal that the only common joint through method development, validation and application is the MTP of the guineafowl and the MTP of the extinct dinosaur. The manuscript critiques lack of ground truthing in extant species (fair critique; this does not appear to be widespread practice), but this study itself applies the methods to different joints (IP and MTP) from which it has been validated against (mostly ankle, some MTP).

- Lines 104 onwards: It is not clear here that the results refer only to locomotor pose space. E.g. “severely restricted LAR potential” and similar statements do make it sound as if the results are discussing total ROM. This is similar to R2’s concern in Point 11 (above).
 - Line 350, 353 “n=11,510” and “n=232” - these numbers are not very informative alone. How many individuals, and how many trials from each individual? How many are walking vs. running (especially important, given that the 95-100 window was decided based on including 100% of walking poses and 98% of running poses). Same questions for the guineafowl MTP (232 joint poses).
 - Line 357: “photographs available on the internet”: I’m sceptical about how powerful the 2D comparison of still images can be in validating articulation scores, without the bones/articular surfaces visible. If it is felt strongly that these data warrant inclusion in the manuscript (rather than weakening it), more detail would be reassuring (are these animals walking/running? Can example photographs with the skeletal overlays be uploaded?)
 - Line 368: “each joint” – Which joints were done in *Deinonychus*? It looks like the IP and MTP from the figures, but I don’t think this is clearly stated in the main manuscript.
- 3) Some general questions on ray-casting method that might benefit from clarification:
- How can this method formalise “proper articulation” in terms of joint distance; can it account for ‘disarticulation’ (distraction; translation along the proximodistal axis)? Is this via the overlap score; and if so, is it possible to distract the joint in such a way that overlap remains high whilst joint distance is implausibly large? The joint spacing has the capacity to alter joint ROM estimates, but I imagine is tricky if the articular cartilage thickness and morphology are not known. Was joint spacing checked against the *in vivo* condition, once the bird legs were articulated?
 - Line 319: Were other versions/weightings of the formula tested against *in vivo* data to see if there is a better/more optimal fit? Or does this formula choice best represent the underlying theory of what the balance of different scores ‘should’ look like for proper articulation?
- 4) Vague areas that could benefit from clarification
- Line 61: “viable” – what does that mean, in this context?

- Line 69: “locomotor morphology” – does this mean the morphology of the locomotor system (e.g. shape of the leg bones) or locomotor style as inferred from morphology, or something else?
 - Line 89-90: Was there any bias in joints or species that fell outside the 95-100 window? (e.g. higher sampled joints such as the guineafowl ankle, or lower sampled/lower quality sampling such as the MTP or 2D photographs?)
 - Line 159: “efficient synovial joint motion” – what does efficient mean in this context?
 - Line 160-161: “stabbing or pinning” vs “slashing or digging” – for a layperson/unfamiliar reader, what would be the difference? (For example, to me slashing and stabbing seem quite similar. If I slash or stab with a knife, isn’t this more about the position/action of my proximal joints, rather than how I am holding the knife? Can it not slash with the claw in any position?)
 - Line 246 parentheses “7x7x7 translations sampled using the prism-based hinge joint translation method proposed by [26]” – a reference is given but the explanation doesn’t add anything for the unfamiliar reader, it might as well just be the reference. A brief elaboration would be helpful of how translations were incorporated.
- 5) Minor: some parts of the manuscript are confusingly worded. I have listed below although I appreciate this may be overstepping my remit as a reviewer and straying into writing style preferences – apologies if so.
- Line 52 “precludes excluding very many poses at all” – the double negative of preclude and exclude is a bit confusing. Similarly the start of the sentence being a double negative (“not unlikely”) at Line 151.
 - Line 52, 78 and 248 “conservative” – the term may mean different things to different people, and although its use is defined in-text via parentheses, it still could be needlessly confusing where another term could be clearer and more concise. To some, a conservative estimate of articulation or ROM is one that includes only poses that are definitely biologically possible in life – a safe or cautious estimate for inferring possible poses (i.e. minimising ‘false positives’). Whereas in the manuscript ‘conservative’ is used to mean estimates that might include biologically impossible poses as well as those used in life – the opposite, cautious in the sense that it does not eliminate possible as well as impossible poses (i.e. minimising ‘false negatives’). Neither is right or wrong, but another term (e.g. permissive, relaxed) would be clearer in my opinion.
 - Line 139 “articulation analysis successfully distinguishes the subset of joint poses used during dinosaurian locomotion”. I appreciate that the guineafowl and emu are dinosaurs, but use of dinosaurian without clarification here risks confusing readers (particularly non-dinosaur folks) that this could be referring to all dinosaurs including extinct ones. Although correct, for clarity the authors might consider referring to ‘birds’ when talking about the guineafowl/emu, or at least clarifying “during extant dinosaurian terrestrial locomotion” here.
 - Line 197: “third proximal pedal phalanx” – just a bit confusing, because it could sound like ‘third most proximal’. Suggest ‘proximal phalanx of digit III’?
 - Line 250: suggest rewording to bring condyle to the fore e.g. “fit to the condyle of the distal tibiotarsal...” just because it otherwise comes quite late in the sentence and otherwise isn’t clear that the cylinder is being fit to the condyle of each bone, rather than the whole bone.

- Line 274-276: Info in parentheses interrupts the subject and flow of the sentence. I'd suggest putting the clarification (which is appreciated!) elsewhere, such as the top of a sentence.

Summary

As mentioned previously, I feel the method proposed in the manuscript is novel and impactful. I agree with the authors that the methods for defining articulation in joints can be made more objective, and applaud their innovative development of a new method for doing so. I think it would be of interest to the journal's readership and to workers in the field.

There are some areas of common or outstanding concern, however, which I feel it would be necessary to address before publication. The discussion/rebuttal has become lengthy and many areas are repetitive examples of the same general topic, so I have tried to summarise the main points as I see them.

Firstly, issues around aims and generalisations of the manuscript, raised by reviewers in reference to various themes or statements (**R2: Points 11, 12, 18, 22, and R4 Comment 1**). These included the general messaging and purpose of the paper; the generalisations or narrowing of categories having the effect of minimising past work; and the important idea that this ray-casting method proposes to refine locomotor ROMs and not estimate general ROM.

Secondly, methodological concerns. R2 expresses repeated concern about the starting point of the extinct dinosaur locomotor reconstruction and related to that, I ask questions about the potential circularity of the starting point (**R2: Points 12, 26**). R2 and I also share interest in increased clarity, guidance, or comment on how the cutoff window for articulation scores is chosen (**R2: Points 10, 15, 16, 17, 24, and R4 Comment 2**).

Independently of R2, I have questions about the development and validation of the method primarily using guineafowl ankle data, and its subsequent application to the extinct dinosaurian metatarsophalangeal (MTP) and interphalangeal (IP) joints (**R4 Comment 2**). The quantity and quality of guineafowl MTP joint data and especially 2D emu data strike me as questionable. Why were no IP data used from extant animals given that these are most of the joints that are 'reconstructed' in *Deinonychus*' foot? How much are the external 2D photographs of the guineafowl and emu MTP and ankle joints actually able contribute? The inclusion of the latter feels hasty and combined with lack of clarity/detail very late in the Methods risks coming across as 'padding' that weakens reader confidence in the method & manuscript, rather than strengthens it as was probably intended. I am worried that it results in unfavourable perception of the study, when combined with the above concerns. This is a shame as I feel the method itself is genuinely exciting and addresses an important part of the field.

I hope that my contribution to this process has been useful and thank the authors, reviewers and editor again for the opportunity to comment.

Manafzadeh et al.: Responses to Third Round of Review

We thank Reviewer 4 for their balanced mediation and thorough further feedback, and we thank the Editor for allowing us to submit a fourth version of our manuscript.

Reviewer 4 offered their perspective on 26 existing points from the previous two rounds of review and raised 5 new points of their own, resulting in 31 total points to address.

Of these 31 points raised, Reviewer 4 deemed 15 closed and requested further action on 16. We have made changes to the text of our manuscript to resolve all 16 outstanding points. We summarize our actions below and respond in detail to each highlighted point on the following pages.

Existing Points mediated by R4:

- 1) Deemed closed
- 2) Deemed closed
- 3) Deemed closed
- 4) Deemed closed
- 5) Deemed closed
- 6) Text change
- 7) Deemed closed
- 8) Deemed closed
- 9) Deemed closed
- 10) Text change
- 11) Text change
- 12) Text change
- 13) Deemed closed
- 14) Deemed closed
- 15) Text change
- 16) Text change
- 17) Deemed redundant
- 18) Text change
- 19) Text change
- 20) Deemed closed
- 21) Deemed closed
- 22) Text change
- 23) Deemed closed
- 24) Deemed redundant
- 25) Text change
- 26) Text change

New Points raised by R4:

- 27) Text change
- 28) Text change
- 29) Text change
- 30) Text change
- 31) Text change

Thank you for the opportunity to contribute to the peer review process of this interesting paper, proposing a novel method to improve objectivity in our field. Where there is scope for progress and improvement, there will also likely be lively debate. I hope that my opinion is useful to all parties involved.

I appreciate that I have been asked primarily as a referee/additional opinion on the discussion between Reviewer 2 (R2) and the authors, to assess how the concerns of R2 have been addressed, rather than review the manuscript. However, the Editor notes that feedback on the manuscript more generally is welcomed. I thought it sensible to read the manuscript and record my own thoughts independently before diving into the R2/author discussion, for the purposes of objectivity. I do not intend for these comments to contribute further to the author workload (as I understand the manuscript has already been seen by three prior reviewers), and I understand if my comments are taken only as context to the R2/author discussion rather than additional actionable tasks.

I have formatted my opinion as three sections. First, specific point-by-point comments on the 26 points of concern in the R2/author discussion. Then my own comments on the manuscript. Finally, since the resulting doc is quite long and has a fair amount of repetition, a summary of main points.

Opinion on Response to Reviewers (Round 2)

In the first part of the Response discussion, R2 and the authors refer to the previous rounds of review (both of R2 and of other reviewers). I cannot see these previous rounds, and nor do I think I need to, however I note that I may be missing the full context of these discussions. I have done my best to express an opinion, but apologise if I have misunderstood any of the earlier context. I have duplicated the Response discussion and added my comments in yellow text for clarity.

We thank Reviewer 4 for their efforts and interest in our manuscript. We offer our own responses to Existing Points 1-26 throughout in bolded red text. We have added page breaks for each point to improve readability.

Although we sincerely appreciate the Reviewer's clarification that their own comments need not be taken as additional revision tasks, we found their perspective to be very helpful and have therefore also made several changes to our manuscript in Response to New Points 27-31 (i.e., Reviewer 4's Points 1-5).

Point 1 (closed)

R2:

There is a point R1 made which matched some of my comments before, and as such I would like to address some of the author's response to R1, and mix it in with some comments from my review (page 2-5, 13): I agree with reviewer 1 about taking caution in the novelty in this advance. As pointed out by R1, some of the language used in the manuscript (and response to reviewers here) is too strong. I have pointed out some aspects of this throughout my second review.

Authors:

We clarify that R1 never suggested we should "[take] caution in the novelty of this advance"; their concern was with whether previous paleontological work would be disregarded as a result of our statements, stating "I caution the tone of the introduction in challenging previous work on range of motion" (R1 round 1). In response, we already took efforts to ameliorate this concern in the previous round of review by making clarifying phrasing changes in our previous submission's Lines 51-52, 64, 158. R1 expressed satisfaction with these revisions."

R4:

These comments refer to R1's previous round of comments. If R1 is now satisfied, I think this matter should be considered closed – unless, R2 expresses specific concerns re: the revised manuscript that has been submitted.

Thank you. We agree. This matter is closed.

Point 2 (closed)

R2:

Page 2: this is tangential to the paper, but the earliest ROM reconstructions from skeletal material were completed decades before 2010. For example, the work on masticatory biomechanics where gape angle is estimated from skeletal collections (e.g., Bill Hylander, Callum Ross, Chris Vinyard, Elizabeth Dumont). These studies generally investigated mandibular rotation (i.e., opening the mouth and twisting about the superior/inferior axis) and translation of the TMJ (usually in an anterior/posterior direction).

Authors:

We thank the Reviewer for pointing this out; we are well aware of this and have written on this previously (for example, in the review by Manafzadeh & Gatesy, 2022 ICB). ROM reconstruction from bones has been going on for centuries, and quantitatively at least as early as the early 1900s, far before the masticatory examples cited here (see Manafzadeh & Gatesy, 2022 ICB for specific citations). The 2010 date stated in our response – it does not appear in the manuscript – was in specific reference to computational, virtual ROM reconstructions, as made clear by our existing phrasing in the response: “The earliest virtual ROM reconstructions (e.g., Mallison, 2010; Pierce et al., 2012)” (emphasis added here).

Manafzadeh, A. R., & Gatesy, S. M. (2022). Advances and challenges in paleobiological reconstructions of joint mobility. *Integrative and Comparative Biology*, 62(5), 1369-1376.

R4:

This point is acknowledged as tangential to the paper, and not part of the manuscript. I suggest that this matter should be considered closed.

Thank you. We agree. This matter is closed.

Point 3 (closed)

R2:

Page 4: the idea that translation and rotation of a joint must be considered together, simultaneously, is not new to (paleo)biomechanics. In addition to the mandible example (discussed above) this has also been considered at length in the human knee.

Authors:

We humbly suggest that this is, contrary to the Reviewer's statement, a very new idea to paleobiomechanics and computational ROM reconstructions. Although the mandible examples discussed above as well as studies on numerous other joints have cursorily alluded to or to some extent investigated both rotation and translation, a systematic and reproducible study algorithmically investigating combinations of rotations and translations had never been conducted in paleobiomechanics.

This is why we were both motivated and able to publish an entire paper on the subject two years ago (Manafzadeh & Gatesy, 2021 J. Anat.), highlighting the shortcomings of previous approaches that had not engaged with combinations of rotation and translation as systematically as necessary to capture biological reality. Subsequent studies considering systematic combinations of rotations and translations (e.g., Bishop et al., 2023 Methods in Ecology and Evolution; Wiseman et al., 2022 IOB) have explicitly drawn inspiration from our advance.

Bishop, P. J., Brocklehurst, R. J., & Pierce, S. E. (2023). Intelligent sampling of high-dimensional jointmobility space for analysis of articular function. *Methods in Ecology and Evolution*, 14(2), 569- 582.

Manafzadeh, A. R., & Gatesy, S. M. (2021). Paleobiological reconstructions of articular function require all six degrees of freedom. *Journal of Anatomy*, 239(6), 1516-1524.

Wiseman, A. L., Demuth, O. E., Pomeroy, E., & De Groote, I. (2022). Reconstructing articular cartilage in the Australopithecus afarensis hip joint and the need for modeling six degrees of freedom. *Integrative Organismal Biology*, 4(1), obac031.

R4:

I agree with the authors that translation and simultaneous rotations have, at the very least, traditionally been underused in estimating ROM of extinct animals. Until recently (and including studies that continue to be published), it seems usual to explore single rotational degrees of freedom at a time. Translation may or may not be explored. Because this has been the norm in our field, I think emphasising the idea that translations and rotations need to be considered together in a reproducible and objective way is defensible, even if it is not a new concept.

Thank you. We agree. This matter is closed.

Point 4 (closed)

R2:

Page 4/5: I agree with the reviewers that a) the inclusion of a six DoF joint into biomechanical models and b) the inclusion of more complex joints into paleontology is unique (for some subfields), but agree with the reviewer that the authors do not have the data to make such authoritative statements.

Authors:

The Reviewer has repeated the same concern within this round of review. As noted above, we clarify that R1's only concern expressed in Pages 4/5 was with whether previous paleontological work would be disregarded as a result of our statements. In the previous round of review we already clarified our phrasing in our previous submission's Lines 51-52, 64,158 to be as explicit as possible about the scope of our advance and to better prevent this unintended interpretation.

We reiterate that R1 expressed satisfaction with these revisions.

R4:

As with Point 1, these comments appear to primarily refer to R1's previous round of comments. If R1 is now satisfied, I think this matter should be considered closed – unless, R2 expresses specific concerns re: the revised manuscript that has been re-submitted.

Thank you. We agree. This matter is closed.

Point 5 (closed)

R2:

Page 5: the author's express a desire to have the response to the reviewers included in their manuscripts, and I agree, such information is needed in this one. This was one of the big issues I had with this manuscript the first time I read it: I do not believe Nat Comms provides the space necessary for the authors to properly discuss the topic or methodology due to space limitations of the journal.

Authors:

Appropriateness for the journal is at the discretion of the Editor, who has communicated that our manuscript is a good fit for publication in Nature Communications.

R4:

The reviews are routinely available/published, I understand, due to the Journal guidelines, which sounds like it would satisfy all parties. From the reviewer email: "Nature Communications uses a transparent peer review system for published work, and therefore publishes reviewer comments and the authors' rebuttal alongside the paper. By submitting a reviewer report you agree to the publication of the comments made to the authors (any confidential comments to the editor will not be published)."

My understanding is that the appropriateness of the Journal (in terms of space limitations/formatting) is of Editorial rather than reviewer concern. Appropriateness due to content/subject matter would be of reviewer concern.

Thank you. We agree. This matter is closed.

Point 6 (change made)

R2:

Page 13: I understand the author's feel I am being overly pessimistic about the generalizability of this metric, but, unfortunately, the author's have not offered sufficient evidence to the alternative, and my concerns are supported by the evidence presented in supp. Fig 9 (with regards to the glenohumeral joint). My issue was not whether or not you could use the raycasting method on other joints (of course you can!), it's about whether the results will be useful. The authors have expanded their limitations discussion in the manuscript, which is appreciated, but I believe you should have a better understanding of the limitations of a method in a manuscript which is stating, quite conclusively "this is the way forward." This speaks to the "dual nature" of the manuscript: one being the reconstruction of the pedal pose of an extinct animal, and the other presenting the new method. It is, essentially, two studies wrapped into one, and being presented in a short communication.

Authors:

We are gratified to learn that the Reviewer does understand that our raycasting approach can easily be applied to other joints. As stated in our Discussion since our original submission: "comparing articulation score distributions against in vivo kinematics for further joints and taxa will illuminate the broader utility of articulation analysis, facilitating the widespread application of these data in vertebrate functional reconstruction" (previous submission Lines 172-174).

In stating this, we have made clear to the reader that the only way to determine "whether the results will be useful" (Reviewer's phrasing) is through comparisons with in vivo data that are specific to the research question at hand. We cannot possibly test the full diversity of vertebrate joints in one manuscript, and neither we nor the Reviewer can evaluate how "useful" articulation analysis will prove for the reconstruction of any particular behavior at any particular joint without conducting such tests. However, our paper demonstrates the potential great utility of this approach using our selected case study, while also laying the necessary groundwork in conceptual and methodological advances for its future expansion.

To the Reviewer's concern about the glenohumeral joint in Supplementary Figure 9, then, we emphasize that the exemplar configurations we have displayed will successfully receive different articulation scores – we have now added quantitative articulation subscores to the Supplementary Figure 9 caption to make this as clear as possible – and determining how "useful" these differences in score are will rely on future collection of and comparison with in vivo shoulder kinematics.

Dedicated analysis of the human glenohumeral joint is far beyond the intended scope of this manuscript and this example was added to assuage this Reviewer's concerns in the previous round about the broader generalizability of this approach, an addition which is hopefully also of interest to the journal's broad readership. As of interest to the Reviewer, we offer the following screenshots of extremes of in vivo glenohumeral rotation kindly provided by an orthopedic biomechanics colleague to emphasize that in a wide range of rotational poses, joint translation is limited such that articular congruence (informally assessed here based on apparent matching of joint surface curvature) appears to stay much higher than in our low-scoring 6 dof configuration figured in Supplementary Figure 9. This would tentatively suggest that raycasting analysis of this joint will in fact have future value for functional reconstruction of this joint, though

we emphasize the need for thorough, formal articulation analysis and comparison with in vivo kinematics. If the Reviewer has a particular interest in the human glenohumeral joint, we hope they will implement our new approach to investigate shoulder articulation in more detail!

To concern about the “dual nature” of our manuscript: the Reviewer has repeated the same concern within this round of review. We again, as above, defer to the judgment of the Editor, who has communicated that our manuscript as written is a good fit for Nature Communications.

R4:

My reading of the manuscript is that this method is being proposed as a way forward (rather than the way forward), the need and usefulness of which is being presented to be critiqued and tested by others. This intention is clarified by the authors’ response above “we hope they will implement our new approach to investigate shoulder articulation in more detail”. Perhaps the back-and-forth between R2 and the authors indicates that this intention could be made clearer in the manuscript for the benefit of other readers who may similarly interpret the paper’s offering (“why not this?”) as imperative (“you must do this”).

I do think there is some concern in developing the method and validating/ground-truthing it against predominantly one joint (the ankle of guineafowl), then applying the method to different joints (IP & MTP joints of Deinonychus; see my comments/feedback on the manuscript itself). This concern may be separate from R2’s point; it’s a little difficult to tell – apologies if so.

But as a short communication pitching a new method and its application, or the concept of a dual manuscript, it not problematic in of itself. The authors could choose to do a longer form manuscript and/or split it into two studies with multiple joint/joint types. Indeed, it may make the conclusions more robust and “sell” the method more effectively to do so. But in my opinion it is the authors’ choice how to put this manuscript into the world and receive scrutiny from others working in the field. Carefully caveated (e.g. clear methods signalled, as in my comments/feedback on the manuscript) and perhaps some care with the language describing impact/previous study, I see no reason why the short communication or combined method + application is inappropriate.

Thank you. We agree.

Reviewer 4’s specific concerns about validation are separate from Reviewer 2’s and are therefore addressed in new Point 28, below.

To resolve Reviewer 2’s Point 6, we have made edits to phrasing in two regions of the Discussion to make as explicit as possible that this work represents only a first attempt at establishing articulation analysis as an area of study and that we hope future workers build from our foundation, in Lines 147 and 201-202.

Point 7 (closed)

R2:

Page 9: I thank the authors for their clarification, there was indeed a miscommunication. Pose certainly does need to be defined, and early on in the manuscript (but see comments on this manuscript below). In its current form, joint poses (line 51), is now defined as rotational combinations of joint articulations, which I do not believe is what the author's mean the definition to mean. I believe a proper definition would be something closer to "position of body segments, relative to each other, that are connected by a joint". This is closer to the definition given in L40-41 for a different term (joint configurations). As an aside, the authors also misunderstood my comment, as I was not suggesting in my comment that motion in animals was deterministic (and neither is motion in robots, not the least bit because of the inclusion of AI and machine learning, but also because of things like vibrations and inertial momentum, making precision manufacturing by robots difficult/impossible today 😊)

For the title, I would recommend changing "locomotor joint poses" to "joint poses during locomotion" for clarity.

Authors:

The Reviewer is repeating a concern from the previous round of review. We clarify that we have defined joint poses as combinations of excursions in all three rotational degrees of freedom (not "rotational combinations of joint articulations;" we are unsure what combinations of articulations would mean and therefore do not use this language). Likewise, we have defined joint configurations as combinations of excursions in all six (all three rotational and all three translational) degrees of freedom. These definitions were added explicitly to the manuscript in last submission's Lines 40-41 and Line 51 during the last round of review to increase clarity. We re-emphasize as discussed in the previous round of review that there is no "proper definition" for joint poses, and that different fields use the word pose differently – we offered several citations to this effect in the previous round of review, and repeat that response here:

the word "pose" has been used in the orthopedic, anthropological, paleontological, zoological, computer animation, and biomechanical literature to refer to everything from a joint's excursion in a single rotational degree of freedom (e.g., Senter, 2009), to a joint's excursions in all three rotational degrees of freedom (e.g., Manafzadeh & Padian, 2018; this paper), to a joint's excursions in all six degrees of freedom (e.g., Bishop et al., 2023) to the combination of joint "poses" (defined as any of the previous) throughout a limb (e.g., Lauer et al., 2022) or even an entire individual/character.

Bishop, P. J., Brocklehurst, R. J., & Pierce, S. E. (2023). Intelligent sampling of high - dimensional joint mobility space for analysis of articular function. *Methods in Ecology and Evolution*, 14(2), 569-582.

Lauer, J., Zhou, M., Ye, S., Menegas, W., Schneider, S., Nath, T., ... & Mathis, A. (2022). Multi-animal pose estimation, identification and tracking with DeepLabCut. *Nature Methods*, 19(4), 496-504.

Manafzadeh, A. R., & Padian, K. (2018). ROM mapping of ligamentous constraints on avian hip mobility: implications for extinct ornithomirans. *Proceedings of the Royal Society B: Biological Sciences*, 285(1879), 20180727.

Senter, P. (2009). Pedal function in deinonychosaurs (Dinosauria: Theropoda): a comparative study. *Bulletin of the Gunma Museum of Natural History*, 13, 1-14

Therefore, other readers would certainly disagree with both us and this Reviewer about their intuition for the “proper definition” of “pose,” and it is most important for clarity and accessibility of our work that we are explicit about what we mean by this word for this specific manuscript.

Again, this is a change we already implemented in the previous round. We already changed the title of our submission to “...locomotor joint poses” in the last round of review in response to this Reviewer’s concerns that “...locomotor poses” did not clearly enough reflect that our definition of pose was on a per-joint basis. The title the Reviewer now suggests is fully synonymous with our current title and purely a difference of semantic preference, so we respectfully decline the suggested change. Instead, to increase clarity as fully as possible, we have added a parenthetical to the Introduction in new Line 66 (joint poses actually used during terrestrial locomotion (i.e., locomotor joint poses)), explicitly drawing the connection the Reviewer requests, and have changed two instances of “locomotor pose” to “locomotor joint pose” (new Lines 85 and 89) in the Results for better consistency with the existing title.

R4:

The manuscript’s current use/definition of ‘joint pose’ and ‘locomotor joint pose’ is both clear and appropriate, in my opinion.

Thank you. We agree. This matter is closed.

Point 8 (closed)

R2:

Page 11: In response to there being no other methods present, It does not need to be another 6 DoF method, it just needs to be another method for estimating joint poses, and the authors have actually discussed an alternative methodology previously in this review (i.e., the “prism method” for estimated ROM). While imperfect, it would be useful to see how results from that method, or the “qualitative method” of joint poses discussed later in the response to R1, compared to results from this method. If results are similar in terms of locomotor reconstruction (forward dynamic simulations), then does this more complicated ray casting method of estimation actually need to be used? If not, it creates a good argument for using this model (which has its own inaccuracies) over other models. I stand by my earlier comment that a comparison is needed to provide tangible evidence for how this method is superior and provides superior estimates for extinct animals.

Authors:

The Reviewer is repeating a concern from the previous round of review. The Reviewer, in the previous round, stated “But the paper neither presented a method for evaluating joint articulation or compared this method to others in terms of reconstructions (showing that this method is more accurate),” so their desired comparison was of methods for “evaluating joint articulation.” We reiterate, as stated in our previous round of responses, that there are no other reproducible methods in existence to accomplish the comparison that the Reviewer requests.

The Reviewer now raises suggestions for two potential “alternative methodolog[ies].” First, they bring up the “prism method.” We invented the prism method two years ago (Manafzadeh & Gatesy, 2021 J Anat). The prism method in no way evaluates articulation. It is simply a method for sampling joint translations in hinge joints. In fact, we already implement the prism method in the present study when sampling 6 dof joint configurations, and then we evaluate articulation at each viable (non-interpenetrating) joint configuration using our new raycast-based approach. This has been explicit since the initial submission in the Methods section, where we state:

“At each rotational pose, 343 potential translation combinations were allowed ($7 \times 7 \times 7$ translations sampled using the prism-based hinge joint translation method proposed by [26])” (new Lines 245- 247)

and

“We then conducted an articular raycast at each viable (i.e., non-interpenetrating) configuration.” (new Lines 256-257).

Manafzadeh, A. R., & Gatesy, S. M. (2021). Paleobiological reconstructions of articular function require all six degrees of freedom. *Journal of Anatomy*, 239(6), 1516-1524.

Second, they bring up the “qualitative method ... discussed in the response to R1.” The qualitative method of evaluating joint articulation is just that – qualitative – and is therefore subjective based on the interpretations of the specific researcher implementing the method. That said, we already include the results of this method as applied to the foot of *Deinonychus* as

Supplementary Figure 8; we added this in the last round of review in response to a different comment by this same Reviewer. It appears the Reviewer may have missed this addition. We reiterate our response from the last round:

The most quantitative hypothesis of Deinonychus joint motion to date was published by Senter (2009), who relied on manual manipulation and subjective assessment of proper articulation in an attempt to identify each joint's full ROM (critically, not its locomotor ROM, which is a subset of full ROM [see Manafzadeh et al., 2021]). Notably, Senter's hypotheses sometimes underestimate even the locomotor ROM we recover here!

To make more explicit in the paper that we are contributing new data to our understanding of Deinonychus, we have created new Supplementary Figure 8 to directly compare our results with Senter's. Therefore, we posit that the raycast-based approach we use here is not "more complicated," but that it is the only objective method currently available to the field for quantitatively evaluating joint articulation. We cannot claim that our method is "superior," nor do we intend to – we intend to provide the first and only method for accomplishing this goal. We are indeed hopeful that this will serve as the foundation for the future development of methods superior to ours, such that future comparison is possible!

R4:

My understanding/awareness of previous methods is that they give an idea of 'possible' vs 'not possible' poses, i.e. binary yes/no. One could compare these ROM estimates to the ROMs actually used in locomotion – and in fact the manuscript does this, in Fig 2b.

This manuscript's method, using articulation score, proposes a quantitative refinement of the binary yes/no.

In this sense, I would tend to agree with the authors. Although previous methods have made substantial efforts towards objectivity, these methods are not quantifiable in the same way as the articulation score (to my knowledge & understanding). Given the caveat above, I personally can't see how a meaningful comparison could be made.

Thank you. We agree. This matter is closed.

Point 9 (closed)

R2:

If higher articulation scores do not represent stability, what do they biologically represent? This needs to be explicitly stated in the manuscript, to prevent others from making the same mistake I have. A definition of articulation early on would clear up some of this confusion, as the authors are using terms like “disarticulation”, “misalignment”, and “subluxation,” which compromise the stability of the joint, to explain improper articulation, but are also stating that articulation scores (and thereby, articulation) do not represent stability.

Authors:

The Reviewer is repeating a concern from the previous round of review. In the previous round, the Reviewer stated:

“The reason I say the paper did not present a method for evaluating joint articulation is that it provides a method for quantifying the likelihood of a joint engaging in a pose based on a measure of joint stability, but this does not quantify articulation. Nowhere in this metric can you determine if a joint is articulated or not, just if a joint is likely to engage in a pose during walking/running in a straight line.”

In response to this miscommunication of our intent, we already made the goal of our articulation analysis (“measuring the quality of joint articulation for any viable, static six-degree-of-freedom joint configuration”) more explicit in Lines 60-62 of the previous submission. Also in response, we also already clarified in our Methods (previous submission Lines 257-264) that:

“The articular raycasting approach we propose here aims to capture information about the morphological relationship of a pair of mating articular surfaces in any viable (i.e., non-interpenetrating), static six-degree-of-freedom joint configuration. Conceptually, our approach provides data about the relationship between these surfaces’ 3-D curvatures. We select this emphasis here because interactions between articular surface curvatures are fundamental to joint function (see [36, 64]). The relationship between the curvatures of mating articular surfaces in any given configuration dictates the paths of minimum work along which joints habitually move, as well as the capacity of a joint to effectively distribute and evenly transmit loads.”

And later in lines 280-287 that:

“We propose that the formation of a successful ray hit means that articular curvatures are aligned such that there is a capacity for meaningful biomechanical interaction between the pair of articular surfaces in vivo. Therefore, if even a single ray hit successfully, we assigned the joint configuration an articulation score of greater than zero (see formula, below). If no rays succeeded in hitting the mating articular surface, we considered the joint to be unscorable, and gave the joint configuration an articulation score of zero. Articulation score was then calculated using three parameters – overlap, symmetry, and congruence – each of which receives an individual subscore from 0 to 1.”

These sections of text already explicitly address our understanding of the biological meaning of joint articulation.

We disagree that “disarticulation,” “misalignment,” and “subluxation,” necessarily communicate something about joint stability. These words, like “pose,” are highly fraught, having been taken to mean very different things in different papers across different fields. It is possible the Reviewer has encountered them largely in the context of stability, leading to their impression, but in the absence of any citations provided by them we cannot be sure. We reiterate that the word “stability” has never appeared, and still never appears, in our manuscript. We also already acknowledged in the previous round of responses that: “we do think the Reviewer’s general intuition is correct, and it is reasonable that joint configurations with a high articulation score may be those that are more stable. However, because such a claim would need to be tested experimentally and because making such a claim is not our goal, we avoid using these terms.” (Emphasis added here.)

R4:

Stability, in the sense used by R2, is a bit of a woolly term that could mean different things to different people. To me, stability depends equally on the surrounding muscles, ligaments, menisci etc as it does the bony morphology, and implies specific loading regimes. I disagree that the articulation score, as defined in the manuscript, makes implications about a joint’s stability: different joints (ball- and-socket, condylar, planar and so on) all have different inherent stabilities but I suspect that they would have similar articulation scores / ranges of articulation scores.

In my opinion, this confusion is not an issue in the current manuscript; articulation score is not implied to represent ‘stability’. It seems clear enough to me that articulation score is a ‘plausibility’ measure of pose occurrence in in vivo locomotor behaviours.

Thank you. We agree. This matter is closed.

Point 10 (change made)

R2:

While the authors state that the articulation metric is not probabilistic, they are treating it like a probability in their analysis. I.e., scores above 95 were used further means the probability that scores less than 95 would be engaged in is low. They also discuss it in a probabilistic manner in the manuscript (see comments about current manuscript below). They are not using it as a 95% CI, but they are using it as a probability (i.e., the probability that the joint will be in that pose, due to an articulation score which has been validated with a subset of joints from two species)

Authors:

The Reviewer is repeating a concern from the previous round of review. Again, we have already addressed this concern in the previous round: “we clarify that this metric is not probabilistic, nor does it attempt to highlight “likely” poses. Articulation score can be calculated for a pair of bony articular surfaces in any viable, static six- degree-of-freedom configuration. What we find here is that the joint poses used during locomotion happen to consistently fall in the highest-scoring region of pose space -- but we find this using experimental data; the score itself contains no measure of probability and is not inherently tied to locomotion in any way. We again hope the new Supplementary Text and Figures we have added will make this more clear.”

We are certainly identifying a consistent correlation in extant animals and applying it to new data from extinct animals within a phylogenetically and morphologically informed framework. Perhaps we differ with the Reviewer in terms of our understanding of the meaning of “probabilistic,” because if this general approach is viewed as probabilistic, we would argue that all application of data from extant animals to the fossil record is similarly inherently probabilistic – this is the foundation of modern functional morphology in paleontology. Regardless, the Reviewer does not suggest here that they view this approach as problematic, even if “probabilistic,” nor do they propose a constructive path forward for revision.

R4:

In my reading of the current manuscript, I didn't see anything that claims a score of 95+ implying the likelihood (as a percentage) that the pose is used. Semantically, I suppose that low-scoring poses are less likely (less often) used in locomotion, and one could argue that they should/could be eliminated when reconstructing animal movement. But I don't see an issue with that logic, accepting that some in vivo poses did fall below 95 score and so would be incorrectly eliminated. The probability is discussed in a qualitative way, which to me did not seem problematic.

I do think the manuscript would benefit from more clarity on how the 95 score cutoff was chosen, and how future researchers can decide articulation score cut-offs for their own datasets.

Thank you. We agree. Reviewer 4's suggestion for explanation about the 95 score cutoff selection for *Deinonychus* was to some extent already present at the end of the Methods section; we have now expanded this text for clarity, and have reemphasized the need for identification of new score thresholds for other studies in Lines 408-415. We also now

clarify throughout the text in multiple locations that our *Deinonychus* stride cycle is intended to be a walking stride (e.g., 28, 133, 395).

Point 11 (change made)

R2:

Page 12: The authors write “The score simply offers a quantifier of the quality of joint articulation for any viable, static six-degree-of-freedom configuration of mating bony articular surfaces – not “joint ROM”.” The authors are, in fact, quantifying joint range of motion based on their articulation scores: high articulation scores = inclusion in ROM, low articulation scores = exclusion. In their study, ROM is defined as having an articulation score of 95+.

Authors:

The Reviewer is repeating a concern from the previous round of review. We reiterate our previous statement that we are not quantifying range of motion based on our scores – the Reviewer has kindly quoted that previous statement for us. Our score offers information about the quality of joint articulation across all of joint pose space. We have identified that locomotor poses consistently fall within the subset of pose space with articulation scores of 95-100. The score itself does not quantify range of motion. The subset of pose space identified through articulation score does not reflect the joint’s full range of motion, but rather a locomotor subset of range of motion.

This initially came up because the Reviewer stated in the previous round of review that “What I believe the paper is about is a method for quantifying joint ROM based on a measure of stability.” We have already clarified in several ways why this belief is incorrect – we have reiterated our disagreements about both “quantifying joint ROM” and “a measure of stability,” already discussed once in the last round of review, here and above.

R4:

It seems to me that there is scope for misunderstanding on an important point of the manuscript. The authors clarify here “The subset of pose space identified through articulation score does not reflect the joint’s full range of motion, but rather a locomotor subset of range of motion.”

This is worth making more clear in the manuscript, particularly early on and for parts I have commented separately in my own feedback as being general/minimising of prior work. I think it would be easy to take a similar interpretation as R2 does from the manuscript alone – and I similar initially misunderstood until reading the ‘Response to reviewers’ – that the raycasting method is not primarily a method to more objectively estimate ROM, but rather that refining the ROMs likely/commonly used by a joint during locomotion.

I feel that the generalisations in the manuscript which I mention in my own comments/feedback are working against the authors’ intentions, by making it sound as if the method is about general ROM. Even if later clarified to be more specifically about locomotion, this can be missed by readers.

Thank you – this comment has helped us to better understand the source of some of Reviewer 2’s misunderstanding and to better frame our Introduction to clarify the goal of our study for the reader. Reviewer 4 is absolutely correct that we do not seek to

reconstruct full potential range of motion in this study, and that this should be made more clear early on in the manuscript.

We have removed all references to ROM in the Abstract and Introduction, and accordingly made several phrasing changes throughout those sections (e.g., Lines 16-17, 40-41, 64), to more set up more targeted and precise context for this study.

Point 12 (change made)

R2:

With regards to the whole-foot combinations of joint poses, part of the reason I used the words “opinion” and “random” is due to the writing of the author’s on L290 (L362 in the resubmission). The authors write “An inverse kinematic animation rig was used to animate a rough stride cycle for the pes of Deinonychus using inspiration from published extant avian toe tip motion” To me, this sentence comes across as the author’s constructed a model of extant avian toe tip motion, and then – combined with the joint poses – used it as inspiration to create the combined joint poses presented in the paper. In this case, inspiration would be similar to an artist using a set of trees in a forest as inspiration to create one, composite tree. In this case, the composite tree is influenced by the author’s opinion heavily, with no necessary protocol (i.e., random) as to why some features were chosen, and others were not. If this is what the author’s did, then my comment stands true. If not, then this again speaks towards not enough detail being given in the paper for the study to be properly explained or replicated.

Authors:

The Reviewer is repeating a concern from the previous round of review. They include our text that “An inverse kinematic animation rig was used to animate a rough stride cycle for the pes of Deinonychus using inspiration from published extant avian toe tip motion.” We pointed out in the previous round of review that this is common practice in the field of paleontology, stating in the last round that:

Here the hypothesis of covariation we chose to employ, as stated in Line 362, is based on that of avian feet. This is one reasonable hypothesis for non-avian dinosaur intra-pes coordination and is regularly applied by dinosaur paleontologists (e.g., Falkingham & Gatesy, 2014; Gatesy et al., 1999).

We have now edited our text in new Lines 361-365 to include reference to Falkingham & Gatesy (new reference 70) and to make more reproducible the specific way in which this inspiration was implemented in the current study. We reiterate that such practice is regularly applied in paleontology and that the goal of the present study was to reconstruct one of many possible well-supported stride cycles for Deinonychus, reminding the Reviewer as we discussed extensively in the previous round that natural variation is to be expected in the joint poses and configurations used in locomotor stride cycles. The edited text is as follows:

An inverse kinematic animation rig was used to animate a rough stride cycle for the pes of Deinonychus using inspiration from published extant avian toe tip motion by importing published videos and reconstructed animations of avian toe tip motion into Maya and aligning the metatarsus and distal phalanges of Deinonychus to the avian metatarsus and distal phalanges [69-70].

We also re-emphasize to the Reviewer that this inspiration was used simply to generate a starting point, first-pass “rough stride cycle” that was then modified into alignment with our articulation data, as explicitly described in the remainder of the Methods section.

R4:

I don't know enough about inverse kinematic animation or forward kinematic animation to critique this as a 'starting point'. But the description of the method seems justifiable for readers to judge and critique plausibility of the stride cycle for themselves, in my opinion.

The manuscript could ameliorate concerns by flagging this as 'starting point' more clearly in the Results (I appreciate this is covered in Methods, but this comes later in the manuscript due to the Journal format, and due to brevity could be missed). I think it is important to be clear that the starting point for the extinct dinosaur's locomotor pose 'reconstruction' was an existing kinematic dataset of bird locomotion stride cycle.

I would also query whether it could be considered circular (or otherwise self-reinforcing) if the articulation scores are validated/ground-truthed from a bird's locomotor kinematics (predominantly guineafowl), and the starting point *Deinonychus* kinematics are also derived from birds (presumably also guineafowl, although it's not stated in the manuscript which bird 'inspires' the *Deinonychus* stride cycle)? Does this really tell us something new about *Deinonychus*, or are these essentially *Deinonychus* bones animated with guineafowl kinematics?

Is there a method of doing this first step without the bird kinematics, to alleviate concerns of circularity and/or prior assumptions of guineafowl-like stride kinematics?

We agree that we can do better to emphasize the use of avian data as a starting point in the Results section. We have now added this information to the Results in Lines 130-131 as requested.

Regarding potential concerns about the circularity of this work, we emphasize that our goal is to reconstruct *one potential walking stride cycle for *Deinonychus (as stated in Line 169). Therefore, although additional variant stride cycles could certainly be generated without an avian starting point as the Reviewer suggests, the full cycle we present here is one possibility that is supported by *Deinonychus* articulation data for all joint configurations displayed at all twelve pedal joints. The avian data used here serves only to offer one viable example of pedal joint coordination to begin from, but we emphasize that the resulting configurations included for *Deinonychus* are supported fully based on the articular morphology and articulation score distributions of *Deinonychus* alone, applying the articulation score cutoff we identified through our *in vivo* data comparison. As the Reviewer points out, we have presented the necessary information for readers to critique the plausibility of this stride cycle themselves.**

Point 13 (closed)

R2:

Page 13: as stated above, the authors have misunderstood my comment. It is not whether this method can be applied to all joints, but rather if the results would be useful or not. From supplementary Figure 9, it certainly seems it would be less than useful for the human glenohumeral joint, as it appears some positions that are subluxations would come across as having high articulation scores, which is a fundamental shortcoming of this method.

Authors:

The Reviewer has repeated the same concern within this round of review. We have already responded to this concern extensively above, but restate here that the exemplar configurations we have displayed will successfully receive different articulation scores – we have now added quantitative articulation subscores to the Supplementary Figure 9 caption to make this as clear as possible – and determining how “useful” these differences in score are will rely on future collection of and comparison with in vivo kinematics.

R4:

In my opinion, this point is not relevant to this paper. Determining utility for other joints and species depends on future research for the species and joints in question. It would be a huge study needed to claim that the raycasting/articulation scoring method was valid for all joint and species. The authors are not making this claim.

Thank you. We agree. This matter is closed.

Point 14 (closed)

R2:

Now, this implies one of two things

- 1) this is a random correlation, in which case it is biologically meaningless
- 2) this is a causative relationship, whereby articulation scores are related to joint motion (rotation and translation)

The authors believe this is a causative relationship (e.g., see wording in abstract, discussed below), which is why they can use it to reconstruct extinct vertebrate locomotion. Assuming this is true, my question from before still stands: what is the underlying logic and biomechanical meaning of ray casting? I.e., why, functionally, was ray casting work? It is touched on in the paper, but not explained. E.g., why does asymmetry in ray length matter? Why does having more rays connecting the two surfaces when there is space between them biomechanically matter? Otherwise it appears to be more-or-less arbitrary why ray casting was chosen, and why it works in this paper.

Authors:

The Reviewer has repeated the same concern within this round of review. We reiterate, as above, that we already implemented substantial revisions during the previous round in response to this Reviewer's concerns about this. Again, in our revised Methods (previous submission Lines 257-264) we stated that:

"The articular raycasting approach we propose here aims to capture information about the morphological relationship of a pair of mating articular surfaces in any viable (i.e., non-interpenetrating), static six-degree-of-freedom joint configuration. Conceptually, our approach provides data about the relationship between these surfaces' 3-D curvatures. We select this emphasis here because interactions between articular surface curvatures are fundamental to joint function (see [36, 64]). The relationship between the curvatures of mating articular surfaces in any given configuration dictates the paths of minimum work along which joints habitually move, as well as the capacity of a joint to effectively distribute and evenly transmit loads." (emphasis added here)

And later in lines 280-287 that:

"We propose that the formation of a successful ray hit means that articular curvatures are aligned such that there is a capacity for meaningful biomechanical interaction between the pair of articular surfaces in vivo. Therefore, if even a single ray hit successfully, we assigned the joint configuration an articulation score of greater than zero (see formula, below). If no rays succeeded in hitting the mating articular surface, we considered the joint to be unscorable, and gave the joint configuration an articulation score of zero. Articulation score was then calculated using three parameters – overlap, symmetry, and congruence – each of which receives an individual subscore from 0 to 1." (emphasis added here)

These sections of text already discuss our explicit justification for selecting raycasting, a choice which was not arbitrary, and explain its biomechanical meaning.

R4:

In my opinion the manuscript, as quoted here by the authors, gives sufficient explanation for why raycasting was chosen.

Further, even if the articular scores are “randomly correlated” with in vivo locomotor pose use, as long as that correlation can be proved reliable (particularly in future studies of other joints/species), the method can still be useful even if it’s biological meaning is not fully understood.

Thank you. We agree. This matter is closed.

Point 15 (change made)

R2:

Page 16: If the point of this method is to create an envelope of possible joint poses, the authors do not need to identify what percentage of sampled poses fall within the 95-100 articulation score envelope, but rather find the articulation score envelope that fits 100% of the joint poses. Otherwise, the authors are contradicting themselves (or believe 1.9% of the in vivo data is incorrect). I stand by my comment that articulation scores from 95-100 should not be used, but rather this window needs to be calculated from the in vivo data, and then used to re-reconstruct extinct animal locomotion.

Authors:

We note that we calculated and included the percentage of measured in vivo locomotor poses that fall within the 95-100 envelope in direct response to a request from this Reviewer. We think this is a useful addition that has improved our manuscript and thank them for the motivation. The Reviewer is repeating a concern from the previous round of review. A goal of this study was to reconstruct one of many possible locomotor stride cycles for the extinct animal *Deinonychus* using support from joint articulation data. Because ALL avian walking data fell within the 95-100 envelope, we maintain our judgment that it is reasonable to use this range to reconstruct one of many possible walking stride cycles for *Deinonychus*, even if a small number of the poses used during swing phase of highest- speed avian running were found to exit the the 95- 100 score range.

R4:

The authors' judgement for using the 95-100 window does seem reasonable. However, it could be made more clear why the 95-100 score envelope was chosen in the first place. I can see R2's point that it would be useful to define the articulation score cutoff/window from the experimental data (it was only because of R2 and careful re-reading of the manuscript that I realised the 95-100 window is not defined by the locomotor experimental data as I had assumed).

If doing so creates an unhelpfully large window, could other weightings of the combined articulation score 'fit' the data better?

As discussed in response to Point 10 above, we have added additional information regarding how we selected our 95-100 window to the text in Lines 408-415 for clarity, and have emphasized that the selection of thresholds will differ for future studies.

Along the same lines, we suggest that if a study aims to reconstruct a dinosaurian running stride in particular, then it would have to expand its score threshold – and as the Reviewer suggests, other weightings might be necessary to fit the data in a meaningful way. However, because we reconstructed a walking stride here, the formula as implemented here met our needs.

Point 16 (change made)

R2:

Page 17: “We contend that it would not change the reconstruction, because these poses would not be ruled out even if the envelope were expanded as the Reviewer suggests.” The authors have misunderstood my comment. Of course, the poses would not be ruled out, but new poses that were ruled out before could now be chosen.

Authors:

The Reviewer has repeated the same concern within this round of review. As described directly above, the stride cycle we reconstruct here would not be ruled out, which is what is relevant for the goal of this study.

R4:

Whether or not poses are ruled out of *Deinonychus*' reconstructed stride cycle specifically is less interesting than the wider point R2 asks here (and above with point 15) of how an appropriate score cut-off is decided.

For future studies of other joints or species, how will the authors (or others) decide what is an appropriate cut-off window for reconstructions? The authors' decision to cut off at 95 and only eliminate 1.9% of running poses does seem reasonable. But what is 'reasonable judgement'? What if it were 5% of poses? 10%? What is (or could be) the basis for making the decision? Was there any bias in the kind of joints that fell outside the 95-100 window (mentioned in my comments on the manuscript itself, later)?

The Response does not address this wider point. And related to the my comment on circularity above, it doesn't seem a revelation that using the starting kinematics from bird data and the articulation scores validated against bird data result in a reconstructed stride cycle that is tolerant of more relaxed articulation scores. (If anything, the expanded envelope would result in a less tweaked/more faithful replication of the bird's stride cycle.) I don't see this as a good argument against more clearly defining the articulation score cutoff from the in vivo data. Even if the solution is just clearer acknowledgment within the manuscript that 95-100 was judged by the authors for this specific dataset (joint(s) and species), as balance between including all walking poses and only eliminating a subjectively acceptable minority of running poses.

As discussed in response to Points 10 and 15 above, we have added additional information regarding how we selected our 95-100 window to the text in Lines Lines 408-415, and have emphasized that other studies will require new thresholds. We also now clarify throughout the text in multiple locations that our *Deinonychus* stride cycle is intended to be a walking stride (e.g., 28, 133, 395).

(The specific question here about 'bias' is repeated and addressed in response to new Point 30, below.)

Point 17 (deemed redundant)

R2:

Page 17: I misunderstood the previous read of the manuscript, and had thought the authors had run forward dynamic simulations, using the joint envelopes as boundary conditions. It is now my understanding the authors used the extant data to roughly position the bones of the extinct animal in the correct position, then manually altered joint positions so they fit within this envelope of possible positions. If this is correct, there is certainly a degree of artistic flair in the reconstruction that needs to be discussed, and the sensitivity of these reconstructions (inter- and intra-observer error) needs to be looked at, given this method is being advertised as a “new way forward” (my words, not the authors) for reconstructing joint poses and locomotor poses of extinct animals.

Authors:

The Reviewer has repeated the same concern within this round of review. We addressed this above, and as stated, edited our text in new Lines 361-365 to include reference to Falkingham & Gatesy (new reference 70) and to make more reproducible the specific way in which this inspiration was implemented in the current study. As discussed in the two responses immediately above this one, our goal was to reconstruct one of many possible walking stride cycles for Deinonychus using support from joint articulation data. The aspect of our approach that we suggest is a “new way forward” (we will agree with the Reviewer’s words) is the ability to test a reconstructed stride – no matter how it is generated – using data from joint articulation. Our joint articulation results have, of course, been heavily tested using the sensitivity analyses presented in the manuscript, because that is the extent of the advance we are proposing.

R4:

As far as I can tell, this seems to be related / the same points as raised Point 12, 15 and 16. If so, then the same comments stand as for those points.

We agree; this is addressed in Points 12, 15, and 16.

Point 18 (change made)

R2:

Page 17: “However, the final sentence of this comment...” the authors have misread my comment. What the authors have “ground-truthed” was whether they could put joints in realistic poses. What I was saying what you cannot say your method has led to a more accurate reconstruction of the gait of the extinct animals because 1) it was not compared to results obtained using traditional joint constraints, and 2) it was not compared to how this extinct animal actually moved. If you wanted to use extant models for this purpose, you would have to take the data from one of the extant animals and do a similar reconstruction that you did with the extinct animal (potentially with bones from another individual of that extant species – e.g., another guinea fowl or emu) using both traditional joint constraints and these new constraints (estimated using ray casting) and see which one produces results closer to your experimental data. Again, if both produce similar results, the traditional method is easier and takes less time, so why should we bother with raycasting?

Authors:

The Reviewer has repeated the same concern within this round of review. We extensively responded to the Reviewer’s concerns about comparison of our method above; this comment appears to be a reiteration of the same concern. We repeat the final portion of our response here:

“we posit that the raycast-based approach we use here is not “more complicated,” but that it is the only objective method currently available to the field for quantitatively evaluating joint articulation.

We cannot claim that our method is “superior,” nor do we intend to – we intend to provide the first and only method for accomplishing this goal. We are indeed hopeful that this will serve as the foundation for the future development of methods superior to ours, such that future comparison is possible!”

R4:

As far as I can tell, this seems to be related / the same point as raised in Point 8 (i.e. why not compare to a previous method). If so, then the same comment stands as for Point 8.

Additionally, from my understanding, the approach suggested by R2 seems to me that it would be completely circular. Because of the way the ‘starting point’ would use extant data, ‘reconstructing’ the stride using the kinematics and articulation scores from the same species would (should) result in the starting kinematics being essentially unchanged in the reconstruction. A better suggestion for ground-truthing would be to reconstruct locomotor kinematics a different extant species (i.e. other bird, not guineafowl or emu) with similar joint morphology.

To me, the significance of this concern depends on whether the paper is primarily intended to:

i) propose a method as proof-of-concept, demo an application (the conclusions of which are less important than it being an example of how application could work), and let others validate (play with it until it breaks)

or

ii) conclusively showcase a new, finalised method as the next way forward in the field, illustrated with a robust application generating novel, concrete conclusions

If the latter, then I would tend to agree with R2 that validation against another extant species would be a valuable step to consider including. But I think R2, myself and the authors are probably all seeing the different intentions/purposes of the study, causing disparity of opinion on what is necessary for the manuscript to be published. I do not think it has to be a completely finished, polished and fully validated method to still be a valuable contribution to scientific discourse (i.e. option (i) above is very much of interest to the readership and worth publishing) – as long as this is clearly signalled and communicated in the manuscript. The solution may lie in tweaking the messaging of the manuscript's purpose, rather than necessarily needing additional analyses.

Thank you. We agree.

As discussed in Point 6, we have made edits to phrasing in two regions of the Discussion to make as explicit as possible that this work represents only a first attempt at establishing articulation analysis as an area of study and that we hope future workers build from our foundation, in Lines 147 and 201-202.

In the second part of the Response discussion, R2 and the authors discuss the revised manuscript itself.

Point 19 (change made)

R2:

Abstract: "... locomotor joint poses consistently have high articulation scores. We then exploited this predictive relationship to constrain reconstruction of a pedal stride cycle..." These lines state that it is possible to predict joint poses based on articulation scores, but the authors failed to show (or statistically test) any type of predictive relationship between articulation score and pedal stride cycle in their extant individuals. That is to say, they showed pedal stride cycles were encompassed in the polygonal space created by articulation scores (you can predict the number of legs a zebra has), but never that articulation scores can predict pedal stride cycle (you can predict an species by knowing the organism has four legs).

Authors:

The Reviewer has repeated the same concern within this round of review. We have already responded to the Reviewer's concerns about probability above. We reiterate that the kind of reasoning the Reviewer appears to take issue with is the basis of all application of data from extant animals to the fossil record. We engage in this application here as responsibly as possible, within a phylogenetically and morphologically informed framework. The approach we take is standard to the field of paleontology.

R4:

The main issue here seems to be the manuscript's use of "predictive". It may just be the term "predictive" being perceived differently, as either:

- i) can 'predict' locomotor poses that are more likely to be used by ranking them via an articulation score window (my understanding is yes, that is what is being done here)
- ii) can predict how an animal must move based on articulation scores (no; the method can only refine the window of likely from possible joint poses)

I think this be resolved for both parties, without changing the meaning of the sentence, by removing the term "predictive". (i.e. changing the sentence to "we then exploited this relationship")

As suggested, we have removed "predictive" from Line 28 (and again later in Line 413).

Point 20 (closed)

R2:

L33: I must have missed it the last time – a joint is not an organ Authors:

Many sources would disagree. The Biology of the Synovial Joint book cited in that sentence states that “This unique collection of reviews has arisen due to the belief of the Editors that joints need to be studied as a whole organ.” We align with this viewpoint and would similarly argue that a joint is an organ. This point has no bearing on the paper’s data or conclusions.

R4:

This appears to be a minor point of personal disagreement. But I would agree with the authors that a joint can be considered an organ. Most common definitions of an organ are some permutation of “collection of tissues, as a structural unit, performing a function”, which a joint can be considered to satisfy.

Thank you. We agree. This matter is closed.

Point 21 (closed)

R2:

L40-41: this definition of joint configurations (the excursion of all three rotational and translational DoF) are what the authors stated was “poses” in the response to the reviewers.

Authors:

We cannot identify the claimed location in the response to reviewers. Our definitions of poses and configurations have, to our knowledge, stayed consistent, and are certainly consistent within the manuscript. Poses = 3 rotational dof, configurations = 3 rotational and 3 translational dof.

R4:

To me it seemed clear that ‘configuration’ and ‘poses’ were used synonymously in the manuscript. I don’t see the need to change anything on this point in the current form of the manuscript (if that is what is being suggested by R2 here).

We clarify that they were not used synonymously, but rather each had its own distinct and consistent definition. In any case, we agree that no change is necessary; thank you. This matter is closed.

Point 22 (change made)

R2:

L48-49: much related to gape in primates has been ground-truthed, so this statement is incorrect Authors:

We are certain that the Reviewer's statement is generally true, however, to our knowledge and despite substantial searching, we suggest that it is false in the context of what is discussed in lines 48-49 (joint articulation criteria for virtual ROM analyses). Metrics such as "maximum bony gape," while very interesting, have not involved analysis of articulation between joint surfaces in living animals, which is how we define ground-truthing articulation criteria. This task would require the use of XROMM-derived data or rigid body marker tracking. Given that one of the co-developers of XROMM is an author on this paper, we question whether such analyses have been conducted. Published studies could always exist that we are unaware of, but the Reviewer did not provide specific citations.

R4:

I picked up on this statement (line 48-49) in my own first-pass manuscript comments, concerning generalising/narrowing statements that are made. Because 2/4 reviewers take issue with how this statement is constructed, I suggest this means it should be revisited. As currently worded, this statement excludes other studies that come to mind to the reviewers and therefore probably also the wider readership (primate gape as mentioned by R2, echidna shoulder ROM as explored in Regnault, Lai & Pierce 2021).

We definitely agree that misinterpretation of our intent by 2/4 Reviewers calls for rewording.

The work by Regnault et al. cited by Reviewer 4 includes lots of wonderful data about muscles, but the authors did not aim to collect data about joint articulation as it relates to interactions between articular surfaces, which is what we had more specifically hoped to convey here. We have therefore edited this statement in Line 59 to clarify that ground-truthing refers specifically to the collection of data about interactions between articular surfaces.

Point 23 (closed)

R2:

L81-82: In their response, the authors stressed that they were not discussing the “probability” of a joint ending up in a pose, but the writing here (“... mapping our articulation scores onto this mobility estimate enhances our knowledge of each pose beyond a binary “possible” or “impossible”...”) implies they are bringing joint articulation out of a deterministic, and into a probabilistic space. The authors are contradicting themselves.

Authors:

The Reviewer has repeated the same concern within this round of review. An “[enhanced] knowledge” is just that -- increased data. There is no allusion to probability, and we suspect that the relationship between articulation score and in vivo pose utilization is highly non-linear. We have already responded extensively above (and in the previous round of review) to the reviewer’s concerns about probability.

R4:

I agree that this discussion on probably has been covered in earlier points. My opinion as in those earlier points; I do not share the concern on probability.

Thank you. We agree. This matter is closed.

Point 24 (deemed redundant)

R2:

L86-90: This is a major issue with the 95-100 window, as it does not encompass all possible ranges of motion. As these envelopes of possible joint poses are being used as boundary constraints for models of locomotion for extinct animals, they must encompass the entire possible ROM for the extant animals, otherwise they are poorly defined constraints.

Authors:

The Reviewer has repeated the same concern within this round of review. We have already responded to this concern above. As stated, our goal was to reconstruct one of many possible walking stride cycles for *Deinonychus* using support from joint articulation data. We also point out that experimental data have revealed that locomotor poses occupy a small subset of a joint's full possible ROM (see Manafzadeh et al., 2021 PNAS). Therefore, contrary to the Reviewer's statement that "envelopes... must encompass the entire possible ROM for the extant animals, otherwise they are poorly defined constraints," we would argue that identifying the entire possible ROM instead of what we have done here would be a very poor constraint.

Manafzadeh, A. R., Kambic, R. E., & Gatesy, S. M. (2021). A new role for joint mobility in reconstructing vertebrate locomotor evolution. *Proceedings of the National Academy of Sciences*, 118(7), e2023513118.

R4:

This discussion has been covered in earlier points (Points 15 & 16). To save repeating, my opinion & concerns are the same as in those earlier points.

We agree; addressed in Points 15-16.

Point 25 (change made)

R2:

L95-98: Text needs to be added to the supplementary material explaining the sensitivity analyses. As it stands, it is not clear what sensitivity analysis was done from the figures and captions alone. Results from the sensitivity analysis should be explained as well.

Authors:

The methods already describe what was done, stating in current Lines 334-339 that:

“Translational sensitivity analysis was performed for guineafowl individual 1 by allowing 1,331 translational combinations over a larger range (two additional increments of equal size at each end of the X, Y, and Z translation ranges) at five-degree angular resolution, and sensitivity to articulation score formula was evaluated by conducting an analysis for this individual at the original rotational and translational resolution, weighting congruence by single-condyle overlap rather than full-joint overlap.”

The results already summarize the results, stating in current Lines 95-98 that:

“We found that our articulation score distributions are robust to sensitivity analyses conducted using additional individuals, increased translational allowance, and alternative score formulas (Supplementary Fig. 3-5).”

The captions of Supplementary Figures 3-5 offer both sets of information with additional detail, already stating that:

Supplementary Figure 3: “Results from the guineafowl individual figured in the main text, displayed at five-degree angular resolution in (a) and one-degree angular resolution in (d); a second individual, displayed at five-degree angular resolution in (b) and one-degree angular resolution in (e); and a third individual, displayed at five-degree angular resolution in (c) and one-degree angular resolution in (f), are grossly similar, especially within the highest-scoring region.” (emphasis added here)

Supplementary Figure 4: “Results from both the translational allowance throughout this paper, displayed at five-degree angular resolution in (a) and (b) and one-degree angular resolution in (c), and an increased translational allowance (see Methods), displayed at five-degree angular resolution in (d) and (e) and one-degree angular resolution in (f), demonstrate that although increasing translational allowance increases the region of pose space coded as viable (and therefore the region of pose space receiving articulation scores), the highest-scoring region of pose space remains identical.” (emphasis added here)

Supplementary Figure 5: “Results from both the articulation score formula implemented throughout this paper, displayed at five-degree angular resolution in (a) and one-degree angular resolution in (c), and an alternative formula that instead weights each condyle’s congruence by only its own overlap (rather than average overlap; see Methods), displayed at five-degree angular resolution in (b) and one-degree angular resolution in (d), are grossly similar.”

In the absence of more specific requests from the Reviewer, we are not sure what additional detail would be helpful. The details already provided allow full reproduction of the sensitivity analyses and offer our interpretation of their results.

R4:

The information appears to be provided in various places. However, perhaps R2's concern could be addressed by having this information collected together in one place (e.g. within Supplementary info, a section on sensitivity analyses that collates & repeats the methods and results) for the interested reader's ease of access?

In the spirit of this suggestion, but in lieu of making an entirely new Supplementary Information section, we have now repeated the information previously available in the methods and results in the Supplementary Figure Captions (Supplementary Figs. 3, 4, and 5).

Point 26 (change made)

R2:

L140-143: It was not the findings in this paper that enabled the reconstruction of the stride cycle, but “using the findings of this paper as constraints for XXX simulations...”

Authors:

We suggest this is semantic disagreement and respectfully decline the suggestion. R4:

I can see both sides of this point. Some would not consider this a reconstruction in the truest sense because existing avian kinematics were used as the starting point. The avian kinematic starting point is worth making clear here in my opinion because the Methods come after the Results / Discussion.

A suggestion for compromise between both views could be insertion of a short clarifier at line 140- 143 (e.g. suggested insertion underlined here: “Applying this finding to the foot of Deinonychus thus enabled us to reconstruct a potential six DOF stride cycle for a non-avian paravian dinosaur, from an avian kinematic start point, refining qualitative inferences made by previous workers based on observation and physical manipulation of fossil bones”).

We believe that the suggested insertion might inadvertently create further confusion, causing readers who do not engage with the Methods section to misinterpret the role that avian kinematics played in this study. Therefore, as an alternative compromise, we change the phrasing here in Line 169 to “constrain reconstruction of” rather than “reconstruct,” placing this sentence in alignment with phrasing in the Abstract and incorporating Reviewer 2’s desired concept of “constraints”.

This concludes the points of concern discussed by R2 and the authors in the Rebuttal document.

I have added my thoughts and comments below on the manuscript, from the first reading. Some of these comments are minor; others pick up on similar concerns to R2. As mentioned, I do not intend for these to saddle the authors with an additional review to address, but provide context for my opinions in the Points above. I summarise areas of major or common concern at the end of this document.

My (Reviewer 4) feedback on manuscript

We thank the Reviewer for going above and beyond and devoting the time to offer us with their own perspectives on our manuscript; we believe the resulting changes have improved the clarity of our submission. We reiterate as above that we sincerely appreciate the Reviewer's clarification that their own comments need not be taken as additional revision tasks.

Point 27 (changes made)

1) The method proposed in the manuscript is novel and impactful. I agree with the authors that the methods for defining articulation in joints can be made more objective, and applaud their innovative development of a new method for doing so. However, in parts of the manuscript it does feel as if existing work (which this method builds on) is downplayed or generalised, and/or the topic at hand has been narrowed to a degree, in order to emphasise its impact.

o An example is the text from line 46 onwards (“Current articulation criteria for virtual ROM analysis...”). Re: the critique of subjectivity on whether a joint “looks right”: arguably the use of shape primitives and anatomical joint axes to articulate digital joints are less subjective than this statement implies, and from my understanding this concept is in fact employed in this study too (in Methods, line 248-252, to set joint translation limits). Similarly, “have not been ground-truthed using data from extant animals”: I am aware of at least one other study beyond R2’s example, that evaluates how digital joint articulation choices affect ROM in an extant species (Point 22, above).

Addressed above in Point 22.

o Line 130; “existing computational articulation criteria, which rely on simple closest- point distances”. References for this statement? My understanding, perhaps outdated, is that many studies set articulation via shape primitives paired to bone surfaces. That seems like it is also “exploiting information about local articular surface curvature”, albeit in a different way to this study. If I haven’t misunderstood and that is what is referred to here, then ‘simple closest-point distances’ might need some unpacking or referencing, to avoid coming across as minimising other studies’ efforts towards objectivity.

We have softened our language to allow for the Reviewer’s interpretation that other existing approaches might also exploit local curvature in different ways, and added a citation to an example of a recent study (AutoBend) that relies on simple point-point distances, in Lines 154-156.

o Line 52 presents an opinion as established fact or widespread concern “This sobering conclusion calls into question [all studies to date]...” The conclusion follows from the authors’ previous work [26]. Is this questioning of all studies to date solely the authors’ opinion, based on their previous study? If so, consider clarifying (“In our opinion, this sobering conclusion calls into question...”) If it is an agreed concern that others have also expressed, can that be referenced?

We have made the suggested change (“In our opinion...”) in Line 63.

Point 28 (changes made)

2) I appreciate the pressure for concision in the manuscript, as well as the journal format having much of the detail (Methods) last, but this sometimes results in details that are very pertinent feeling obscured.

o Line 62-68 talks about analysing joints in two birds, comparing (validating) results against *in vivo* locomotor kinematics, and then using the validated method to reconstruct locomotor joint angles in an extinct dinosaur. However, in the Methods, the raycasting method seems to be only validated against the ankle and MTP of guineafowl (and actually predominantly the ankle) in locomotion. The emu ankle and MTP and most of the guineafowl MTP joint angles are analysed using still 2D photographs, where the bones are not visible. No IP joints are analysed or validated in extant birds. Then the raycasting method is applied to the MTP and IP joints of the extinct dinosaur (despite these joints not being the focus of the method development or validation). I am concerned that the lack of clarifying detail early on unintentionally misleads the reader and inflates the actual scope of the study. Rather than robustly analysing 3D/6 DoF movements of three joints in two extant species and application to a third extinct species (as implicit throughout earlier parts of the manuscript), the Methods reveal that the only common joint through method development, validation and application is the MTP of the guineafowl and the MTP of the extinct dinosaur. The manuscript critiques lack of ground truthing in extant species (fair critique; this does not appear to be widespread practice), but this study itself applies the methods to different joints (IP and MTP) from which it has been validated against (mostly ankle, some MTP).

We appreciate the Reviewer's concern. For all joints, we have used the best *in vivo* data available to us. Certainly, we could have restricted our study to the guineafowl ankle joint for which we have 6DoF XROMM-derived data, and then used the findings from this single joint to make broad-sweeping inferences about the function of bicondylar joints in dinosaur hindlimbs – and we note this practice would be in line with or exceeding the level of *in vivo* data most often included in existing studies similar to ours.

However, in an effort to be as rigorous as possible, we felt it important to prove to ourselves (and to the reader) that broader inference about the articulation of dinosaurian bicondylar hindlimb joints is well-supported by additional *in vivo* data. To do so, we tested for the presence of our articular score relationship in other joints from which we could obtain *in vivo* data – even if these further comparative data were not of the same complexity and quality as our XROMM-derived data. If a consistent relationship were to be present in both the bicondylar avian ankle and MTP joints, and in more than one species, we reasoned it was well-justified to extend its application to similarly bicondylar IP joints. In all tested cases (guineafowl and emu, ankle and MTP), the relationship between locomotor poses and articulation score was maintained, suggesting that our inferences about articulation in dinosaurian hindlimb bicondylar joints do indeed hold up more broadly, and forming the basis of our application of these results to the bicondylar pedal joints of *Deinonychus*.

To the Reviewer's related concern about the alignment of skeletal elements to light photographs – this practice is broadly common in decades of literature for the measurement of simple flexion-extension angles from videos of animal motion. We certainly agree that we would be skeptical of this practice for the measurement of any additional degrees of freedom, but we hoped simply to validate match with articulation

score distribution for excursions in the primary locomotor rotation for these additional joints. The emu images we used are protected by copyright and therefore we cannot reproduce them in the manuscript, but for the Reviewer's reference (and this can be redacted from open review if needed), here is an example of this alignment, demonstrating the minimal soft tissue at these joints and the ease with which a forward kinematic rig can be aligned accurately to them in, again, solely flexion-extension excursions:

[Redacted]

That said, we certainly do not want our manuscript to mislead readers into thinking that all of our data from extant animals are XROMM-derived. As a result, we made certain to be explicit in the Results that the *in vivo* data for joints other than the Guineafowl ankle are only for flexion-extension rotations. In case “experimentally derived” was misleading in our Abstract and Introduction and seemed to signal XROMM-derived data, we have removed this phrase from Lines 22 and 75.

o Lines 104 onwards: It is not clear here that the results refer only to locomotor pose space. E.g. “severely restricted LAR potential” and similar statements do make it sound as if the results are discussing total ROM. This is similar to R2's concern in Point 11 (above).

We have clarified that this phrase is in reference to locomotion in Line 127. See also our actions in response to Point 11, discussed above.

o Line 350, 353 “n=11,510” and “n=232” - these numbers are not very informative alone. How many individuals, and how many trials from each individual? How many are walking vs. running (especially important, given that the 95-100 window was decided based on including 100% of walking poses and 98% of running poses). Same questions for the guineafowl MTP (232 joint poses).

We have added the requested breakdowns to lines 391-397. We also direct the Reviewer to our response about sampling in Point 30, below.

o Line 357: “photographs available on the internet”: I'm sceptical about how powerful the 2D comparison of still images can be in validating articulation scores, without the bones/articular surfaces visible. If it is felt strongly that these data warrant inclusion in the manuscript (rather

than weakening it), more detail would be reassuring (are these animals walking/running? Can example photographs with the skeletal overlays be uploaded?)

Addressed above, in this Point.

o Line 368: “each joint” – Which joints were done in Deinonychus? It looks like the IP and MTP from the figures, but I don’t think this is clearly stated in the main manuscript.

We have now made this more explicit in Methods Lines 397-398.

Point 29 (changes made)

3) Some general questions on ray-casting method that might benefit from clarification:

o How can this method formalise “proper articulation” in terms of joint distance; can it account for ‘disarticulation’ (distraction; translation along the proximodistal axis)? Is this via the overlap score; and if so, is it possible to distract the joint in such a way that overlap remains high whilst joint distance is implausibly large? The joint spacing has the capacity to alter joint ROM estimates, but I imagine is tricky if the articular cartilage thickness and morphology are not known. Was joint spacing checked against the *in vivo* condition, once the bird legs were articulated?

These are interesting questions that came up in previous rounds of Review that it seems this Reviewer did not have access to (but we imagine will be available in open review history if others wish to see the discussion in the future), leading to changes already in the manuscript.

In brief, articular congruence suffers as joint surfaces move farther apart, even if overlap remains constant – this specific situation is demonstrated and described in Supplementary Figure 14c.

In this way, our articulation score analysis prevents high scores at unrealistic joint spacings, and for the animals studied here, in ways corresponding with true *in vivo* translations. Whether it will be equally successful for joints containing large amounts of articular cartilage remains will need to be tested by future studies involving *in vivo* kinematics from those joints.

o Line 319: Were other versions/weightings of the formula tested against *in vivo* data to see if there is a better/more optimal fit? Or does this formula choice best represent the underlying theory of what the balance of different scores ‘should’ look like for proper articulation?

Both! The sensitivity analysis displayed in Supplementary Figure 5 tests an alternative weighting of the formula, but it was selected to give equal importance to symmetry and congruence with emphasis on overlap. Although this formula revealed a useful relationship in our study, we acknowledge that modification of its specific weightings may be necessary to reveal similarly useful relationships in other taxa/joints, which we now make more explicit in Lines 411-415.

Point 30 (changes made)

4) Vague areas that could benefit from clarification

o Line 61: “viable” – what does that mean, in this context?

Changed to “non-interpenetrating.”

o Line 69: “locomotor morphology” – does this mean the morphology of the locomotor system (e.g. shape of the leg bones) or locomotor style as inferred from morphology, or something else?

The former was intended, though either interpretation would be fine in the context of the general point being made; no change.

o Line 89-90: Was there any bias in joints or species that fell outside the 95-100 window? (e.g. higher sampled joints such as the guineafowl ankle, or lower sampled/lower quality sampling such as the MTP or 2D photographs?)

The current discussion of poses falling outside the 95-100 window is limited to the guineafowl ankle, for which we have the most thorough sampling and best quality data. We refrain from such discussion for the additional joints studied because we acknowledge – as the Reviewer alludes to here – that their sampling is limited, and is only in one rotational degree of freedom; no change. As of interest, we note that 100% of all sampled poses fell within the 95-100 window for all other joints studied.

o Line 159: “efficient synovial joint motion” – what does efficient mean in this context?

This phrasing is pulled directly from the citation to MacConaill (in that sentence), who discusses efficiency in terms of minimal work necessary for motion; no change.

o Line 160-161: “stabbing or pinning” vs “slashing or digging” – for a layperson/unfamiliar reader, what would be the difference? (For example, to me slashing and stabbing seem quite similar. If I slash or stab with a knife, isn’t this more about the position/action of my proximal joints, rather than how I am holding the knife? Can it not slash with the claw in any position?)

The former classically refers to inserting the claw with a rigid digit, whereas the latter implies arc-like motion of the claw through flesh (using pose changes at more proximal joints) – clarification added to Lines 189-190 for less familiar readers.

o Line 246 parentheses “7x7x7 translations sampled using the prism-based hinge joint translation method proposed by [26]” – a reference is given but the explanation doesn’t add anything for the unfamiliar reader, it might as well just be the reference. A brief elaboration would be helpful of how translations were incorporated.

Supplementary Figure 10 displays this visually; we have now added a call to this Figure here (now Line 279).

Point 31 (changes made)

5) Minor: some parts of the manuscript are confusingly worded. I have listed below although I appreciate this may be overstepping my remit as a reviewer and straying into writing style preferences – apologies if so.

o Line 52 “precludes excluding very many poses at all” – the double negative of preclude and exclude is a bit confusing. Similarly the start of the sentence being a double negative (“not unlikely”) at Line 151.

We admit we like our double negatives here. Stylistic preference; no change.

o Line 52, 78 and 248 “conservative” – the term may mean different things to different people, and although its use is defined in-text via parentheses, it still could be needlessly confusing where another term could be clearer and more concise. To some, a conservative estimate of articulation or ROM is one that includes only poses that are definitely biologically possible in life – a safe or cautious estimate for inferring possible poses (i.e. minimising ‘false positives’). Whereas in the manuscript ‘conservative’ is used to mean estimates that might include biologically impossible poses as well as those used in life – the opposite, cautious in the sense that it does not eliminate possible as well as impossible poses (i.e. minimising ‘false negatives’). Neither is right or wrong, but another term (e.g. permissive, relaxed) would be clearer in my opinion.

The first instance noted has been changed to “permissive” as requested; the other two are maintained as “conservative” for consistency with our previous published work.

o Line 139 “articulation analysis successfully distinguishes the subset of joint poses used during dinosaurian locomotion”. I appreciate that the guineafowl and emu are dinosaurs, but use of dinosaurian without clarification here risks confusing readers (particularly non-dinosaur folks) that this could be referring to all dinosaurs including extinct ones. Although correct, for clarity the authors might consider referring to ‘birds’ when talking about the guineafowl/emu, or at least clarifying “during extant dinosaurian terrestrial locomotion” here.

Added “extant” as requested.

o Line 197: “third proximal pedal phalanx” – just a bit confusing, because it could sound like ‘third most proximal’. Suggest ‘proximal phalanx of digit III’?

While we would tend to agree, we chose this phrasing for consistency with existing literature; no change.

o Line 250: suggest rewording to bring condyle to the fore e.g. “fit to the condyle of the distal tibiotarsal...” just because it otherwise comes quite late in the sentence and otherwise isn’t clear that the cylinder is being fit to the condyle of each bone, rather than the whole bone.

Sentence reordered as requested.

o Line 274-276: Info in parentheses interrupts the subject and flow of the sentence. I’d suggest putting the clarification (which is appreciated!) elsewhere, such as the top of a sentence.

Another case of stylistic preference; no change.

R4 Summary

As mentioned previously, I feel the method proposed in the manuscript is novel and impactful. I agree with the authors that the methods for defining articulation in joints can be made more objective, and applaud their innovative development of a new method for doing so. I think it would be of interest to the journal's readership and to workers in the field.

There are some areas of common or outstanding concern, however, which I feel it would be necessary to address before publication. The discussion/rebuttal has become lengthy and many areas are repetitive examples of the same general topic, so I have tried to summarise the main points as I see them.

Firstly, issues around aims and generalisations of the manuscript, raised by reviewers in reference to various themes or statements (R2: Points 11, 12, 18, 22, and R4 Comment 1). These included the general messaging and purpose of the paper; the generalisations or narrowing of categories having the effect of minimising past work; and the important idea that this ray-casting method proposes to refine locomotor ROMs and not estimate general ROM.

Secondly, methodological concerns. R2 expresses repeated concern about the starting point of the extinct dinosaur locomotor reconstruction and related to that, I ask questions about the potential circularity of the starting point (R2: Points 12, 26). R2 and I also share interest in increased clarity, guidance, or comment on how the cutoff window for articulation scores is chosen (R2: Points 10, 15, 16, 17, 24, and R4 Comment 2).

Independently of R2, I have questions about the development and validation of the method primarily using guineafowl ankle data, and its subsequent application to the extinct dinosaurian metatarsophalangeal (MTP) and interphalangeal (IP) joints (R4 Comment 2). The quantity and quality of guineafowl MTP joint data and especially 2D emu data strike me as questionable. Why were no IP data used from extant animals given that these are most of the joints that are 'reconstructed' in *Deinonychus*' foot? How much are the external 2D photographs of the guineafowl and emu MTP and ankle joints actually able contribute? The inclusion of the latter feels hasty and combined with lack of clarity/detail very late in the Methods risks coming across as 'padding' that weakens reader confidence in the method & manuscript, rather than strengthens it as was probably intended. I am worried that it results in unfavourable perception of the study, when combined with the above concerns. This is a shame as I feel the method itself is genuinely exciting and addresses an important part of the field.

I hope that my contribution to this process has been useful and thank the authors, reviewers and editor again for the opportunity to comment.

We again thank Reviewer 4 for their thorough engagement with our work; we are grateful for their perspective and enthusiasm. The points summarized here were addressed as they appeared above.

Reviewers' Comments:

Reviewer #4:

Remarks to the Author:

Response to latest round of revisions:

I'd like to thank the Authors and Editor in inviting me to review the revisions made to the manuscript.

I'm glad that my comments were of assistance, and also thank the authors for summarising the actions taken in an easy-to-review way. The only thing that was a little tricky was following the exact places where specific changes had been made, as the line numbers cited in the response don't seem to match up with the numbers in the most recent version of the manuscript. However, on reading the revised manuscript, it seems much clearer in messaging and methodology thanks to the authors' changes. These were the major areas of discussion/concern between Reviewer 2, myself, and the authors previously and I have no outstanding concerns. I recommend that the manuscript be published.

I only have a few minor suggestions from the most recent edits which I'd recommend be made, to improve clarity. However, these are very minor and I don't feel the need to review the manuscript again.

Final recommended changes:

i. Line 21 (abstract) re: "We applied our methodology to four bicondylar hindlimb joints of two extant dinosaurs".

I would recommend changing / clarifying the four joints part, as it is unintentionally misleading to the reader. It sounds as if four different anatomical joints were examined, rather than 2x ankle and 2x MTP. Suggest something like "We applied our methodology to bicondylar hindlimb joints (ankle, metatarsophalangeal) of two extant dinosaurs" or "We applied our methodology to the bicondylar ankle and metatarsophalangeal joints of two extant dinosaurs"?

ii. Similar to above, Line 379 re: "avian bicondylar hindlimb joints". I recommend being clear that these were the ankle and MTP, applied to the MTP and IP joints of *Deinonychus*. The authors reasoning as explained in Response to point 28 was perfectly logical (specifically, "we reasoned it was well-justified to extend its application to similarly bicondylar IP joints"). That reasoning is perhaps not so evident to the reader in the final manuscript, so it would be great to include (concisely, 1-2 sentences) somewhere by way of explanation – particularly to pre-empt from the reader similar concern / discussion that we had about applicability of the method developed/validated against different anatomical joints (predominantly ankle vs. IP).

iii. Line 110 re: "By using avian locomotor kinematics..." & Line 370 "published extant avian toe tip motion"

I recommend clarifying the species of the avian locomotor kinematics, and noting that this is a different dataset than the avian locomotor kinematics used here in calculating the articular scores. Don't make the reader hunt through references for this info. Otherwise, the risk is – as the authors mention in their Response to Point 26 – that the reader might be confused which data are being used as the starting point and/or concerned about circularity of the constrained reconstruction.

iv. Line 363 & Point 28 re: photographs from the internet.

Thanks to the authors for clarifying that this is a commonly accepted practice. It was helpful to me to see the illustrated example, and it may be to other readers too. I empathise with the minefield of copyright issues, but I have had success in requesting reproduction of copyrighted images (without payment) from copyright holders. Due to how helpful the illustration is, if at all possible, I do recommend an example in the supplementary information (similar to the one provided in the Response).

v. Supplementary video 4 appears corrupted (unable to play) – apologies that I forgot to mention it in the last review.

Sophie Regnault (Reviewer 4)

MANAFZADEH ET AL. RESPONSES TO REVIEWERS VERSION 5

Reviewer #4 (Remarks to the Author):

Response to latest round of revisions:

I'd like to thank the Authors and Editor in inviting me to review the revisions made to the manuscript.

I'm glad that my comments were of assistance, and also thank the authors for summarising the actions taken in an easy-to-review way. The only thing that was a little tricky was following the exact places where specific changes had been made, as the line numbers cited in the response don't seem to match up with the numbers in the most recent version of the manuscript. However, on reading the revised manuscript, it seems much clearer in messaging and methodology thanks to the authors' changes. These were the major areas of discussion/concern between Reviewer 2, myself, and the authors previously and I have no outstanding concerns. I recommend that the manuscript be published.

I only have a few minor suggestions from the most recent edits which I'd recommend be made, to improve clarity. However, these are very minor and I don't feel the need to review the manuscript again.

We thank Dr. Regnault once again.

Final recommended changes:

i. Line 21 (abstract) re: "We applied our methodology to four bicondylar hindlimb joints of two extant dinosaurs". I would recommend changing / clarifying the four joints part, as it is unintentionally misleading to the reader. It sounds as if four different anatomical joints were examined, rather than 2x ankle and 2x MTP. Suggest something like "We applied our methodology to bicondylar hindlimb joints (ankle, metatarsophalangeal) of two extant dinosaurs" or "We applied our methodology to the bicondylar ankle and metatarsophalangeal joints of two extant dinosaurs"?

We have modified this portion of the text (Line 21) to avoid confusion. Given the strict word limit on the Abstract, we have chosen to omit "four."

ii. Similar to above, Line 379 re: "avian bicondylar hindlimb joints". I recommend being clear that these were the ankle and MTP, applied to the MTP and IP joints of Deinonychus. The authors reasoning as explained in Response to point 28 was perfectly logical (specifically, "we reasoned it was well-justified to extend its application to similarly bicondylar IP joints"). That reasoning is perhaps not so evident to the reader in the final manuscript, so it would be great to include (concisely, 1-2 sentences) somewhere by way of explanation – particularly to pre-empt from the reader similar concern / discussion that we had about applicability of the method developed/validated against different anatomical joints (predominantly ankle vs. IP).

As above, we have modified this portion of the text (around Line 379) to avoid confusion. Given the increased freedom for word count here, we have made our language more specific about the specific joints being referred to instead of the more vague 'bicondylar'.

iii. Line 110 re: "By using avian locomotor kinematics..." & Line 370 "published extant avian toe tip motion" I recommend clarifying the species of the avian locomotor kinematics, and noting that this is a different dataset than the avian locomotor kinematics used here in calculating the articular scores. Don't make the reader hunt through references for this info. Otherwise, the risk is – as the authors mention in their Response to Point 26 – that the reader might be confused which data are being used as the starting point and/or concerned about circularity of the constrained reconstruction.

We have clarified 'guineafowl' in the locations indicated.

iv. Line 363 & Point 28 re: photographs from the internet.

Thanks to the authors for clarifying that this is a commonly accepted practice. It was helpful to me to see the illustrated example, and it may be to other readers too. I empathise with the minefield of copyright issues, but I have had success in requesting reproduction of copyrighted images (without payment) from copyright holders. Due to how helpful the illustration is, if at all possible, I do recommend an example in the supplementary information (similar to the one provided in the Response).

Upon reviewing this journal's policies, we regret that we will be unable to reproduce the image offered in the responses to reviewers.

v. Supplementary video 4 appears corrupted (unable to play) – apologies that I forgot to mention it in the last review.

We are not sure why this was the case, as the video plays fine on our end and this issue was not raised in previous rounds of review. It is possible that there was an issue with upload/conversion on the journal site for this latest resubmission. In any case, we appreciate the Reviewer pointing this out, and we have exported and uploaded a fresh copy of the video in case it helps.

Sophie Regnault (Reviewer 4)